# MODEL COMPRESSION VIA SYMMETRIES OF THE PARAMETER SPACE

## ABSTRACT

We provide a theoretical framework for neural networks in terms of the representation theory of quivers, thus revealing symmetries of the parameter space of neural networks. An exploitation of these symmetries leads to a model compression algorithm for radial neural networks based on an analogue of the QR decomposition. The algorithm is lossless; the compressed model has the same feedforward function as the original model. If applied before training, optimization of the compressed model by gradient descent is equivalent to a projected version of gradient descent on the original model.

## 1 INTRODUCTION

Recent work has shown that representation theory, the formal study of symmetry, provides the foundation for various innovative techniques in deep learning (Cohen & Welling, 2016; Kondor & Trivedi, 2018; Ravanbakhsh et al., 2017; Cohen & Welling, 2017). Much of this previous work considers symmetries inherent to the input and output spaces, as well as distributions and functions that respect these symmetries. By contrast, in this paper, we expose a broad class of symmetries intrinsic to the parameter space of the neural networks themselves. We use these parameter space symmetries to devise a model compression algorithm that reduces the widths of the hidden layers, and hence the number of parameters. Unlike representation-theoretic techniques in the setting of equivariant neural networks, our methods are applicable to deep learning models with non-symmetric domains and non-equivariant functions, and hence pertain to some degree to all neural networks.

Specifically, we formulate a theoretical framework for neural networks inspired by quiver representation theory, a mathematical field with connections to symplectic geometry and Lie theory (Kirillov Jr, 2016; Nakajima et al., 1998). This approach builds on that of Armenta & Jodoin (2020) and of Wood & Shawe-Taylor (1996), but is arguably simpler and encapsulates larger symmetry groups. Formally, a *quiver* is another name for a finite directed graph, and a *representation* of a quiver is the assignment of a vector space to each vertex and a compatible linear map to each edge.

Our starting point is to regard the vector space of parameters for a neural network as a representation of a particular quiver, namely, the *neural quiver* (Figure 1). The advantage of this viewpoint is that representations of quivers carry rich symmetry groups via change-of-basis transformations; such operations can be viewed as symmetries of the neural network parameter space.

Moreover, these symmetries may be factored out without affecting the feedforward function, making our method a lossless model compression algorithm (Serra et al., 2020). Model compression has

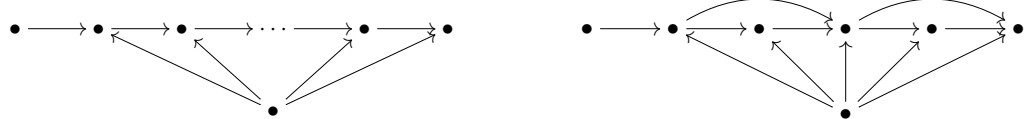

Figure 1: (a) Left: the neural quiver. Each vertex in the top row represents a layer of the neural network and each arrow indicates a linear mapping. The bottom vertex is the "bias" vertex. (b) Right: a quiver with skip connections.

become critically important as models have grown to billions of parameters; with compression, enormous models may be reduced and run on smaller systems with faster inference (Buciluǎ et al., 2006; Cheng et al., 2017; Frankle & Carbin, 2018; Zhang et al., 2018). Whereas many previous approaches to model compression are based on weight pruning, quantization, matrix factorization, or knowledge distillation, similar to Sourek et al. (2020), our approach exploits symmetries of neural network parameter spaces.

The size of the parameter space symmetry group is determined by properties of the activation functions. We focus on radial activation functions, as in Weiler & Cesa (2019); Sabour et al. (2017); Weiler et al. (2018a;b); these interact favorably with certain QR decompositions, and, consequently, the model compression is significant compared to the more common pointwise (or 'local') activations. We refer to neural networks with radial activation functions as *radial neural networks*.

Given a radial neural network, our results produce a new network with fewer neurons in each hidden layer and the same feedforward function, and hence the same loss for any batch of training data. Moreover, the value of the loss function after a step of gradient descent applied to the compressed model is the same as the value of the loss function after a step of *projected gradient descent* applied to the original model. As we explain below, in projected gradient descent, one subtracts a truncation of the gradient, rather than the full gradient. When the compression is significant enough, the compressed model takes less time per epoch to train and reaches local minima faster.

To state these results slightly more precisely, recall that the parameters of a neural network with layer widths $(n_0, n_1, \ldots, n_L)$ consist of an $n_i \times (n_{i-1}+1)$ matrix $W_i$ of weights for each layer $i$, where we include the bias as an extra column. These are grouped into a tuple $\mathbf{W} = \left(W_i \in \mathbb{R}^{n_i \times (n_{i-1}+1)}\right)_i$. We define the *reduced widths* recursively as $n_i^{\mathrm{red}} = \min\left(n_i\,,\,n_{i-1}^{\mathrm{red}}+1\right)$ for $i = 1, \ldots, L-1$, with $n_0^{\mathrm{red}} = n_0$ and $n_L^{\mathrm{red}} = n_L$. Note that $n_i^{\mathrm{red}} \leq n_i$ for all $i$.

**Theorem 1.1** (Informal version of Theorems 4.3 and 4.7). *Suppose a neural network has $L$ layers with widths $(n_0, \ldots, n_L)$, parameters $\mathbf{W}$, and radial activation functions. Let $f_\mathbf{W} : \mathbb{R}^{n_0} \to \mathbb{R}^{n_L}$ be the feedforward function of the network.*

1. *There exists an algorithm to produce a reduced radial neural network with layer widths $(n_0, n_1^{\mathrm{red}}, \ldots, n_{L-1}^{\mathrm{red}}, n_L)$, parameters $\mathbf{R}$, and the same feedforward function $f_\mathbf{R} = f_\mathbf{W}$.*

2. *Training $f_\mathbf{R}$ with gradient descent is an equivalent optimization problem to training $f_\mathbf{W}$ with projected gradient descent.*

This theorem can be interpreted as a model compression result: the reduced (or compressed) neural network $\mathbf{R}$ has the same accuracy as the original neural network $\mathbf{W}$, and there is an explicit relationship between the gradient descent optimization problems for the two neural networks. This result is not just theoretical; it emerges from a practical and efficient algorithm depending on successive QR decompositions. We describe this procedure below (Algorithm 1) and implement it in Python.

To summarize, our contributions are as follows:

1. A theoretical framework for neural networks based on quiver representation theory;
2. A QR decomposition for radial neural networks;
3. An implementation of a lossless model compression algorithm for radial neural networks;
4. A theorem relating gradient descent optimization of the original and compressed networks.

We view this work as a step in the direction of improving learning algorithms by exploiting symmetry inherent to neural network parameter spaces. As such, we expect our framework and results to generalize in several ways, including: (1) further reductions of the hidden widths, (2) incorporating certain non-radial activation functions, (3) encapsulating networks beyond MLPs, such as convolutional, recurrent, and graph neural networks, (4) integration of regularization techniques. We include a detailed discussion of the limitations of our results, as well as future directions, in Section 6.

## 2 RELATED WORK

**Quiver Representation Theory and Neural Networks.** Armenta & Jodoin (2020) give an approach to understanding neural networks in terms of quiver representations. Our work generalizes

their approach as it (1) accommodates both pointwise and non-pointwise activation functions, (2) taps into larger symmetry groups, and (3) connects more naturally to gradient descent. Jeffreys & Lau (2021) and Manin & Marcolli (2020) also place quivers in the context of neural networks; our approach is inspired by similar algebro-geometric and categorical perspectives, but differs in our emphasis on practical consequences for optimization techniques at the core of machine learning. These works have a number of precursors. One is the study of the "non-negative homogeneity" (also known as "positive scaling invariance") property of ReLU activation functions (Dinh et al., 2017; Neyshabur et al., 2015; Meng et al., 2019), which is a special case of the symmetry studied in this paper. Wood & Shawe-Taylor (1996) regard layers in a neural network as representations of finite groups and consider only pointwise activation functions; by contrast, our framework captures Lie groups as well as non-pointwise activations. Our quiver approach to neural networks shares parallels with the "algebraic neural networks" of Parada-Mayorga & Ribeiro, and special cases of their formalism amount to representations of quivers over base rings beyond $\mathbb{R}$, such as the ring of polynomials. In a somewhat different data-analytic context, Chindris & Kline (2021) use quiver representation theory in order to untangle point clouds; though they do not use neural networks.

**Equivariant Neural Networks.** Previously, representation theory has been used to design neural networks which incorporate symmetry as an inductive bias. A variety of architectures such as $G$-convolution, steerable CNN, and Clebsch-Gordan networks are constrained by various weight-sharing schemes to be equivariant or invariant to various symmetry groups (Cohen et al., 2019; Weiler & Cesa, 2019; Cohen & Welling, 2016; Chidester et al., 2018; Kondor & Trivedi, 2018; Bao & Song, 2019; Worrall et al., 2017; Cohen & Welling, 2017; Weiler et al., 2018b; Dieleman et al., 2016; Lang & Weiler, 2021; Ravanbakhsh et al., 2017). Our approach, in contrast, does not rely on symmetry of the input domain, output space, or mapping. Rather, our method exploits symmetry of the parameter space and thus applies more generally to domains with no obvious symmetry. From the point of view of model compression, equivariant networks do achieve reduction in the number of trainable parameters through weight-sharing for fixed hidden dimension widths; however, in practice, they may use larger layer widths and consequently have larger memory requirements than non-equivariant models. Sampling or summing over large symmetry groups may make equivariant models computationally slow as well (Finzi et al., 2020; Kondor & Trivedi, 2018).

**Model Compression and Weight Pruning.** A major goal in machine learning is to find methods for compressing models in order to reduce the number of trainable parameters, decrease memory usage, or accelerate inference and training (Cheng et al., 2017). Our approach toward this goal differs significantly from most existing methods in that it is based on the inherent symmetry of neural network parameter spaces. One prior model compression method is *weight pruning*, which removes redundant, small, or unnecessary weights from a network with little loss in accuracy (Han et al., 2015; Blalock et al., 2020; Karnin, 1990). Recent work has shown pruning can be done during training by identifying and removing weights of less relevance and reverting other weights to earlier values (Frankle & Carbin, 2018). Related work shows effective pruning may be done at initialization (Lee et al., 2019; Wang et al., 2020). *Gradient-based pruning* identifies low saliency weights by estimating the increase in loss resulting from their removal (LeCun et al., 1990; Hassibi & Stork, 1993; Dong et al., 2017; Molchanov et al., 2016). A complementary approach is *quantization*, in which the bit depth of weights is decreased (Wu et al., 2016; Howard et al., 2017; Gong et al., 2014). *Knowledge distillation* works by training a small model to mimic the performance of a larger model or ensemble of models (Buciluǎ et al., 2006; Hinton et al., 2015; Ba & Caruana, 2013). *Matrix Factorization* methods replace fully connected layers with lower rank or sparse factored tensors (Cheng et al., 2015a;b; Tai et al., 2015; Lebedev et al., 2014; Rigamonti et al., 2013; Lu et al., 2017) and can often be applied before training. Our method involves a generalized QR decomposition, which is a type of matrix factorization; however, rather than aim for a rank reduction of linear layers, we leverage this decomposition in order to reduce hidden layer widths via change-of-basis operations on the hidden representations. Closest to our method are lossless compression methods. Serra et al. (2021; 2020) identify and remove stable neurons in ReLU networks. Sourek et al. (2020) also exploit symmetry in parameter space to remove redundant neuron. However, their symmetries are induced from permutation equivariance; ours follow from the symmetries of the radial activation.

## 3 NEURAL NETWORKS AS QUIVER REPRESENTATIONS

We turn our attention to quivers and their representations. Neural networks may be viewed as representations of a specific quiver with the additional data of activation functions; backpropogation can also be formulated in these terms. The advantage of such perspectives is that representations come equipped with symmetry groups. We provide proofs and context for this section in Appendix D.

### 3.1 QUIVER REPRESENTATIONS

A **quiver** is a finite directed graph. A **dimension vector** for a quiver $\mathcal{Q}$ consists of a non-negative integer for every vertex, that is, a tuple $\mathbf{d} = (d_i)_i$ indexed by the vertices of $\mathcal{Q}$. A **representation** of $\mathcal{Q}$ with dimension vector $\mathbf{d}$ consists of a matrix for each edge, where the matrix $A_e$ corresponding to an edge $\overset{s}{\bullet} \overset{e}{\longrightarrow} \overset{t}{\bullet}$ must be of size $d_t \times d_s$. In other words, a representation is a tuple of matrices $(A_e)_e$ indexed by the edges of $\mathcal{Q}$, with sizes constrained by the dimension vector. Note that each $A_e$ defines a linear map $\mathbb{R}^{d_s} \to \mathbb{R}^{d_t}$. The **space of representations** $\mathsf{Rep}(\mathcal{Q}, \mathbf{d})$ is defined as the set of all possible representations of a quiver $\mathcal{Q}$ with dimension vector $\mathbf{d}$. This space can be expressed as the product of matrix spaces: $\mathsf{Rep}(\mathcal{Q}, \mathbf{d}) = \prod_{e \in E} \mathbb{R}^{d_{t(e)} \times d_{s(e)}}$. Here $E$ is the edge set of $\mathcal{Q}$, and $s(e)$ and $t(e)$ denote the source and target vertices of an edge $e$.

### 3.2 THE NEURAL QUIVER

For the remainder of the paper, we focus on a particular quiver. Let $L$ be a positive integer. The **neural quiver** $\mathcal{Q}_L$ is the quiver with $L+2$ vertices shown in Figure 1. The vertices in the top row are indexed from $i = 0$ to $i = L$. The first vertex ($i = 0$) in the top row is called the **input vertex**, the last vertex ($i = L$) is the **output vertex**, and every other vertex ($1 \leq i \leq L - 1$) in the top row is a **hidden vertex**. The vertex at the bottom is called the **bias vertex** and is connected to each vertex in the top row except for $i = 0$. When discussing representations of the neural quiver, we exclusively consider dimension vectors whose value at the bias vertex is 1. Hence, a dimension vector for the neural quiver will refer to an $(L+1)$-tuple of positive integers $\mathbf{n} = (n_0, \ldots, n_L)$.

With these conventions in hand, we observe that, by definition, a representation of the neural quiver with dimension vector $\mathbf{n}$ consists of an $n_i \times n_{i-1}$ matrix $A_i$ and a vector $b_i \in \mathbb{R}^{n_i}$, for each $i = 1, \ldots, L$. Correspondingly, there is an affine map $\widetilde{W}_i : \mathbb{R}^{n_{i-1}} \to \mathbb{R}^{n_i}$ given by $x \mapsto A_i x + b_i$, and an $n_i \times (1 + n_{i-1})$ matrix given by $W_i = [b_i \ A_i]$. As a result, we have:

**Lemma 3.1.** *A representation of $\mathcal{Q}_L$ with dimension vector $\mathbf{n}$ is equivalent to the data of an affine map $\mathbb{R}^{n_{i-1}} \to \mathbb{R}^{n_i}$ for each $i = 1, \ldots, L$. There is an isomorphism of vector spaces:*

$$\mathsf{Rep}(\mathcal{Q}_L, \mathbf{n}) \simeq \mathbb{R}^{n_1 \times (1+n_0)} \times \mathbb{R}^{n_2 \times (1+n_1)} \times \cdots \times \mathbb{R}^{n_L \times (1+n_{L-1})}.$$

**Notation**: We henceforth denote a representation of $\mathcal{Q}_L$ as a tuple $\mathbf{W} = (W_i)_{i=1}^L$, where each $W_i$ belongs to $\mathbb{R}^{n_i \times (1+n_{i-1})}$, and denote the corresponding affine maps as $\widetilde{W}_i : \mathbb{R}^{n_{i-1}} \to \mathbb{R}^{n_i}$.

### 3.3 CHANGE-OF-BASIS SYMMETRY

Recall that the **general linear group** $\mathrm{GL}_n(\mathbb{R})$ consists of the set of invertible $n$ by $n$ matrices. The unit is the identity $n$ by $n$ matrix, denoted $\mathrm{id}_n$. There is a linear action of the general linear group $\mathrm{GL}_n(\mathbb{R})$ on the vector space $\mathbb{R}^n$ given by matrix multiplication: $\vec{x} \mapsto G\vec{x}$ for $G \in \mathrm{GL}_n(\mathbb{R})$ and $\vec{x} \in \mathbb{R}^n$. This action encodes change-of-basis transformations: the entries of the vector $G\vec{x}$ are the coefficients when expressing the original vector $\vec{x}$ in the basis specified by the columns of $G^{-1}$.

Given a dimension vector $\mathbf{n}$ for the neural quiver, the **parameter symmetry group** is defined as:

$$\mathrm{GL}_{\mathbf{n}}^{\mathrm{hidden}} := \mathrm{GL}_{n_1}(\mathbb{R}) \times \mathrm{GL}_{n_2}(\mathbb{R}) \times \cdots \times \mathrm{GL}_{n_{L-1}}(\mathbb{R})$$

An element of this group consists of the choice of a transformation of $\mathbb{R}^{n_i}$ for each hidden vertex $i$, and results in a corresponding transformation of the matrices $W_i$ appearing in any representation $\mathbf{W}$ of the neural quiver with dimension vector $\mathbf{n}$. To be explicit, a particular choice $\mathbf{G} = (G_i)_{i=1}^{L-1}$

of transformations results in the following linear transformation of representations[1]:

$$\mathbf{W} = (W_i)_{i=1}^L \quad \mapsto \quad \mathbf{G} \cdot \mathbf{W} := \left( G_i \, W_i \begin{bmatrix} 1 & 0 \\ 0 & G_{i-1}^{-1} \end{bmatrix} \right)_{i=1}^L, \tag{3.1}$$

where $G_0 = \mathrm{id}_{n_0}$ and $G_L = \mathrm{id}_{n_L}$. In other words: there is a linear action of the parameter symmetry group $\mathrm{GL}_{\mathbf{n}}^{\mathrm{hidden}}$ on the vector space $\mathrm{Rep}(\mathcal{Q}_L, \mathbf{n})$ of representations of the neural quiver.

## 3.4 Neural networks

For simplicity, in the body of this paper, we focus on the most basic neural network architecture, namely that of multi-layer perceptrons (MLPs). These the neural networks whose underlying architecture is specified by the neural quiver. We include a discussion of more complex quivers (such as that in Figure 1b) in the appendix. We begin with the following somewhat abstract definition:

**Definition 3.2.** An $\mathcal{Q}_L$-**neural network** consists of the following:

1. **Hyperparameters.** A positive integer $L$ and a dimension vector $\mathbf{n} = (n_0, n_1, n_2, \ldots, n_L)$ for the neural quiver $\mathcal{Q}_L$.

2. **Trainable parameters.** A representation $\mathbf{W} = (W_1, \ldots, W_L)$ of the neural $\mathcal{Q}_L$ with dimension vector $\mathbf{n}$. So each $W_i$ is a matrix in $\mathbb{R}^{n_i \times (1+n_{i-1})}$.

3. **Activation functions.** Piece-wise differentiable functions $a_i : \mathbb{R}^{n_i} \to \mathbb{R}^{n_i}$ for $i = 1, \ldots, L$. These are grouped into a tuple $\mathbf{a} = (a_1, \ldots, a_L)$.

For the remainder of this paper, unless specified otherwise, "neural network" will refer to "$\mathcal{Q}_L$-neural network". We emphasize, however, that there is a general notion of a $\mathcal{Q}$-neural network for any acyclic quiver $\mathcal{Q}$ with unique sink (see Appendix D). Additionally, as indicated above, in the case of the neural quiver, a $\mathcal{Q}_L$-neural network is equivalent to an MLP.

We denote by $\mathsf{Neur}(\mathcal{Q}_L, \mathbf{n})$ the vector space of neural networks with $L$ layers and dimension vector $\mathbf{n}$, and regard elements therein as pairs $(\mathbf{W}, \mathbf{a})$. The **neural function** (or **feedforward function**) of a $\mathcal{Q}_L$-neural network $(\mathbf{W}, \mathbf{a})$ is defined as:

$$f_{(\mathbf{W},\mathbf{a})} : \mathbb{R}^{n_0} \to \mathbb{R}^{n_L} \qquad \vec{x} \mapsto a_L \circ \widetilde{W}_L \circ \cdots \circ a_2 \circ \widetilde{W}_2 \circ a_1 \circ \widetilde{W}_1(\vec{x})$$

where $\widetilde{W}_i : \mathbb{R}^{n_{i-1}} \to \mathbb{R}^{n_i}$ is the affine map corresponding to $W_i$ (see Section 3.2).

The optimization problem for neural networks can be formulated in terms of representations of the neural quiver. Fix a dimension vector $\mathbf{n}$ for $\mathcal{Q}_L$, and activation functions $\mathbf{a}$. To a batch of training data $\{(\vec{x}_j, \vec{y}_j)\} \subseteq \mathbb{R}^{n_0} \times \mathbb{R}^{n_L}$, the associated **loss function** on the space $\mathrm{Rep}(\mathcal{Q}_L, \mathbf{n})$ is defined as

$$\mathcal{L} : \mathrm{Rep}(\mathcal{Q}_L, \mathbf{n}) \to \mathbb{R} \qquad \mathcal{L}(\mathbf{W}) = \sum_j \mathcal{C}\left( f_{(\mathbf{W},\mathbf{a})}(\vec{x}_j), \vec{y}_j \right)$$

where $\mathcal{C} : \mathbb{R}^{n_L} \times \mathbb{R}^{n_L} \to \mathbb{R}$ is a cost function. The map

$$\gamma : \mathrm{Rep}(\mathcal{Q}_L, \mathbf{n}) \to \mathrm{Rep}(\mathcal{Q}_L, \mathbf{n}) \qquad \gamma(\mathbf{W}) = \mathbf{W} - \eta \nabla_{\mathbf{W}} \mathcal{L}$$

performs one step of gradient descent with learning rate $\eta$. Hence, the backpropagation algorithm can be regarded as taking place on the space $\mathrm{Rep}(\mathcal{Q}_L, \mathbf{n})$ of representations of the neural quiver.

## 4 Main theoretical results

In this section, we first define radial neural networks. Then, adopting the framework developed in the previous section, we proceed to state a QR decomposition for such networks, present an algorithm to compute the decomposition, and relate the decomposition to gradient descent.

---

[1]For each $i$, the matrix $\begin{bmatrix} 1 & 0 \\ 0 & G_{i-1}^{-1} \end{bmatrix}$ is block diagonal with one $1 \times 1$ block consisting of the single entry '1', and one $n_{i-1} \times n_{i-1}$ block consisting of the inverse of the matrix $G_{i-1} \in \mathrm{GL}_{n_{i-1}}(\mathbb{R})$.

### 4.1 RADIAL NEURAL NETWORKS

We begin by stating the definition of radial functions. A piece-wise differentiable function $a : \mathbb{R}^n \to \mathbb{R}^n$ is called **radial** if $a(v) = h(|v|)\frac{v}{|v|}$ for some $h : \mathbb{R} \to \mathbb{R}$. Hence, $a$ sends each input vector $v \in \mathbb{R}^n$ to a scalar multiple of itself, and that scalar depends only on the norm of the vector.

**Example 4.1.** (1) The squashing function, where $h(r) = \frac{r^2}{r^2+1}$. (2) Shifted ReLU, where $h(r) =$ ReLU$(r - b)$ for $r > 0$ a real number $b$. See Weiler & Cesa (2019) and the references therein for more examples and discussion of radial functions. See Figure 2

Let $\mathbf{n}$ be a dimension vector for the neural quiver $\mathcal{Q}_L$. A neural network $(\mathbf{W}, \mathbf{a}) \in$ Neur$(\mathcal{Q}_L, \mathbf{n})$ is a **radial neural network** if each $a_i$ is a radial function on $\mathbb{R}^{n_i}$. Hence, there are functions $h_i : \mathbb{R} \to \mathbb{R}$ such that $a_i(v) = h_i(|v|)\frac{v}{|v|}$ for each $i$. Let $O(\mathbf{n})$ denote the following product of the orthogonal groups:

$$O(\mathbf{n}) = O(n_1) \times O(n_2) \times \cdots \times O(n_{L-1}).$$

Note that $O(\mathbf{n})$ is a subgroup of the parameter symmetry group GL$_\mathbf{n}^\text{hidden}$ and hence acts on Rep$(\mathcal{Q}, \mathbf{n})$. Radial functions have the flexibility of defining the activation functions of a neural network with arbitrary dimension vector. To be more explicit:

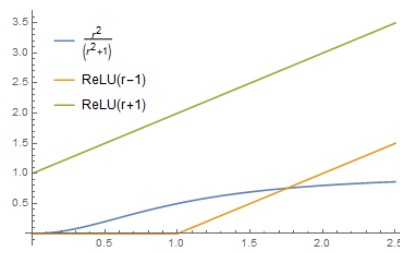

Figure 2: Different radial activations.

**Construction 4.2.** Suppose $\mathbf{h} = (h_1, \ldots, h_L)$ is a tuple of functions[2] $\mathbb{R} \to \mathbb{R}$, and suppose $\mathbf{n}$ is a dimension vector for $\mathcal{Q}_L$. Consider the tuple of activation functions: $\mathbf{a}_\mathbf{h}^{(\mathbf{n})} = \left( h_1^{(n_1)}, \ldots, h_L^{(n_L)} \right)$.

Any representation $\mathbf{W}$ of $\mathcal{Q}_L$ with dimension vector $\mathbf{n}$ defines a neural network $\left( \mathbf{W}, \mathbf{a}_\mathbf{h}^{(\mathbf{n})} \right)$ as in Definition 3.2. We write simply $(\mathbf{W}, \mathbf{h})$ for this network and $f_{(\mathbf{W}, \mathbf{h})}$ for its neural function.

### 4.2 THE QR DECOMPOSITION

We now formulate the QR decomposition for representations of the neural quiver, which interacts favorably with radial neural networks. Proofs and further context appear in Appendix E.

The first step is to associate a 'reduced' dimension vector to each dimension vector of the neural quiver. Let $\mathbf{n} = (n_0, n_1, \ldots, n_L)$ be a dimension vector for $\mathcal{Q}_L$. Its **reduction** $\mathbf{n}^\text{red} = (n_0^\text{red}, n_1^\text{red}, \ldots, n_L^\text{red})$ is defined recursively[3] by setting $n_0^\text{red} = n_0$, then $n_i^\text{red} = \min(n_i , n_{i-1}^\text{red} + 1)$ for $i = 1, \ldots, L - 1$, and finally $n_L^\text{red} = n_L$. We abbreviate Rep$(\mathcal{Q}_L, \mathbf{n})$ by Rep$(\mathbf{n})$ and Rep$(\mathcal{Q}_L, \mathbf{n}^\text{red})$ by Rep$(\mathbf{n}^\text{red})$. Observe that $n_i^\text{red} \leq n_i$ for each $i$. Therefore, taking a representation with dimension vector $\mathbf{n}^\text{red}$, i.e., a certain tuple of matrices, one can pad each matrix with the necessary number of rows of zeros below and columns of zeros on the right in order to produce a representation with dimension vector $\mathbf{n}$. Thus we have an inclusion: $\iota : $ Rep$(\mathbf{n}^\text{red}) \hookrightarrow$ Rep$(\mathbf{n})$.

Fix a tuple $\mathbf{h} = (h_1, \ldots, h_L)$ of functions $h_i : \mathbb{R} \to \mathbb{R}$. As in Construction 4.2, we attach a radial neural network (and hence a neural function) to every representation of $\mathcal{Q}_L$.

**Theorem 4.3.** *Let* $\mathbf{n}$ *be a dimension vector and fix* $\mathbf{h}$ *as above. For any* $\mathbf{W} \in$ Rep$(\mathbf{n})$ *there exist:*

$$\mathbf{Q} \in O(\mathbf{n}), \qquad \mathbf{R} = (R_1, \ldots, R_L) \in \text{Rep}(\mathbf{n}^\text{red}), \qquad \text{and} \qquad \mathbf{U} \in \text{Rep}(\mathbf{n}),$$

*such that:*

1. *The matrices* $R_1, \ldots, R_{L-1}$ *are upper triangular.*

2. *The following equality holds:* $\mathbf{W} = \mathbf{Q} \cdot \mathbf{R} + \mathbf{U}$

3. *The networks defined by* $\mathbf{W}$ *and* $\mathbf{R}$ *have identical neural functions:* $f_{(\mathbf{W}, \mathbf{h})} = f_{(\mathbf{R}, \mathbf{h})}$.

---

[2]We assume each $h_i$ is piece-wise differentiable and the limit $\lim_{r \to 0} \frac{h_i(r)}{r}$ exists.

[3]For neural networks without bias vectors, one defines the reduced dimension vector $\overline{\mathbf{n}}^\text{red}$ using instead the recursion $\overline{n}_i^\text{red} = \min(n_i, \overline{n}_{i-1}^\text{red})$. See Appendix H.1.

In the the second point above, we identify $\mathbf{R}$ with its image in $\mathrm{Rep}(\mathbf{n})$ under $\iota$, and invoke the action of $O(\mathbf{n})$ on $\mathrm{Rep}(\mathbf{n})$. We say that $\mathbf{R}$ is the *reduced representation* corresponding to $\mathbf{W}$. The proof of Theorem 4.3 (appearing in Appendix E.2) relies on Algorithm 1 – a constructive algorithm proceeding layer by layer and computing a QR decomposition at each stage.

**Remark 4.4.** Part (2) of Theorem 4.3 is a quiver representation analogue of the usual QR decomposition (c.f. Section B.2); indeed, each matrix (except possibly the last) in $\mathbf{R}$ is upper triangular and $\mathbf{Q}$ is a tuple of orthogonal matrices.

---

**Algorithm 1:** QR Dimensional Reduction (`QRDimRed`)

---

**input** : $\mathbf{W} \in \mathrm{Rep}(\mathbf{n})$
**output** : $\mathbf{Q}, \mathbf{R}, \mathbf{U}$ from Theorem 4.3

$\mathbf{Q}, \mathbf{R}, \mathbf{U} \leftarrow [\,], [\,], [0]$           `// initialize output matrix lists`
$V_1 \leftarrow W_1$           `// next layer transformed weights`
**for** $i \leftarrow 1$ **to** $L-1$ **do**           `// iterate through layers`
    **if** $n_i^{\mathrm{red}} < n_i$ **then**
        $Q_i, R_i \leftarrow$ QR-decomp$(V_i,\ mode\ =\ complete)$     `// `$V_i = Q_i \circ \mathrm{inc}_i \circ R_i$
    **else**
        $Q_i, R_i \leftarrow$ QR-decomp$(V_i)$     `// `$V_i = Q_i \circ R_i$
    **end**
    Append $Q_i$ to $\mathbf{Q}$
    Append $R_i$ to $\mathbf{R}$         `// reduced weights for layer `$i$
    Append $W_{i+1} \circ \begin{bmatrix} 1 & 0 \\ 0 & Q_i \end{bmatrix} \circ p_i \circ \begin{bmatrix} 1 & 0 \\ 0 & Q_i^{-1} \end{bmatrix}$ to $\mathbf{U}$     `// matrix multiplication`
    Set $V_{i+1} \leftarrow W_{i+1} \circ \begin{bmatrix} 1 & 0 \\ 0 & Q_i \end{bmatrix} \circ \widetilde{\mathrm{inc}}_i$     `// transform next layer`
**end**
Append $V_L$ to $\mathbf{R}$

**return** $\mathbf{Q}, \mathbf{R}$, and $\mathbf{U}$

---

**Notation**: The symbol '$\circ$' denotes matrix multiplication. We initialize $\mathbf{U}$ as a list with a single element: the zero $n_1 \times (1 + n_0)$ matrix. For $i = 1, \ldots, L-1$, we have the standard inclusions $\mathrm{inc}_i = \mathrm{inc}_{n_i^{\mathrm{red}}, n_i} : \mathbb{R}^{n_i^{\mathrm{red}}} \hookrightarrow \mathbb{R}^{n_i}$ and $\widetilde{\mathrm{inc}}_i = \mathrm{inc}_{1+n_i^{\mathrm{red}}, 1+n_i} : \mathbb{R}^{1+n_i^{\mathrm{red}}} \hookrightarrow \mathbb{R}^{1+n_i}$ into the first $n_i^{\mathrm{red}}$ and $1 + n_i^{\mathrm{red}}$ coordinates, respectively. As matrices, they have ones along the main diagonal and zeros elsewhere. The map $p_i : \mathbb{R}^{n_i+1} \to \mathbb{R}^{n_i+1}$ zeros the first $1 + n_i^{\mathrm{red}}$ coordinates and leaves the remaining coordinates unchanged. As a matrix, $p_i$ is block diagonal with two square blocks; the top block is the zero matrix of size $n_i^{\mathrm{red}} + 1$, and the bottom is the identity matrix of size $n_i - n_i^{\mathrm{red}}$.

By definition, either $n_i^{\mathrm{red}} = n_{i-1}^{\mathrm{red}} + 1$ or $n_i^{\mathrm{red}} = n_i$. In the former case, $n_{i-1}^{\mathrm{red}} + 1 \le n_i$ and QR-decomp with 'mode = complete' computes the QR decomposition of the $n_i \times (1 + n_{i-1}^{\mathrm{red}})$ matrix $V_i$ as $Q_i \circ \mathrm{inc}_i \circ R_i$ where $Q_i \in O(n_i)$ and $R_i$ is upper-triangular of size $n_i^{\mathrm{red}} \times n_i^{\mathrm{red}}$. In the latter case, $n_{i-1}^{\mathrm{red}} + 1 \ge n_i$ and QR-decomp computes the QR decomposition of the $n_i \times (1 + n_{i-1}^{\mathrm{red}})$ matrix $V_i$ as $Q_i \circ R_i$ where $Q_i \in O(n_i)$ and $R_i$ is upper-triangular of size $n_i \times (1 + n_{i-1}^{\mathrm{red}})$.

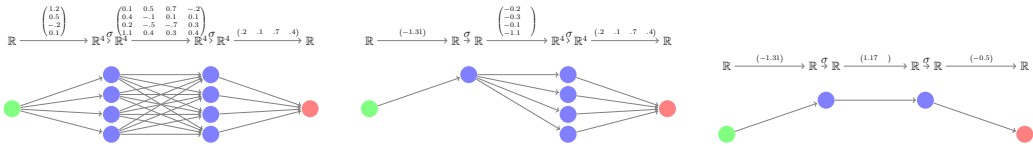

Figure 3: Parameter reduction in 3 steps. Since the activation function is radial, it commutes with orthogonal transformations. This example has no bias and thus has $\overline{\mathbf{n}}^{\mathrm{red}} = (1, 1, 1, 1)$ instead of $(1, 2, 3, 1)$. (See Footnote 3.) The number of trainable parameters reduces from 24 to 3.

**Example 4.5.** Suppose the dimension vector is $\mathbf{n} = (1, 8, 16, 8, 1)$, so the original radial neural network model has $\sum_{i=1}^{4}(n_{i-1} + 1)n_i = 305$ trainable parameters. The reduced dimension vector

is $\mathbf{n}^{\mathrm{red}} = (1, 2, 3, 4, 1)$. One computes that the compressed model has $\sum_{i=1}^{4}(n_{i-1}^{\mathrm{red}} + 1)(n_i^{\mathrm{red}}) = 34$ trainable parameters. Another example appears in Figure 3.

## 4.3 PROJECTED GRADIENT DESCENT

In this section, we use the QR decomposition of Theorem 4.3 to relate projected gradient descent for a representation to (usual) gradient descent for the corresponding reduced representation. Proofs and further context appear in Appendix F.

We adopt the notation of the previous sections. In particular, we fix a tuple $\mathbf{h} = (h_1, \ldots, h_L)$; as in Construction 4.2, this can be used to define a neural network for any representation of $\mathcal{Q}_L$, with any dimension vector. Furthermore, we fix a batch of training data in $\mathbb{R}^{n_0} \times \mathbb{R}^{n_L}$. Using $\mathbf{h}$, we obtain loss functions $\mathcal{L} : \mathsf{Rep}(\mathbf{n}) \to \mathbb{R}$ and $\mathcal{L}_{\mathrm{red}} : \mathsf{Rep}(\mathbf{n}^{\mathrm{red}}) \to \mathbb{R}$ (see Section 3.4). For a fixed learning rate, there are resulting one-step gradient descent maps given by:

$$\gamma : \mathsf{Rep}(\mathbf{n}) \to \mathsf{Rep}(\mathbf{n}) \qquad\qquad \gamma_{\mathrm{red}} : \mathsf{Rep}(\mathbf{n}^{\mathrm{red}}) \to \mathsf{Rep}(\mathbf{n}^{\mathrm{red}})$$
$$\mathbf{W} \mapsto \mathbf{W} - \eta \cdot \nabla_{\mathbf{W}}\mathcal{L} \qquad\qquad \mathbf{X} \mapsto \mathbf{X} - \eta \cdot \nabla_{\mathbf{X}}\mathcal{L}_{\mathrm{red}}$$

**Definition 4.6.** Define $\mathsf{Pr} : \mathsf{Rep}(\mathbf{n}) \to \mathsf{Rep}(\mathbf{n})$ to be the map that, given a representation $\mathbf{T} \in \mathsf{Rep}(\mathbf{n})$, zeros the bottom left $(n_i - n_i^{\mathrm{red}}) \times (1 + n_{i-1}^{\mathrm{red}})$ submatrix of $T_i$, for each $i$. Schematically,

$$\mathsf{Pr} : \quad \mathbf{T} = \left( \begin{bmatrix} * & * \\ * & * \end{bmatrix} \right)_{i=1}^{L} \quad \longmapsto \quad \mathsf{Pr}(\mathbf{T}) = \left( \begin{bmatrix} * & * \\ 0 & * \end{bmatrix} \right)_{i=1}^{L}.$$

The one-step projected gradient descent map on $\mathsf{Rep}(\mathbf{n})$ at learning rate $\eta > 0$ is defined as:

$$\gamma_{\mathrm{proj}} : \mathsf{Rep}(\mathbf{n}) \to \mathsf{Rep}(\mathbf{n}) \qquad\qquad \mathbf{W} \mapsto \mathbf{W} - \eta \cdot \mathsf{Pr}(\nabla_{\mathbf{W}}\mathcal{L})$$

Hence, while all entries of each matrix $W_i$ in the representation $\mathbf{W}$ contribute to the computation of the gradient $\nabla_{\mathbf{W}}\mathcal{L}$, only those not in the bottom left submatrix get updated under the projected gradient descent map $\gamma_{\mathrm{proj}}$. To ease notation in the statement of the following result, we identify elements of $\mathsf{Rep}(\mathbf{n}^{\mathrm{red}})$ with their images in $\mathsf{Rep}(\mathbf{n})$ under $\iota$.

**Theorem 4.7.** *Let $\mathbf{W} \in \mathsf{Rep}(\mathbf{n})$ and let $\mathbf{Q}$, $\mathbf{R}$, and $\mathbf{U}$ be the outputs of Algorithm 1. Fix a learning rate $\eta > 0$. For any $k \geq 0$, we have:*

$$\gamma^k(\mathbf{W}) = \mathbf{Q} \cdot \gamma^k(\mathbf{Q}^{-1} \cdot \mathbf{W}) \qquad\qquad and \qquad\qquad \gamma_{\mathrm{proj}}^k(\mathbf{Q}^{-1} \cdot \mathbf{W}) = \gamma_{\mathrm{red}}^k(\mathbf{R}) + \mathbf{Q}^{-1} \cdot \mathbf{U}.$$

While the proof of the second statement of Theorem 4.7 is quite technical, the proof of the first relies on a basic interaction of orthogonal transformations with gradient descent, which we briefly illustrate (Appendices C and F contain more details). Let $\mathcal{L} : \mathbb{R}^p \to \mathbb{R}$ be a function, and let $\gamma : \mathbb{R}^p \to \mathbb{R}^p$ be the gradient descent map with respect to $\mathcal{L}$. Suppose $Q \in O(p)$ is an orthogonal transformation of $\mathbb{R}^p$ that preserves the level sets of $\mathcal{L}$, that is, $\mathcal{L}(Q(v)) = \mathcal{L}(v)$ for all $v \in \mathbb{R}^p$. Then $Q$ commutes with gradient descent, i.e. $\gamma(Q(v)) = Q(\gamma(v))$ for all $v \in \mathbb{R}^p$. See Figure 4.

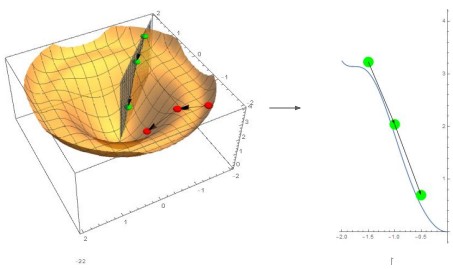

Figure 4: If the loss is invariant with respect to an orthogonal transformation $Q$ of the parameter space, then optimization of the network by gradient descent is also invariant with respect to $Q$. In this example, projected and usual gradient descent match; this is not the case in higher dimensions, as explained through examples in the appendix (see F.4 and H.1).

## 5 EXPERIMENTS

In addition to the theoretical results in this work, we provide an implementation of `QRDimRed` as described in Algorithm 1. We (1) empirically validate that our implementation satisfies the claims of Theorems 4.3 and Theorem 4.7 and (2) quantify real-world performance.

**Empirical verification of Theorem 4.3** We verify the Theorem using a small model and synthetic data. We learn the function $f(x) = e^{-x^2}$ using $N = 121$ samples $x_j = -3 + j/20$ for $0 \leq j < 121$. We model $f_\mathbf{W}$ as a radial neural network with layer widths $\mathbf{n} = (1, 6, 7, 1)$ and activation the radial shifted sigmoid $h(x) = 1/(1 + e^{-x+b})$. Applying `QRDimRed` gives a radial neural network $f_\mathbf{R}$ with widths $\mathbf{n}^{\text{red}} = (1, 2, 3, 1)$. Theorem 4.3 implies that the neural functions of $f_\mathbf{W}$ and $f_\mathbf{R}$ are equal. Over 10 random initializations of $\mathbf{W}$, the mean absolute error $(1/N) \sum_j |f_\mathbf{W}(x_j) - f_\mathbf{R}(x_j)| = 1.31 \cdot 10^{-8} \pm 4.45 \cdot 10^{-9}$. Thus $f_\mathbf{W}$ and $f_\mathbf{R}$ agree up to machine precision.

**Empirical verification of Theorem 4.7** Similarly, we verify the conclusions of Theorem 4.7 using synthetic data. The claim is that training $f_{\mathbf{Q}^{-1} \cdot \mathbf{W}}$ with objective $\mathcal{L}$ by projected gradient descent coincides with training $f_\mathbf{R}$ with objective $\mathcal{L}_{\text{red}}$ by usual gradient descent. We verified this for 3000 epochs at learning rate 0.01. Over 10 random initializations of $\mathbf{W}$, the loss functions match up to machine precision with $|\mathcal{L} - \mathcal{L}_{\text{red}}| = 4.02 \cdot 10^{-9} \pm 7.01 \cdot 10^{-9}$.

**Reduced model trains faster.** Due to the relation between projected gradient descent of the full network $\mathbf{W}$ and gradient descent of the reduced network $\mathbf{R}$, our method may be applied before training to produce a smaller model class which *trains* faster without sacrificing accuracy. We test this hypothesis in learning the function $f : \mathbb{R}^2 \to \mathbb{R}^2$ sending $x = (t_1, t_2)$ to $(e^{-t_1^2}, e^{-t_2^2})$ using $N = 121^2$ samples $(-3 + j/20, -3 + k/20)$ for $0 \leq j, k < 121$. We model $f_\mathbf{W}$ as a radial neural network with layer widths $\mathbf{n} = (2, 16, 64, 128, 16, 2)$ and activation the radial sigmoid $h(r) = 1/(1 + e^{-r})$. Applying `QRDimRed` gives a radial neural network $f_\mathbf{R}$ with widths $\mathbf{n}^{\text{red}} = (2, 3, 4, 5, 6, 2)$. We trained both models until the training loss was $\leq 0.01$. Running on a system with an Intel i5-8257U@1.40GHz and 8GB of RAM and averaged over 10 random initializations, the reduced network trained in $15.32 \pm 2.53$ seconds and the original network trained in $31.24 \pm 4.55$ seconds.

## 6 CONCLUSIONS AND FUTURE WORK

In this paper, we have adopted the formalism of quiver representation theory in order to establish a theoretical framework for neural networks. While drawing inspiration from previous work, our approach is novel in that it (1) reveals a large group of symmetries of neural network parameter spaces, (2) leads to a version of the QR decomposition in the context of radial neural networks, and (3) precipitates an algorithm to reduce the widths of the hidden layers in such networks.

Our main results, namely Theorems 4.3 and 4.7, may potentially generalize in several ways. First, these theorems are only meaningful if $n_i > n_{i-1} + 1$ for some $i$ (otherwise $\mathbf{n}^{\text{red}} = \mathbf{n}$). Many networks such as decoders, super-resolution mappings, and GANs fulfill this criteria, but others do not. Our techniques may achieve meaningful width reductions in other cases. Second, the hypothesis that the activation functions are radial is crucial for our results, as they commute with orthogonal matrices, and hence interact favorably with QR decompositions. However, similar dimensional-reduction procedures may well be possible for other activation functions which relate to other matrix decompositions. In particular, the bottleneck effect discovered by Serra et al. (2018) for ReLU networks bears a striking similarity to our ascending condition. Third, though Theorem 4.7 does not hold for coordinate-dependent optimizers like Adam, other optimizers in the same spirit may be compatible. Fourth, there may be enhancements of our results that incorporate regularization and assert provable improvements to generalization. For example, the loss function of a radial neural network with dimension vector $\mathbf{n}$ remains invariant for the $O(\mathbf{n})$ action after adding an $L^2$ regularization term. Finally, the conceptual flexibility of quiver representation theory can encapsulate neural networks beyond MLPs, including CNNs, equivariant neural networks, RNNs, GNNs, and others.

Our result in Theorem 4.7 relates projected gradient descent $\gamma_{\text{proj}}$ for the original network to gradient descent $\gamma$ for the reduced network. In order to make theoretical conclusions about optimization of the original versus the reduced network, it is necessary to relate gradient descent on both spaces. To realize this, we conjecture, based on preliminary experimental evidence, that our version of projected gradient descent is a first-order approximation of usual gradient descent; we are pursuing work to make this relationship precise.

ETHICS STATEMENT

Our work is primarily focused on theoretical foundations of machine learning, however, it does have a direct application in the form a model compression. Model compression is largely beneficial to the world since it allows for inference to run on smaller systems which use less energy. On the other hand, when models may be run on smaller systems such as smartphones, it is easier to use deep models covertly, for example, for facial recognition and surveillance. This may make abuses of deep learning technology easier to hide.

REPRODUCIBILITY STATEMENT

The theoretical results of this paper, namely Theorem 4.3 and Theorem 4.7, may be independently verified through their proofs, which we include in their entirety in Appendices E and F, including all necessary definitions, lemmas, and hypotheses in precise and complete mathematical language. The empirical verification of Section 5 may be reproduced using the code included with the supplementary materials. In addition, Algorithm 1 is written in detailed pseudocode, allowing readers to recreate our algorithm in a programming language of their choosing.

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

## A    Organization of appendices

This paper is a contribution to the mathematical foundations of machine learning, and our results are motivated by expanding the applicability and performance of neural networks. At the same time, we use precise mathematical formalism throughout, and emphasize the rich abstract structures underlying neural networks. The purposes of these appendices are several:

1. To clarify the mathematical conventions and terminology, thus making the paper more accessible.

2. To provide full proofs of the main results.

3. To develop context around various constructions appearing in the main text.

4. To discuss in detail examples, special cases, and generalizations of our results.

We now give a summary of the contents of the appendices. Appendix B consists of preliminary material. We first review the necessary background on linear algebra and group theory, and then state the version of the QR decomposition that is relevant for this paper.

Appendix C serves to establish notation used in the context of gradient descent, and to prove an interaction between gradient descent and orthogonal transformations (Proposition C.2).

Appendix D supplements 3. Specifically, in Appendix D, we provide expanded foundations for our proposed framework of neural networks as quiver representations. In particular, we give a proof of Lemma 3.1, explain symmetries on the space of all neural networks with a specified dimension vector, and prove equivariance results. Finally, we explain how the "non-negative homogeneity" property of pointwise ReLU activation functions is a special case of our results (c.f Proposition D.5 and Remark D.7).

In Appendix E, we first establish elementary results related to radial functions and radial neural networks in Section E.1, and then give a proof of the QR Decomposition result (Theorem 4.3) in Section E.2. The proof relies on Algorithm 1.

In Appendix F, we focus on projected gradient descent. Specifically, we first introduce an 'interpolating space' that relates the space of representations with a given dimension vector to the space of representations with the reduced dimension vector (Section F.1). Using the interpolating space, we define a projected version of gradient descent (Section F.2) and give a proof of Theorem 4.7, which relates projected gradient descent and the QR decomposition for radial neural networks.

We devote Appendices H and I to special cases and generalizations of the constructions and results of the main text. Specifically, we consider a version of our results for neural networks with no biases (Section H.1), a generalization involving shifts in radial functions (Section H.2), and a formulation and generalization of our results using the language of category theory (Appendix I).

## B    Linear algebra

In this appendix, we first review basic notation and concepts from linear algebra, group theory, and representation theory. We then state a version of the QR decomposition. References include Petersen et al. (2012), Quarteroni et al. (2007), and Dummit & Foote (2003).

### B.1    Basics

For positive integers $m$ and $n$, let $\mathrm{Hom}(\mathbb{R}^m, \mathbb{R}^n)$ or $\mathbb{R}^{m \times n}$ denote the vector space of $n$ by $m$ matrices with entries in $\mathbb{R}$, that is, the vector space of linear functions (also known as **homomorphisms**) from $\mathbb{R}^m$ to $\mathbb{R}^n$. The **general linear group** $\mathrm{GL}_n(\mathbb{R})$ consists of the set of invertible $n$ by $n$ matrices, with the operation of matrix multiplication. Equivalently, $\mathrm{GL}_n(\mathbb{R})$ consists of all linear automorphisms of $\mathbb{R}^n$, with the operation of composition. Such automorphisms are given precisely by change-of-basis transformations. The unit is the identity $n$ by $n$ matrix, denoted $\mathrm{id}_n$. The **orthogonal group** $O(n)$ is the subgroup of $\mathrm{GL}_n(\mathbb{R})$ consisting of all matrices $Q$ such that $Q^T Q = \mathrm{id}_n$. Such a matrix is called an **orthogonal transformation**.

Let $G$ be a group. An **action** (or **representation**) of $G$ on the vector space $\mathbb{R}^n$ is the data of an invertible $n$ by $n$ matrix $\rho(g)$ for every group element $g \in G$, such that (1) for all $g, h \in G$, the matrix $\rho(gh)$ is the product of the matrices $\rho(g)$ and $\rho(h)$, and (2) for the identity element $1_G \in G$, we have $\rho(1_G) = \mathrm{id}_n$. In other words, an action amounts to a group homomorphism $\rho : G \to \mathrm{GL}_n(\mathbb{R})$. We often abbreviate $\rho(g)(v)$ as simply $g \cdot v$, for $g \in G$ and $v \in \mathbb{R}^n$. A function $f : \mathbb{R}^n \to \mathbb{R}$ (non-linear in general) is **invariant** for the action of $G$ if $f(g \cdot v) = f(v)$ for all $g \in G$ and $v \in \mathbb{R}^n$.

Suppose $m \leq n$. Then $\mathrm{inc}_{m,n} : \mathbb{R}^m \hookrightarrow \mathbb{R}^n$ denotes the **standard inclusion** into the first $m$ coordinates:

$$(v_1, \ldots, v_m) \mapsto (v_1, \ldots, v_m, 0, \ldots, 0)$$

The **standard projection** $\pi_{n,m} : \mathbb{R}^n \twoheadrightarrow \mathbb{R}^m$ onto the first $m$ coordinates is defined as:

$$(v_1, \ldots, v_m, v_{m+1}, \ldots, v_n) \mapsto (v_1, \ldots, v_m)$$

An **affine map** from $\mathbb{R}^m$ to $\mathbb{R}^n$ is one of the form

$$x \mapsto Ax + b$$

for some $n \times m$ matrix $A$ and $n$-dimensional vector $b$. The set of affine maps from $\mathbb{R}^m$ to $\mathbb{R}^n$ can be identified with the vector space $\mathrm{Hom}(\mathbb{R}^{m+1}, \mathbb{R}^n)$, as we now explain. First, given $A$ and $b$ as above, we form the $n \times (m+1)$ matrix $[b \; A]$. Conversely, the affine map corresponding to $W \in \mathrm{Hom}(\mathbb{R}^{m+1}, \mathbb{R}^n)$ is given by

$$\widetilde{W} = W \circ \mathrm{ext}_m : \mathbb{R}^m \to \mathbb{R}^n; \qquad x \mapsto W\left(\mathrm{ext}_m(x)\right),$$

where $\mathrm{ext}_m$ is defined as follows:

**Definition B.1.** For $m \geq 1$, let

$$\mathrm{ext}_m : \mathbb{R}^m \to \mathbb{R}^{m+1}$$

be the map taking $v = (v_1, v_2, \ldots, v_m) \in \mathbb{R}^m$ to $(1, v_1, v_2, \ldots, v_m) \in \mathbb{R}^{m+1}$.

## B.2 THE QR DECOMPOSITION

In this section, we recall the QR decomposition and note several relevant facts. For integers $n$ and $m$, let $\mathrm{Hom}^{\mathrm{upper}}(\mathbb{R}^m, \mathbb{R}^n)$ denote the vector space of upper triangular $n$ by $m$ matrices.

**Theorem B.2** (QR Decomposition). *The following map is surjective:*

$$O(n) \times \mathrm{Hom}^{\mathrm{upper}}(\mathbb{R}^m, \mathbb{R}^n) \longrightarrow \mathrm{Hom}(\mathbb{R}^m, \mathbb{R}^n)$$
$$Q \,, \; R \quad \mapsto \quad Q \circ R$$

In other words, any matrix can be written as the product of an orthogonal matrix and an upper-triangular matrix. When $m \leq n$, the last $n - m$ rows of any matrix in $\mathrm{Hom}^{\mathrm{upper}}(\mathbb{R}^m, \mathbb{R}^n)$ are zero, and the top $m$ rows form an upper-triangular $m$ by $m$ matrix. These observations lead to the following special case of the QR decomposition:

**Corollary B.3.** *Suppose $m \leq n$. The following map is surjective:*

$$\mu : O(n) \times \mathrm{Hom}^{\mathrm{upper}}(\mathbb{R}^m, \mathbb{R}^m) \longrightarrow \mathrm{Hom}(\mathbb{R}^m, \mathbb{R}^n)$$
$$Q \,, \; R \quad \mapsto \quad Q \circ \mathrm{inc}_{m,n} \circ R$$

We make some remarks:

1. There are several algorithms for computing the QR decomposition of a given matrix. One is Gram–Schmidt orthogonalization, and another is the method of Householder reflections. The latter has computational complexity $O(n^2 m)$ in the case of a $n \times m$ matrix with $n \geq m$. The package `numpy` includes a function `numpy.linalg.qr` that computes the QR decomposition of a matrix using Householder reflections.

2. The QR decomposition is not unique in general, or, in other words, the map $\mu$ is not injective in general. For example, if $n > m$, each fiber of $\mu$ contains a copy of $O(n - m)$.

3. The QR decomposition is unique (in a certain sense) for invertible square matrices. To be precise, let $B_n^+$ be the subset of of $\mathrm{Hom}^{\mathrm{upper}}(\mathbb{R}^n, \mathbb{R}^n)$ consisting of upper triangular matrices with positive entries along the diagonal. If $n = m$, then both $B_n^+$ and $O(n)$ are subgroups of $\mathrm{GL}_n(\mathbb{R})$, and the multiplication map $O(n) \times B_n^+ \to \mathrm{GL}_n(\mathbb{R})$ is bijective. However, the QR decomposition is not unique for non-invertible square matrices.

## C  SYMMETRY AND GRADIENT DESCENT

In this appendix, we establish a gradient descent formalism that appears throughout the paper. Additionally, we explain the interaction between gradient descent and orthogonal transformations, which is a key aspect of our main results.

Fix a positive integer $p$. Let $V = \mathbb{R}^p$ and $\mathcal{L} : V \to \mathbb{R}$ be a differentiable function[4]. Semantically, $p$ is the number of independent parameters, $V$ is the parameter space, and $\mathcal{L}$ is the loss function associated to a batch of training data. We write elements of $V$ as tuples, e.g. $v = (v_1, \ldots, v_p)$, or as column vectors. We write elements of the dual vector space $V^*$ as row vectors. The **differential** and **gradient** of $\mathcal{L}$ are defined as:

$$d\mathcal{L} : V \longrightarrow \mathrm{Hom}(V, \mathbb{R}) = V^* \qquad\qquad \nabla\mathcal{L} : V \longrightarrow V$$

$$v \mapsto d_v\mathcal{L} = \left[ \left.\frac{\partial \mathcal{L}}{\partial x_1}\right|_v \quad \cdots \quad \left.\frac{\partial \mathcal{L}}{\partial x_p}\right|_v \right] \qquad\qquad v \mapsto \nabla_v\mathcal{L} = \left( \left.\frac{\partial \mathcal{L}}{\partial x_1}\right|_v, \ldots, \left.\frac{\partial \mathcal{L}}{\partial x_p}\right|_v \right)$$

**Remark C.1.** Suppose that $\mathcal{L}$ is in fact a linear map. Then $\mathcal{L}$ belongs to $V^*$ and $d\mathcal{L}$ is the constant map taking each $v \in V$ to $\mathcal{L}$. That is, $d_v\mathcal{L} = \mathcal{L}$ as maps $V \to \mathbb{R}$.

A **step of gradient descent** with respect to $\mathcal{L}$ at learning rate $\eta > 0$ is the function

$$\gamma = \gamma_\eta : V \longrightarrow V$$
$$v \mapsto v - \eta \cdot \nabla_v\mathcal{L}$$

Hence, gradient descent updates the $i$-th coordinate $v_i$ of $v$ to $v_i - \eta \cdot \frac{\partial \mathcal{L}}{\partial x_i}|_v$, for $i = 1, \ldots, p$. When $\eta$ is clear from context, we abbreviate $\gamma_\eta$ by $\gamma$. For any $k \geq 0$, the $k$-fold composition of $\gamma$ is $\gamma^k := \gamma \circ \gamma \circ \cdots \circ \gamma$, with $\gamma^0$ being the identity map on $V$.

Throughout the appendices, all proofs involving gradient descent reduce to the $\eta = 1$ case. The arguments for general $\eta > 0$ and can be obtained either by inserting the scalar $\eta$ where necessary, or by rescaling the function $\mathcal{L}$.

**Proposition C.2.** *Let $Q$ be an orthogonal transformation of $V = \mathbb{R}^p$ such that $\mathcal{L} \circ Q = \mathcal{L}$. Then, for any $v \in V$ and $k \geq 0$, we have:*

$$Q\left(\nabla_v\mathcal{L}\right) = \nabla_{Q(v)}\mathcal{L} \qquad \text{and} \qquad \gamma^k(Q(v)) = Q(\gamma^k(v)).$$

*Proof.* We give a proof in the case $\eta = 1$. The general case follows similarly. The hypothesis $\mathcal{L} \circ Q = \mathcal{L}$ implies that:

$$\nabla_v\mathcal{L} = (d_v\mathcal{L})^T = (d_v(\mathcal{L} \circ Q))^T = (d_{Q(v)}\mathcal{L} \circ d_vQ)^T = (d_{Q(v)}\mathcal{L} \circ Q)^T = Q^T(d_{Q(v)}\mathcal{L})^T$$
$$= Q^T(\nabla_{Q(v)}\mathcal{L})$$

where $d_{Q(v)}Q = Q$ since $Q$ is a linear map. The fact that $Q$ is an orthogonal transformation implies that $Q^{-1} = Q^T$. The first claim follows. For the second claim, we first compute:

$$\gamma(Q(v)) = Q(v) - \nabla_{Q(v)}\mathcal{L} = Q(v) - Q(\nabla_v\mathcal{L}) = Q(v - \nabla_v\mathcal{L}) = Q(\gamma(v))$$

Inductive reasoning shows that $\gamma^k(Q(v)) = Q(\gamma^k(v))$ for all $k \geq 0$. $\qquad\qquad\square$

The following two results are not necessary for the results of the current paper, but may be of interest to readers. We omit the proofs (which are straightforward).

---

[4]This discussion extends easily to the case where $\mathcal{L}$ is piece-wise differentiable.

**Proposition C.3.** *Suppose that a group $G$ acts on $V$ by orthogonal transformations and that $\mathcal{L}$ is $G$-invariant. Then there is a well-defined gradient descent map on the set of $G$-orbits on $V$:*

$$\widetilde{\gamma} : V/G \to V/G.$$

**Proposition C.4.** *Let $M$ be a linear endomorphism of $V = \mathbb{R}^p$ such that $\mathcal{L} \circ M = \mathcal{L}$.*

1. *For any $v \in V$, we have: $\nabla_v \mathcal{L} = M^T(\nabla_{M(v)} \mathcal{L})$.*

2. *If $M = P$ is an orthogonal projection, i.e. $P^2 = P = P^T$, then, for any $v \in V$ and $k \geq 0$, we have:*
$$\gamma^k(v) = v - P(v) + \gamma^k(P(v)).$$

## D  THE NEURAL QUIVER

In this appendix, we expand on Section 3 of the main text. We explain in more detail the relation between representations of the neural quiver and affine maps, describe an action of $\mathrm{GL}_{\mathbf{n}}^{\mathrm{hidden}}$ on the space of neural networks $\mathsf{Neur}(\mathcal{Q}_L, \mathbf{n})$, and prove equivariance results related to this action. References for general quiver representation theory include (Nakajima et al., 1998) and (Kirillov Jr, 2016).

### D.1  THE NEURAL QUIVER

We begin by recalling the definition of the neural quiver. For a positive integer $L$, the neural quiver $\mathcal{Q}_L$ is the following quiver with $L + 2$ vertices:

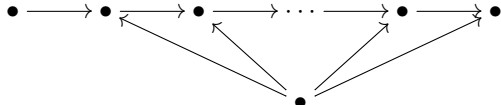

The vertices in the top row are indexed from $i = 0$ to $i = L$. The first vertex ($i = 0$) in the top row is the sole input vertex, the last vertex ($i = L$) is the output vertex, and every other vertex ($1 \leq i \leq L - 1$) in the top row is a hidden vertex. The vertex at the bottom is the sole bias vertex and is connected to each vertex in the top row except for $i = 0$. We only consider dimension vectors whose value at the bias vertex is equal to 1. Hence, a dimension vector for $\mathcal{Q}_L$ will refer to an $(L + 1)$-tuple of positive integers $\mathbf{n} = (n_0, n_1, \ldots, n_L)$. We now restate and give a full proof of Lemma 3.1:

**Lemma D.1.** *(Lemma 3.1.) A representation of $\mathcal{Q}_L$ with dimension vector $\mathbf{n}$ is equivalent to the data of an affine map $\mathbb{R}^{n_{i-1}} \to \mathbb{R}^{n_i}$ for each $i = 1, \ldots, L$. There is an isomorphism of vector spaces:*

$$\mathsf{Rep}(\mathcal{Q}_L, \mathbf{n}) \simeq \bigoplus_{i=1}^{L} \mathrm{Hom}(\mathbb{R}^{1+n_{i-1}}, \mathbb{R}^{n_i}).$$

*Proof.* By definition, a representation of $\mathcal{Q}_L$ with dimension vector $\mathbf{n}$ consists of the data of a matrix $A_i \in \mathrm{Hom}(\mathbb{R}^{n_{i-1}}, \mathbb{R}^{n_i})$ and a vector $b_i \in \mathbb{R}^{n_i}$, for each $i = 1, \ldots, L$. For each $i$, the corresponding affine map $\widetilde{W}_i : \mathbb{R}^{n_{i-1}} \to \mathbb{R}^{n_i}$ is given by $x \mapsto A_i x + b_i$. Conversely, given the data of an affine map $\mathbb{R}^{n_{i-1}} \to \mathbb{R}^{n_i}$, for each $i$, the linear part defines an element of $\mathrm{Hom}(\mathbb{R}^{n_{i-1}}, \mathbb{R}^{n_i})$ and the translation part defines an element of $\mathbb{R}^{n_i}$. This proves the first claim.

The second claim follows from the discussion of affine maps appearing in Section B. Explicitly, given an affine map $x \mapsto A_i x + b_i$ from $\mathbb{R}^{n_{i-1}}$ to $\mathbb{R}^{n_i}$ for each $i = 1, \ldots, L$, one defines an element of $\bigoplus_{i=1}^{L} \mathrm{Hom}(\mathbb{R}^{1+n_{i-1}}, \mathbb{R}^{n_i})$ whose $i$-th factor is the matrix $W_i = [b_i \ A_i]$. Conversely, pre-composing a matrix in $\mathrm{Hom}(\mathbb{R}^{1+n_{i-1}}, \mathbb{R}^{n_i})$ with the map $\mathrm{ext}_{n_{i-1}} : \mathbb{R}^{n_{i-1}} \to \mathbb{R}^{1+n_{i-1}}$ gives an affine map $\widetilde{W}_i$ from $\mathbb{R}^{n_{i-1}}$ to $\mathbb{R}^{n_i}$. $\square$

**Notation**: In light of the preceding lemma, we henceforth denote a representation of $\mathcal{Q}_L$ as a tuple $\mathbf{W} = (W_i)_{i=1}^{L}$, where each $W_i$ belongs to $\mathrm{Hom}(\mathbb{R}^{1+n_{i-1}}, \mathbb{R}^{n_i})$.

Recall that the parameter symmetry group $\mathrm{GL}_{\mathbf{n}}^{\mathrm{hidden}}$ is defined as the product of the groups $\mathrm{GL}_{n_i}(\mathbb{R})$ as $i$ runs over the hidden vertices of the neural quiver $\mathcal{Q}_L$:

$$\mathrm{GL}_{\mathbf{n}}^{\mathrm{hidden}} := \prod_{i=1}^{L-1} \mathrm{GL}_{n_i}(\mathbb{R}).$$

Hence an element of $\mathrm{GL}_{\mathbf{n}}^{\mathrm{hidden}}$ consists of the choice of a transformation of $\mathbb{R}^{n_i}$ for each hidden vertex $i$. There is an action of $\mathrm{GL}_{\mathbf{n}}^{\mathrm{hidden}}$ on $\mathsf{Rep}(\mathcal{Q}_L, \mathbf{n})$ given by[5]:

$$\mathbf{W} = (W_i)_{i=1}^{L} \quad \mapsto \quad \mathbf{g} \cdot \mathbf{W} := \left( g_i \circ W_i \circ \begin{bmatrix} 1 & 0 \\ 0 & g_{i-1}^{-1} \end{bmatrix} \right)_{i=1}^{L},$$

where $\mathbf{g} = (g_i)_{i=1}^{L-1}$ is an element of $\mathrm{GL}_{\mathbf{n}}^{\mathrm{hidden}}$, and $g_0 = \mathrm{id}_{n_0}$ and $g_L = \mathrm{id}_{n_L}$.

## D.2   Neural networks

Recall the following definition, which appears as Definition 3.2 in Section 3.4 of the main text, and is a special case of the more general Definition G.6 appearing below in Appendix G.

**Definition D.2.** Let $L$ be a positive integer. An $\mathcal{Q}_L$-**neural network** consists of the following:

1. **Hyperparameters.** A dimension vector $\mathbf{n} = (n_0, n_1, \ldots, n_L)$ for the neural quiver $\mathcal{Q}_L$.

2. **Trainable parameters.** A representation $\mathbf{W} = (W_1, \ldots, W_L)$ of the neural quiver $\mathcal{Q}_L$ with dimension vector $\mathbf{n}$. So, for $i = 1, \ldots, L$, we have a matrix $W_i \in \mathrm{Hom}(\mathbb{R}^{1+n_{i-1}}, \mathbb{R}^{n_i})$.

3. **Activation functions.** Piece-wise differentiable functions $a_i : \mathbb{R}^{n_i} \to \mathbb{R}^{n_i}$ for $i = 1, \ldots, L$. These are grouped into a tuple $\mathbf{a} = (a_1, \ldots, a_L)$.

As in Section 3.4, we denote by $\mathsf{Neur}(\mathcal{Q}_L, \mathbf{n})$ the vector space of neural networks with $L$ layers and dimension vector $\mathbf{n}$, and define the neural function of a neural network $(\mathbf{W}, \mathbf{a})$ as:

$$f_{(\mathbf{W}, \mathbf{a})} : \mathbb{R}^{n_0} \to \mathbb{R}^{n_L} \qquad x \mapsto a_L \circ \widetilde{W}_L \circ \cdots \circ a_2 \circ \widetilde{W}_2 \circ a_1 \circ \widetilde{W}_1(x)$$

where $\widetilde{W}_i : \mathbb{R}^{n_{i-1}} \to \mathbb{R}^{n_i}$ is the affine map corresponding to $W_i$.

There is an action of the group $\mathrm{GL}_{\mathbf{n}}^{\mathrm{hidden}}$ on $\mathsf{Neur}(\mathcal{Q}_L, \mathbf{n})$ given as follows. The element $\mathbf{g} = (g_i)_{i=1}^{L-1} \in \mathrm{GL}_{\mathbf{n}}^{\mathrm{hidden}}$ transforms $(\mathbf{W}, \mathbf{a}) \in \mathsf{Neur}(\mathcal{Q}_L, \mathbf{n})$ as:

$$(\mathbf{W}, \mathbf{a}) \quad \mapsto \quad \mathbf{g} \cdot (\mathbf{W}, \mathbf{a}) := \left( \left( g_i \circ W_i \circ \begin{bmatrix} 1 & 0 \\ 0 & g_{i-1}^{-1} \end{bmatrix} \right)_{i=1}^{L}, \quad \left( g_i \circ a_i \circ g_i^{-1} \right)_{i=1}^{L} \right), \quad \text{(D.1)}$$

where $g_0 = \mathrm{id}_{n_0}$ and $g_L = \mathrm{id}_{n_L}$. The following lemma shows that the neural function is unchanged by the $\mathrm{GL}_{\mathbf{n}}^{\mathrm{hidden}}$ action, that is, if two neural networks are related by the action of an element in $\mathrm{GL}_{\mathbf{n}}^{\mathrm{hidden}}$, then their neural functions are the same.

**Lemma D.3.** *For any neural network* $(\mathbf{W}, \mathbf{a})$ *in* $\mathsf{Neur}(\mathcal{Q}_L, \mathbf{n})$ *and any* $\mathbf{g}$ *in* $\mathrm{GL}_{\mathbf{n}}^{\mathrm{hidden}}$, *the neural functions of* $(\mathbf{W}, \mathbf{a})$ *and* $\mathbf{g} \cdot (\mathbf{W}, \mathbf{a})$ *coincide:*

$$f_{\mathbf{g} \cdot (\mathbf{W}, \mathbf{a})} = f_{(\mathbf{W}, \mathbf{a})}$$

*Proof.* The element $\mathbf{g} = (g_i) \in \mathrm{GL}_{\mathbf{n}}^{\mathrm{hidden}}$ transforms $\widetilde{W}_i = W_i \circ \mathrm{ext}_{n_{i-1}}$ to $g_i \circ W_i \circ \mathrm{ext}_{n_{i-1}} \circ g_{i-1}^{-1}$ and $a_i$ to $g_i \circ a_i \circ g_i^{-1}$, for $i = 1, \ldots, L$ (where $g_0 = \mathrm{id}_{n_0}$ and $g_L = \mathrm{id}_{n_L}$). Thus, $\mathbf{g}$ transforms the composition $a_i \circ \widetilde{W}_i$ to the composition $g_i \circ a_i \circ \widetilde{W}_i \circ g_{i-1}^{-1}$. The result now follows from the definition of the neural function. $\qquad \square$

---

[5]As in the main text, for each $i$, the matrix $\begin{bmatrix} 1 & 0 \\ 0 & g_{i-1}^{-1} \end{bmatrix}$ is block diagonal with one block of size $1 \times 1$ consisting of the single entry '1', and one block of size $n_{i-1} \times n_{i-1}$ consisting of the matrix $g_{i-1}^{-1}$.

We now define a certain subgroup of $\mathrm{GL}_{\mathbf{n}}^{\mathrm{hidden}}$ that will feature in later discussion.

**Definition D.4.** The **stabilizer** in $\mathrm{GL}_{\mathbf{n}}^{\mathrm{hidden}}$ of a choice of activation functions $\mathbf{a}$ is defined as:

$$Z(\mathbf{a}) = \{\mathbf{g} = (g_i) \in \mathrm{GL}_{\mathbf{n}}^{\mathrm{hidden}} \;\; : \;\; g_i \circ a_i \circ g_i^{-1} = a_i \text{ for all } i = 1, \ldots, L-1\}.$$

Note that each $a_i$ is a function from $\mathbb{R}^{n_i}$ to $\mathbb{R}^{n_i}$, as is each $g_i$, so the equality $g_i \circ a_i \circ g_i^{-1} = a_i$ is as functions from $\mathbb{R}^{n_i}$ to $\mathbb{R}^{n_i}$. As a subgroup of $\mathrm{GL}_{\mathbf{n}}^{\mathrm{hidden}}$, the stabilizer $Z(\mathbf{a})$ acts on $\mathsf{Rep}(\mathcal{Q}_L, \mathbf{n})$.

We now recall from Section 3 the formulation of the optimization problem for neural networks in the language of quiver representations. For the remainder of this section, we fix a dimension vector $\mathbf{n}$ for the the neural quiver $\mathcal{Q}_L$, and activation functions $\mathbf{a} = (a_1, \ldots, a_L)$. To a batch of training data $\mathbb{X} = \{(x_j, y_j)\} \subseteq \mathbb{R}^{n_0} \times \mathbb{R}^{n_L}$, there is an associated **loss function** on $\mathsf{Rep}(\mathcal{Q}_L, \mathbf{n})$ defined as

$$\mathcal{L} = \mathcal{L}_{\mathbb{X}} : \mathsf{Rep}(\mathcal{Q}_L, \mathbf{n}) \to \mathbb{R} \qquad \mathcal{L}(\mathbf{W}) = \sum_j \mathcal{C}(f_{(\mathbf{W}, \mathbf{a})}(x_j), y_j)$$

where $\mathcal{C} : \mathbb{R}^{n_L} \times \mathbb{R}^{n_L} \to \mathbb{R}$ is a cost function on $\mathbb{R}^{n_L}$. Following the formalism of Appendix C, we obtain a gradient descent map

$$\gamma : \mathsf{Rep}(\mathcal{Q}_L, \mathbf{n}) \to \mathsf{Rep}(\mathcal{Q}_L, \mathbf{n})$$

for any learning rate $\eta > 0$. Hence:

> $\to$ **Upshot**: the backpropogation algorithm can be regarded as taking place on the vector space $\mathsf{Rep}(\mathcal{Q}_L, \mathbf{n})$ of representations of the neural quiver $\mathcal{Q}_L$.

To conclude this section, we demonstrate that the loss function is invariant for the action of the group $Z(\mathbf{a})$.

**Proposition D.5.** *Let $\mathbf{a}$ be activation functions, and let $\mathcal{L}$ be the loss function associated to a batch of training data. Then, for all $\mathbf{g} \in Z(\mathbf{a})$ and $\mathbf{W} \in \mathsf{Rep}(\mathcal{Q}_L, \mathbf{n})$, we have:*

$$\mathcal{L}(\mathbf{g} \cdot \mathbf{W}) = \mathcal{L}(\mathbf{W}).$$

*Proof.* One first verifies using the definition of $Z(\mathbf{a})$ that:

$$(\mathbf{g} \cdot \mathbf{W}, \mathbf{a}) = \mathbf{g} \cdot (\mathbf{W}, \mathbf{a})$$

for any $\mathbf{g} \in Z(\mathbf{a})$ and $\mathbf{W} \in \mathsf{Rep}(\mathcal{Q}_L, \mathbf{n})$, where the left-hand side invokes the action of $Z(\mathbf{a})$ on $\mathsf{Rep}(\mathcal{Q}_L, \mathbf{n})$, while the right-hand side invokes the action of $Z(\mathbf{a})$ on $\mathsf{Neur}(\mathcal{Q}_L, \mathbf{n})$. Using this fact and Lemma D.3, we see that: $f_{(\mathbf{g} \cdot \mathbf{W}, \mathbf{a})} = f_{\mathbf{g} \cdot (\mathbf{W}, \mathbf{a})} = f_{(\mathbf{W}, \mathbf{a})}$ and hence, for any $\mathbf{g} \in Z(\mathbf{a})$ and $\mathbf{W} \in \mathsf{Rep}(\mathcal{Q}_L, \mathbf{n})$, we have:

$$\mathcal{L}(\mathbf{g} \cdot \mathbf{W}) = \sum_j \mathcal{C}(f_{(\mathbf{g} \cdot \mathbf{W}, \mathbf{a})}(x_j), y_j) = \sum_j \mathcal{C}(f_{(\mathbf{W}, \mathbf{a})}(x_j), y_j) = \mathcal{L}(\mathbf{W}).$$

$\square$

**Remark D.6.** An alternative proof is as follows. The loss function can be written as the composition

$$\mathsf{Rep}(\mathcal{Q}_L, \mathbf{n}) \xrightarrow{s_{\mathbf{a}}} \mathsf{Neur}(\mathcal{Q}_L, \mathbf{n}) \longrightarrow \mathbb{R}$$

where the first map takes $\mathbf{W}$ to $(\mathbf{W}, \mathbf{a})$ (and hence is a section of the natural projection from $\mathsf{Neur}(\mathcal{Q}_L, \mathbf{n})$ to $\mathsf{Rep}(\mathcal{Q}_L, \mathbf{n})$), and the second map sends $(\mathbf{W}, \mathbf{a})$ to the sum $\sum_j \mathcal{C}(f_{(\mathbf{W}, \mathbf{a})}(x_j), y_j)$. One verifies directly that $s_{\mathbf{a}}$ is equivariant for the action of the group $Z(\mathbf{a})$, while the second map is invariant for $\mathrm{GL}_{\mathbf{n}}^{\mathrm{hidden}}$, of which $Z(\mathbf{a})$ is a subgroup.

**Remark D.7.** Suppose $a_i : \mathbb{R}^{n_i} \to \mathbb{R}^{n_i}$ is pointwise ReLU for each $i$. Then $Z(\mathbf{a})$ consists of those $\mathbf{g} = (g_i) \in \mathrm{GL}_{\mathbf{n}}^{\mathrm{hidden}}$ such that each $g_i \in \mathrm{GL}_{n_i}(\mathbb{R})$ is a diagonal matrix with positive entries along the diagonal. In symbols:

$$Z(\mathbf{a}) = \prod_{i=1}^{L-1} (\mathbb{R}_{>0})^{n_i} .$$

Hence, the "non-negative homogeneity" (or "positive scaling invariance") property of pointwise ReLU activation functions appearing in Dinh et al. (2017); Neyshabur et al. (2015); Meng et al. (2019) emerges as a special case of Proposition D.5.

# E  RADIAL NEURAL NETWORKS AND THE QR DECOMPOSITION

In this appendix, we give a proof of the main theoretical result (Theorem 4.3) involving a QR decomposition for radial neural networks. To this end, we first collect results about radial functions and radial neural networks.

## E.1  RADIAL FUNCTIONS

We first recall the definition of radial functions from Section 4.1 above. Let $h : \mathbb{R} \to \mathbb{R}$ be a piece-wise differentiable function such that $\lim_{r \to 0} \frac{h(r)}{r}$ exists. For any $n \geq 1$, set:

$$h^{(n)} : \mathbb{R}^n \to \mathbb{R}^n; \qquad h^{(n)}(v) := h(|v|) \frac{v}{|v|}.$$

A function $a : \mathbb{R}^n \to \mathbb{R}^n$ is called radial if $a = h^{(n)}$ for some $h : \mathbb{R} \to \mathbb{R}$.

**Lemma E.1.** *Let $a = h^{(n)} : \mathbb{R}^n \to \mathbb{R}^n$ be a radial function on $\mathbb{R}^n$.*

1. *The radial function $a$ commutes with any orthogonal transformation of $\mathbb{R}^n$. In symbols:*
$$a \circ Q = Q \circ a \qquad \text{for any } Q \in O(n).$$

2. *If $m \leq n$ and $\mathrm{inc}_{m,n} : \mathbb{R}^m \hookrightarrow \mathbb{R}^n$ is the standard inclusion, then:*
$$h^{(n)} \circ \mathrm{inc}_{m,n} = \mathrm{inc}_{m,n} \circ h^{(m)}.$$

*Proof.* Suppose $Q \in O(n)$ is an orthogonal transformation of $\mathbb{R}^n$. Since $Q$ is norm-preserving, we have $|Qv| = |v|$ for any $v \in \mathbb{R}^n$. Since $Q$ is linear, we have $Q(\lambda v) = \lambda Q v$ for any $\lambda \in \mathbb{R}$ and $v \in \mathbb{R}^n$. Using the definition of $a = h^{(n)}$ we compute:

$$a(Qv) = \frac{h(|Qv|)}{|Qv|} Qv = \frac{h(|v|)}{|v|} Qv = Q\left(\frac{h(|v|)}{|v|} v\right) = Q(a(v)).$$

The first claim follows. The second claim is an elementary verification. □

**Remark E.2.** More generally, it is straightforward to show that the restriction of the radial function $a$ to a linear subspace of $\mathbb{R}^n$ is a radial function on that subspace.

Next, we recall the definition of a radial neural network (as stated in Section 4.1). Let $L$ be a positive integer and $\mathbf{n}$ be a dimension vector for the neural quiver $\mathcal{Q}_L$. A neural network $(\mathbf{W}, \mathbf{a}) \in \mathrm{Neur}(\mathcal{Q}_L, \mathbf{n})$ is said to be a radial neural network if each activation function $a_i$ is a radial function on $\mathbb{R}^{n_i}$. Let $O(\mathbf{n})$ denote the following product of the orthogonal groups:

$$O(\mathbf{n}) = O(n_1) \times O(n_2) \times \cdots \times O(n_{L-1}).$$

Note that $O(\mathbf{n})$ is a subgroup of $\mathrm{GL}_{\mathbf{n}}^{\text{hidden}}$ and hence acts on $\mathrm{Rep}(\mathcal{Q}_L, \mathbf{n})$. The following proposition involves the loss function associated to a batch of training data as formulated in Section 3.4.

**Proposition E.3.** *For any radial neural network in $\mathrm{Neur}(\mathcal{Q}_L, \mathbf{n})$ and any batch of training data, the loss function $\mathcal{L} : \mathrm{Rep}(\mathcal{Q}_L, \mathbf{n}) \to \mathbb{R}$ is invariant for the action of $O(\mathbf{n})$.*

*Proof.* Let $\mathbf{g} = (g_1, \ldots, g_{L-1}) \in O(\mathbf{n})$. Lemma E.1 implies that the orthogonal transformation $g_i \in O(n_i)$ commutes with the radial function $a_i = h^{(n_i)}$, for each $i$, or, equivalently: $g_i \circ a_i \circ g_i^{-1} = a_i$. Hence $O(\mathbf{n})$ is a subgroup of the centralizer $Z(\mathbf{a})$, and we apply Proposition D.5. □

For clarity, we restate Construction 4.2, which illustrates the flexibility of radial functions in defining the activation functions of a neural network for any given representation of $\mathcal{Q}_L$ with arbitrary dimension vector.

**Construction E.4.** Suppose $\mathbf{h} = (h_1, \ldots, h_L)$ is a tuple of functions [6] $h_i : \mathbb{R} \to \mathbb{R}$, and suppose $\mathbf{n} = (n_0, \ldots, n_L)$ is a dimension vector for $\mathcal{Q}_L$. Consider the tuple of activation functions:

$$\mathbf{a}_{\mathbf{h}}^{(\mathbf{n})} = \left( h_1^{(n_1)}, \ldots, h_L^{(n_L)} \right).$$

Any representation $\mathbf{W}$ of $\mathcal{Q}_L$ with dimension vector $\mathbf{n}$ defines a neural network $\left( \mathbf{W}, \mathbf{a}_{\mathbf{h}}^{(\mathbf{n})} \right)$ as in Definition 3.2. We write simply $(\mathbf{W}, \mathbf{h})$ for this network and $f_{(\mathbf{W}, \mathbf{h})}$ for its neural function.

---

[6]We assume each $h_i$ is piece-wise differentiable and the limit $\lim_{r \to 0} \frac{h_i(r)}{r}$ exists.

### E.2    PROOF OF THEOREM 4.3

In this section, we give a proof of the QR decomposition appearing in Theorem 4.3. In order to do so, we make some recollections from Section 4.2.

Given a dimension vector $\mathbf{n} = (n_0, n_1, \ldots, n_L)$ for the neural quiver $\mathcal{Q}_L$, recall that its reduction $\mathbf{n}^{\text{red}} = (n_0^{\text{red}}, n_1^{\text{red}}, \ldots, n_L^{\text{red}})$ is defined recursively by setting $n_0^{\text{red}} = n_0$, then $n_i^{\text{red}} = \min(n_i, n_{i-1}^{\text{red}} + 1)$ for $i = 1, \ldots, L - 1$, and finally $n_L^{\text{red}} = n_L$. We abbreviate $\mathsf{Rep}(\mathcal{Q}_L, \mathbf{n})$ by $\mathsf{Rep}(\mathbf{n})$ and $\mathsf{Rep}(\mathcal{Q}_L, \mathbf{n}^{\text{red}})$ by $\mathsf{Rep}(\mathbf{n}^{\text{red}})$.

We also had in Section 4.2 the inclusion $\iota : \mathsf{Rep}(\mathbf{n}^{\text{red}}) \hookrightarrow \mathsf{Rep}(\mathbf{n})$, which can be understood in terms of padding matrices with rows and columns of zeros. Another description is as follows. Let $\mathbf{X} = (X_1, \ldots, X_L) \in \mathsf{Rep}(\mathbf{n}^{\text{red}})$. Then $\iota(\mathbf{X})_i = \mathrm{inc}_i \circ X_i \circ \widetilde{\pi}_{i-1}$. Indeed, post-composing $X_i$ with $\mathrm{inc}_i$ amounts to adding $n_i - n_i^{\text{red}}$ rows of zeros on the bottom, while pre-composing $X_i$ with $\widetilde{\pi}_{i-1}$ amounts to adding $n_{i-1} - n_{i-1}^{\text{red}}$ columns of zeros on the right.

Fix a tuple $\mathbf{h} = (h_1, \ldots, h_L)$ of functions $h_i : \mathbb{R} \to \mathbb{R}$. As in Construction 4.2, we attach a radial neural network (and hence a neural function) to every representation of $\mathcal{Q}_L$. As a reminder, we restate Theorem 4.3.

**Theorem E.5.** *(Theorem 4.3) Let $\mathbf{n}$ be a dimension vector for $\mathcal{Q}_L$ and fix $\mathbf{h}$ as above. For any $\mathbf{W} \in \mathsf{Rep}(\mathbf{n})$ there exist:*

$$\mathbf{Q} \in O(\mathbf{n}), \qquad \mathbf{R} = (R_1, \ldots, R_L) \in \mathsf{Rep}(\mathbf{n}^{\text{red}}), \qquad \text{and} \qquad \mathbf{U} \in \mathsf{Rep}(\mathbf{n}),$$

*such that:*

1. *The matrices $R_1, \ldots, R_{L-1}$ are upper triangular.*

2. *The following equality holds: $\mathbf{W} = \mathbf{Q} \cdot \iota(\mathbf{R}) + \mathbf{U}$*

3. *The neural networks defined by $\mathbf{W}$ and $\mathbf{R}$ have identical neural functions: $f_{(\mathbf{W}, \mathbf{h})} = f_{(\mathbf{R}, \mathbf{h})}$.*

Before stating the proof, we first adopt all notational conventions of Section 4.2, especially those listed after the statement of Algorithm 1. Additionally, we note the projection maps:

$$\pi_i : \mathbb{R}^{n_i} \twoheadrightarrow \mathbb{R}^{n_i^{\text{red}}} \qquad\qquad \widetilde{\pi}_i : \mathbb{R}^{1+n_i} \twoheadrightarrow \mathbb{R}^{1+n_i^{\text{red}}}$$

onto the first $n_i^{\text{red}}$ and $1 + n_i^{\text{red}}$ coordinates, respectively. We also have the identity:

$$\mathrm{id}_{n_i+1} = p_{i-1} + \widetilde{\mathrm{inc}}_{i-1} \circ \widetilde{\pi}_{i-1} \tag{E.1}$$

Next, we collect the following basic identities related to the map $\mathrm{ext}_n : \mathbb{R}^n \to \mathbb{R}^{n+1}$ (see Definition B.1) taking $(v_1, \ldots, v_n)$ to $(1, v_1, \ldots, v_n)$. Their proofs are elementary.

**Lemma E.6.** *For $m \leq n$, we have: $\mathrm{ext}_n \circ \mathrm{inc}_{m,n} = \mathrm{inc}_{m+1,n+1} \circ \mathrm{ext}_m$.*

**Lemma E.7.** *For any $n \times m$ matrix $A$, we have[7]: $\mathrm{ext}_n \circ A = \begin{bmatrix} 1 & 0 \\ 0 & A \end{bmatrix} \circ \mathrm{ext}_m$.*

*Proof of Theorem 4.3.* We claim that the outputs $\mathbf{Q}$, $\mathbf{R}$, and $\mathbf{U}$ of Algorithm 1 satisfy the required conditions. The matrices $R_1, \ldots, R_{L-1}$ are upper-triangular by construction, so the first part of the theorem holds.

To show that the neural functions of $(\mathbf{W}, \mathbf{h})$ and $(\mathbf{R}, \mathbf{h})$ coincide, we first show that, for all $i = 1, \ldots, L$, we have:

$$h^{(n_i)} \circ \widetilde{W}_i \circ Q_{i-1} \circ \mathrm{inc}_{i-1} = Q_i \circ \mathrm{inc}_i \circ h^{(n_i^{\text{red}})} \circ \widetilde{R}_i \tag{E.2}$$

---

[7]The matrix $\begin{bmatrix} 1 & 0 \\ 0 & A \end{bmatrix}$ is $(n+1) \times (m+1)$, so the top right 0 abbreviates a row of $m$ zeros and the bottom left 0 abbreviates a column of $n$ zeros.

where $Q_0 = \mathrm{id}_{n_0}$ and $Q_L = \mathrm{id}_{n_L}$. The justification is a computation:

$$h^{(n_i)} \circ \widetilde{W}_i \circ Q_{i-1} \circ \mathrm{inc}_{i-1} = h^{(n_i)} \circ W_i \circ \mathrm{ext}_{n_{i-1}} \circ Q_{i-1} \circ \mathrm{inc}_{i-1}$$

$$= h^{(n_i)} \circ W_i \circ \begin{bmatrix} 1 & 0 \\ 0 & Q_{i-1} \end{bmatrix} \circ \widetilde{\mathrm{inc}}_{i-1} \circ \mathrm{ext}_{n_{i-1}^{\mathrm{red}}}$$

$$= h^{(n_i)} \circ Q_i \circ \mathrm{inc}_i \circ R_i \circ \mathrm{ext}_{n_{i-1}^{\mathrm{red}}} = Q_i \circ \mathrm{inc}_i \circ h^{(n_i^{\mathrm{red}})} \circ \widetilde{R}_i.$$

The first equality uses the definition of $\widetilde{W}_i$ in terms of $W_i$; the second uses the commutativity properties of ext stated in Lemmas E.6 and E.7; the third uses the definitions of $Q_i$, $R_i$, and $V_i$; and the fourth uses the commutativity properties of radial functions as well as the definition of $\widetilde{R}_i$. The fact that $f_{(\mathbf{W},\mathbf{h})} = f_{(\mathbf{R},\mathbf{h})}$ now follows from the definition of the neural function and iterative applications of Equation E.2, noting that $Q_0 \circ \mathrm{inc}_0 = \mathrm{id}_{n_0}$ and $Q_L \circ \mathrm{inc}_L = \mathrm{id}_{n_L}$.

Finally, to show that $\mathbf{W} = \mathbf{Q} \cdot \iota(\mathbf{R}) + \mathbf{U}$, we compute, for $i = 1, \dots, L$:

$$(\mathbf{Q} \cdot \iota(\mathbf{R}) + \mathbf{U})_i = (\mathbf{Q} \cdot \iota(\mathbf{R}))_i + \mathbf{U}_i$$

$$= Q_i \circ \mathrm{inc}_i \circ R_i \circ \widetilde{\pi}_{i-1} \circ \begin{bmatrix} 1 & 0 \\ 0 & Q_{i-1}^{-1} \end{bmatrix} + W_i \circ \begin{bmatrix} 1 & 0 \\ 0 & Q_{i-1} \end{bmatrix} \circ p_{i-1} \circ \begin{bmatrix} 1 & 0 \\ 0 & Q_{i-1}^{-1} \end{bmatrix}$$

$$= W_i \circ \begin{bmatrix} 1 & 0 \\ 0 & Q_{i-1} \end{bmatrix} \circ \widetilde{\mathrm{inc}}_{i-1} \circ \widetilde{\pi}_{i-1} \circ \begin{bmatrix} 1 & 0 \\ 0 & Q_{i-1}^{-1} \end{bmatrix}$$

$$+ W_i \circ \begin{bmatrix} 1 & 0 \\ 0 & Q_{i-1} \end{bmatrix} \circ p_{i-1} \circ \begin{bmatrix} 1 & 0 \\ 0 & Q_{i-1}^{-1} \end{bmatrix}$$

$$= W_i \circ \begin{bmatrix} 1 & 0 \\ 0 & Q_{i-1} \end{bmatrix} \circ \left( \widetilde{\mathrm{inc}}_{i-1} \circ \widetilde{\pi}_{i-1} + p_{i-1} \right) \circ \begin{bmatrix} 1 & 0 \\ 0 & Q_{i-1}^{-1} \end{bmatrix}$$

$$= W_i \circ \begin{bmatrix} 1 & 0 \\ 0 & Q_{i-1} \end{bmatrix} \circ \mathrm{id}_{1+n_{i-1}} \circ \begin{bmatrix} 1 & 0 \\ 0 & Q_{i-1}^{-1} \end{bmatrix}$$

$$= W_i.$$

The first two equalities follow from definitions; the third uses the definitions of $Q_i$, $R_i$, and $V_i$; the fourth uses the distributive law; and the fifth appeals to the Equation E.1. □

## F  PROJECTED GRADIENT DESCENT

In this section, we give a proof of Theorem 4.7, which relates projected gradient descent for a representation with dimension $\mathbf{n}$ to (usual) gradient descent for the corresponding reduced representation with dimension vector $\mathbf{n}^{\mathrm{red}}$.

We adopt all notational conventions from Section 4. In particular, we fix:

- a tuple $\mathbf{h} = (h_1, \dots, h_L)$ of functions, where $h_i : \mathbb{R} \to \mathbb{R}$ for $i = 1, \dots, L$. These can be used to define a neural function $f_{(\mathbf{W},\mathbf{h})}$ (resp. $f_{(\mathbf{X},\mathbf{h})}$) for every representation $\mathbf{W}$ in $\mathrm{Rep}(\mathbf{n})$ (resp. $\mathbf{X}$ in $\mathrm{Rep}(\mathbf{n}^{\mathrm{red}})$). See Construction 4.2 for more details.

- a batch of training data $\{(x_j, y_j)\} \subseteq \mathbb{R}^{n_0} \times \mathbb{R}^{n_L} = \mathbb{R}^{n_0^{\mathrm{red}}} \times \mathbb{R}^{n_L^{\mathrm{red}}}$.

- a cost function $\mathcal{C} : \mathbb{R}^{n_L} \times \mathbb{R}^{n_L} \to \mathbb{R}$

As a result, we have loss functions on $\mathrm{Rep}(\mathbf{n})$ and $\mathrm{Rep}(\mathbf{n}^{\mathrm{red}})$ given by:

$$\mathcal{L} : \mathrm{Rep}(\mathbf{n}) \to \mathbb{R} \qquad\qquad \mathcal{L}_{\mathrm{red}} : \mathrm{Rep}(\mathbf{n}^{\mathrm{red}}) \to \mathbb{R}$$
$$\mathbf{W} \mapsto \sum_j \mathcal{C}(f_{(\mathbf{W},\mathbf{h})}(x_j), y_j) \qquad\qquad \mathbf{X} \mapsto \sum_j \mathcal{C}(f_{(\mathbf{X},\mathbf{h})}(x_j), y_j)$$

as in Section 3.4. Fixing a learning rate $\eta > 0$, there are resulting gradient descent maps on $\mathrm{Rep}(\mathbf{n})$ and $\mathrm{Rep}(\mathbf{n}^{\mathrm{red}})$ given by:

$$\gamma : \mathrm{Rep}(\mathbf{n}) \to \mathrm{Rep}(\mathbf{n}) \qquad\qquad \gamma_{\mathrm{red}} : \mathrm{Rep}(\mathbf{n}^{\mathrm{red}}) \to \mathrm{Rep}(\mathbf{n}^{\mathrm{red}})$$
$$\mathbf{W} \mapsto \mathbf{W} - \eta \cdot \nabla_{\mathbf{W}} \mathcal{L} \qquad\qquad \mathbf{X} \mapsto \mathbf{X} - \eta \cdot \nabla_{\mathbf{X}} \mathcal{L}_{\mathrm{red}}$$

### F.1 THE INTERPOLATING SPACE

In this section, we introduce a subspace $\mathsf{Rep}^{\mathrm{int}}(\mathbf{n})$ of $\mathsf{Rep}(\mathbf{n})$, that, as we will later see, interpolates between $\mathsf{Rep}(\mathbf{n}^{\mathrm{red}})$ and $\mathsf{Rep}(\mathbf{n})$.

Recall the standard inclusions and projections, for $i = 0, 1, \dots, L$:

$$\mathrm{inc}_i = \mathrm{inc}_{n_i^{\mathrm{red}}, n_i} : \mathbb{R}^{n_i^{\mathrm{red}}} \hookrightarrow \mathbb{R}^{n_i} \qquad \widetilde{\mathrm{inc}}_i = \mathrm{inc}_{1+n_i^{\mathrm{red}}, 1+n_i} : \mathbb{R}^{1+n_i^{\mathrm{red}}} \hookrightarrow \mathbb{R}^{1+n_i}$$

$$\pi_i : \mathbb{R}^{n_i} \twoheadrightarrow \mathbb{R}^{n_i^{\mathrm{red}}} \qquad \widetilde{\pi}_i : \mathbb{R}^{1+n_i} \twoheadrightarrow \mathbb{R}^{1+n_i^{\mathrm{red}}}$$

**Definition F.1.** Let $\mathsf{Rep}^{\mathrm{int}}(\mathbf{n})$ denote the subspace of $\mathsf{Rep}(\mathbf{n})$ consisting of those $\mathbf{T} = (T_1, \dots, T_L) \in \mathsf{Rep}(\mathbf{n})$ such that:

$$(\mathrm{id}_{n_i} - \mathrm{inc}_i \circ \pi_i) \circ T_i \circ \widetilde{\mathrm{inc}}_{i-1} \circ \widetilde{\pi}_{i-1} = 0$$

for $i = 1, \dots, L$. Write $\iota_1 : \mathsf{Rep}^{\mathrm{int}}(\mathbf{n}) \hookrightarrow \mathsf{Rep}(\mathbf{n})$ for the natural inclusion.

In other words, the space $\mathsf{Rep}^{\mathrm{int}}(\mathbf{n})$ consists of representations $\mathbf{T} = (T_1, \dots, T_L) \in \mathsf{Rep}(\mathbf{n})$ for which the bottom left $(n_i - n_i^{\mathrm{red}}) \times (1 + n_{i-1}^{\mathrm{red}})$ block of $T_i$ is zero for each $i$:

$$T_i = \begin{bmatrix} * & * \\ 0 & * \end{bmatrix}.$$

We omit the proof of the following proposition, as it involves an elementary analysis of the workings of Algorithm 1.

**Proposition F.2.** *Let* $\mathbf{W} \in \mathsf{Rep}(\mathbf{n})$ *and let* $\mathbf{Q}$, $\mathbf{R}$, *and* $\mathbf{U}$ *be the outputs of Algorithm 1, so* $\mathbf{W} = \mathbf{Q} \cdot \mathbf{R} + \mathbf{U}$. *Then the representation* $\mathbf{Q}^{-1} \cdot \mathbf{W} = \iota(\mathbf{R}) + \mathbf{Q}^{-1} \cdot \mathbf{U}$ *belongs to* $\mathsf{Rep}^{\mathrm{int}}(\mathbf{n})$.

**Definition F.3.** Define a map

$$q_1 : \mathsf{Rep}(\mathbf{n}) \to \mathsf{Rep}^{\mathrm{int}}(\mathbf{n})$$

by taking $\mathbf{T} \in \mathsf{Rep}(\mathbf{n})$ to the representation whose $i$-th slot is

$$W_i - (\mathrm{id}_{n_i} - \mathrm{inc}_i \circ \pi_i) \circ W_i \circ \widetilde{\mathrm{inc}}_{i-1} \circ \widetilde{\pi}_{i-1}.$$

It is straightforward to check that $q_1$ is well-defined, i.e., that the resulting representation belongs to $\mathsf{Rep}^{\mathrm{int}}(\mathbf{n})$ and that $q_1$ is a surjective linear map. The transpose of $q_1$ is the inclusion $\iota_1$. We summarize the situation in the following diagram:

$$\mathsf{Rep}^{\mathrm{int}}(\mathbf{n}) \underset{q_1}{\overset{\iota_1}{\rightleftarrows}} \mathsf{Rep}(\mathbf{n}) \tag{F.1}$$

We observe that the composition $q_1 \circ \iota$ is the identity on $\mathsf{Rep}^{\mathrm{int}}(\mathbf{n})$, while the endomorphism $\iota_1 \circ q_1$ of $\mathsf{Rep}(\mathbf{n})$ takes a representation $\mathbf{W} \in \mathsf{Rep}(\mathbf{n})$ and, for each $i$, zeros all entries in the bottom left $(n_i - n_i^{\mathrm{red}}) \times (1 + n_{i-1}^{\mathrm{red}})$ submatrix of $W_i$.

### F.2 PROJECTED GRADIENT DESCENT AND THE QR DECOMPOSITION

In this section, we define the projected gradient descent map and state a theorem relating projected gradient descent on $\mathsf{Rep}(\mathbf{n})$ with usual gradient descent on $\mathsf{Rep}(\mathbf{n}^{\mathrm{red}})$.

**Definition F.4.** The projected gradient descent map on $\mathsf{Rep}(\mathbf{n})$ with respect to $\mathsf{Rep}^{\mathrm{int}}(\mathbf{n})$ at learning rate $\eta < 0$ is defined as:

$$\gamma_{\mathrm{proj}} : \mathsf{Rep}(\mathbf{n}) \to \mathsf{Rep}(\mathbf{n})$$
$$\mathbf{W} \mapsto \mathbf{W} - \eta \cdot \mathrm{Pr}(\nabla_{\mathbf{W}} \mathcal{L})$$

where $\mathrm{Pr} = \iota_1 \circ q_1$.

One verifies easily that the above definition of projected gradient descent is the same as that given in Definition 4.6. To reiterate, while all entries of each matrix $W_i$ in the representation $\mathbf{W}$ contribute to the computation of the gradient $\nabla_{\mathbf{W}}\mathcal{L}$, only those not in the bottom left submatrix get updated under the projected gradient descent map $\gamma_{\text{proj}}$.

We now restate Theorem 4.7. To ease notation, we identify elements of $\text{Rep}(\mathbf{n}^{\text{red}})$ with their images in $\text{Rep}(\mathbf{n})$ under $\iota$.

**Theorem F.5.** *(Theorem 4.7) Let $\mathbf{W} \in \text{Rep}(\mathbf{n})$ and let $\mathbf{Q}$, $\mathbf{R}$, and $\mathbf{U}$ be the outputs of Algorithm 1. Fix a learning rate $\eta > 0$. For any $k \geq 0$, we have:*

$$\gamma^k(\mathbf{W}) = \mathbf{Q} \cdot \gamma^k(\mathbf{Q}^{-1} \cdot \mathbf{W}) \qquad and \qquad \gamma_{\text{proj}}^k(\mathbf{Q}^{-1} \cdot \mathbf{W}) = \gamma_{\text{red}}^k(\mathbf{R}) + \mathbf{Q}^{-1} \cdot \mathbf{U}.$$

We summarize this result in the following diagram. The left horizontal maps indicate the addition of $\mathbf{Q}^{-1} \cdot \mathbf{U}$, the right horizontal arrows indicate the action of $\mathbf{Q}$, and the vertical maps are various versions of gradient descent. The shaded regions indicate the (smallest) vector space to which the various representations naturally belong.

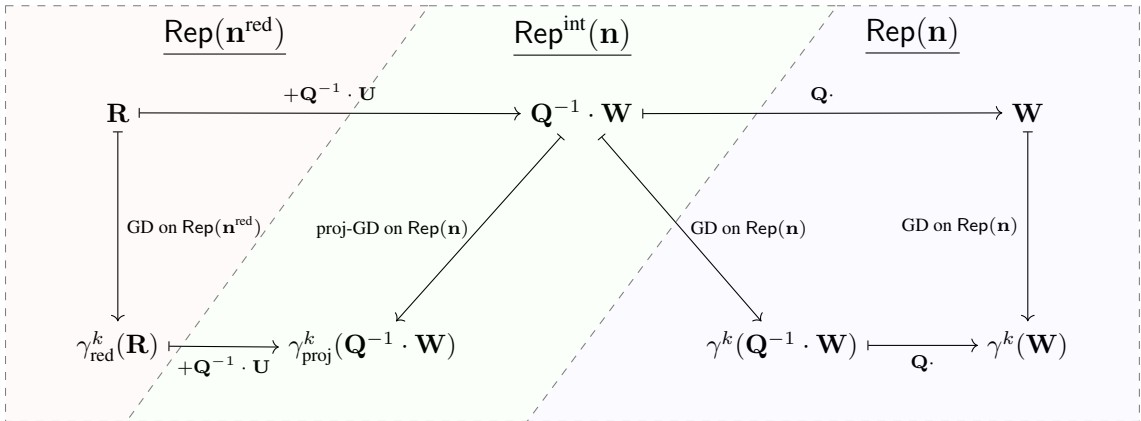

**Remark F.6.** The map $\text{Rep}(\mathbf{n}) \to \text{Rep}(\mathbf{n})$ taking $\mathbf{W}$ to $\mathbf{Q} \cdot \gamma_{\text{proj}}(\mathbf{Q}^{-1} \cdot \mathbf{W})$ is a projected gradient descent map on $\text{Rep}(\mathbf{n})$ with respect to the subspace of $\text{Rep}(\mathbf{n})$ formed by translating $\text{Rep}^{\text{int}}(\mathbf{n})$ by the action of $\mathbf{Q}$, i.e. the subspace $\mathbf{Q} \cdot \text{Rep}^{\text{int}}(\mathbf{n}) = \{\, \mathbf{Q} \cdot \mathbf{T} \mid \mathbf{T} \in \text{Rep}^{\text{int}}(\mathbf{n}) \,\}$.

### F.3    PROOF OF THEOREM 4.7

We begin by explaining the sense in which $\text{Rep}^{\text{int}}(\mathbf{n})$ interpolates between $\text{Rep}(\mathbf{n})$ and $\text{Rep}(\mathbf{n}^{\text{red}})$. One extends Diagram F.1 as follows:

$$\text{Rep}(\mathbf{n}^{\text{red}}) \underset{q_2}{\overset{\iota_2}{\rightleftarrows}} \text{Rep}^{\text{int}}(\mathbf{n}) \underset{q_1}{\overset{\iota_1}{\rightleftarrows}} \text{Rep}(\mathbf{n})$$

- The map
$$\iota_2 : \text{Rep}(\mathbf{n}^{\text{red}}) \hookrightarrow \text{Rep}^{\text{int}}(\mathbf{n})$$
takes $\mathbf{X}$ to the representation whose $i$-th coordinate is $\text{inc}_i \circ X_i \circ \widetilde{\pi}_{i-1}$. It is straightforward to check that $\iota_2$ is a well-defined injective linear map.

- The map
$$q_2 : \text{Rep}^{\text{int}}(\mathbf{n}) \twoheadrightarrow \text{Rep}(\mathbf{n}^{\text{red}})$$
takes $\mathbf{T}$ to the representation whose $i$-th slot is $\pi_i \circ T_i \circ \widetilde{\text{inc}}_{i-1}$. It is straightforward to check that $q_2$ is a surjective linear map. The transpose of $q_2$ is the inclusion $\iota_2$.

**Lemma F.7.** *We have the following:*

1. *The inclusion $\iota : \mathrm{Rep}(\mathbf{n}^{\mathrm{red}}) \hookrightarrow \mathrm{Rep}(\mathbf{n})$ coincides with the composition $\iota_1 \circ \iota_2$, and commutes with the loss functions:*

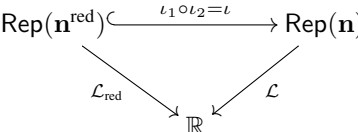

2. *The following diagram commutes:*

$$
\begin{array}{ccc}
\mathrm{Rep}^{\mathrm{int}}(\mathbf{n}) & \xrightarrow{\;\; q_2 \;\;} & \mathrm{Rep}(\mathbf{n}^{\mathrm{red}}) \\
{\scriptstyle \iota_1} \big\downarrow & & \big\downarrow {\scriptstyle \mathcal{L}_{\mathrm{red}}} \\
\mathrm{Rep}(\mathbf{n}) & \xrightarrow{\;\; \mathcal{L} \;\;} & \mathbb{R}
\end{array}
$$

3. *For any $\mathbf{T} \in \mathrm{Rep}^{\mathrm{int}}(\mathbf{n})$, we have: $q_1 \left( \nabla_{\iota_1(\mathbf{T})} \mathcal{L} \right) = \iota_2 \left( \nabla_{q_2(\mathbf{T})} \mathcal{L}_{\mathrm{red}} \right).$*

*Proof.* The identity $\iota = \iota_1 \circ \iota_2$ follows directly from definitions. To prove the commutativity of the diagram, it is enough to show that, for any $\mathbf{X}$ in $\mathrm{Rep}(\mathbf{n}^{\mathrm{red}})$, the neural functions of $\mathbf{X}$ and $\iota(\mathbf{X})$ coincide. This follows easily from the fact that, for $i = 1, \dots, L$, we have:

$$
\pi_i \circ h^{(n_i)} \circ \mathrm{inc}_i = \pi_i \circ \mathrm{inc}_i \circ h^{(n_i^{\mathrm{red}})} = h^{(n_i^{\mathrm{red}})}.
$$

For the second claim, let $\mathbf{T} \in \mathrm{Rep}^{\mathrm{int}}(\mathbf{n})$. It suffices to show that $\iota_1(\mathbf{T})$ and $q_2(\mathbf{T})$ have the same neural function. The key computation is:

$$
\begin{aligned}
\mathrm{inc}_i \circ h^{(n_i^{\mathrm{red}})} \circ \pi_i \circ \widetilde{T}_i \circ \mathrm{inc}_{i-1} &= h^{(n_i)} \circ \mathrm{inc}_i \circ \pi_i \circ T_i \circ \mathrm{ext}_{n_{i-1}} \circ \mathrm{inc}_{i-1} \\
&= h^{(n_i)} \circ \mathrm{inc}_i \circ \pi_i \circ T_i \circ \widetilde{\mathrm{inc}}_{i-1} \circ \mathrm{ext}_{n_{i-1}} \\
&= h^{(n_i)} \circ T_i \circ \widetilde{\mathrm{inc}}_{i-1} \circ \mathrm{ext}_{n_{i-1}} \\
&= h^{(n_i)} \circ T_i \circ \mathrm{ext}_{n_{i-1}} \circ \mathrm{inc}_{i-1} \\
&= h^{(n_i)} \circ \widetilde{T}_i \circ \mathrm{inc}_{i-1}
\end{aligned}
$$

which uses the fact that $(\mathrm{id}_{n_i} - \mathrm{inc}_i \circ \pi_i) \circ T_i \circ \widetilde{\mathrm{inc}}_{i-1} = 0$, or, equivalently, $\mathrm{inc}_i \circ \pi_i \circ T_i \circ \widetilde{\mathrm{inc}}_{i-1} = T_i \circ \widetilde{\mathrm{inc}}_{i-1}$. Applying this relation successively starting with the second-to-last layer $(i = L - 1)$ and ending in the first $(i = 1)$, one obtains the result. For the last claim, one computes $\nabla_{\mathbf{T}}(\mathcal{L} \circ \iota_1)$ in two different ways. The first way is:

$$
\begin{aligned}
\nabla_{\mathbf{T}}(\mathcal{L} \circ \iota_1) &= \left( d_{\mathbf{T}} \left( \mathcal{L} \circ \iota_1 \right) \right)^T = \left( d_{\iota_1(\mathbf{T})} \mathcal{L} \circ d_{\mathbf{T}} \iota_1 \right)^T = \left( d_{\iota_1(\mathbf{T})} \mathcal{L} \circ \iota_1 \right)^T \\
&= \iota_1^T \left( \left( d_{\iota_1(\mathbf{T})} \mathcal{L} \right)^T \right) = q_1 \left( \nabla_{\iota_1(\mathbf{T})} \mathcal{L} \right)
\end{aligned}
$$

where we use the fact that $\iota_1$ is a linear map whose transpose is $q_1$. The second way uses the commutative diagram of the second part of the Lemma:

$$
\begin{aligned}
\nabla_{\mathbf{T}}(\mathcal{L} \circ \iota_1) &= \nabla_{\mathbf{T}} \left( \mathcal{L}_{\mathrm{red}} \circ q_2 \right) = \left( d_{\mathbf{T}} \left( \mathcal{L}_{\mathrm{red}} \circ q_2 \right) \right)^T = \left( d_{q_2(\mathbf{T})} \mathcal{L}_{\mathrm{red}} \circ d_{\mathbf{Z}} q_2 \right)^T \\
&= \left( d_{q_2(\mathbf{T})} \mathcal{L}_{\mathrm{red}} \circ q_2 \right)^T = q_2^T \left( \left( d_{q_2(\mathbf{T})} \mathcal{L}_{\mathrm{red}} \right)^T \right) = \iota_2 \left( \nabla_{q_2(\mathbf{T})} \mathcal{L}_{\mathrm{red}} \right).
\end{aligned}
$$

We also use the fact that $q_2$ is a linear map whose transpose is $\iota_2$. $\qquad \square$

*Proof of Theorem 4.7.* The action of $\mathbf{Q} \in O(\mathbf{n})$ on $\mathrm{Rep}(\mathbf{n})$ is an orthogonal transformation, so the first claim follows from Proposition C.2. For the second claim, it suffices to consider the case $\eta = 1$. The general case follows similarly. We proceed by induction. The base case $k = 0$ amounts to Theorem 4.3. For the induction step, we set

$$
\mathbf{Z}^{(k)} = \iota(\gamma_{\mathrm{red}}^k(\mathbf{R})) + \mathbf{Q}^{-1} \cdot \mathbf{U}.
$$

Each $\mathbf{Z}^{(k)}$ belongs to $\mathsf{Rep}^{\mathrm{int}}(\mathbf{n})$, so $i_1(\mathbf{Z}^{(k)}) = \mathbf{Z}^{(k)}$. Moreover, $q_2\left(\mathbf{Z}^{(k)}\right) = \gamma_{\mathrm{red}}^k(\mathbf{R})$. We compute:

$$
\begin{aligned}
\gamma_{\mathrm{proj}}^{k+1}(\mathbf{Q}^{-1} \cdot \mathbf{W}) &= \gamma_{\mathrm{proj}}\left(\gamma_{\mathrm{proj}}^k(\mathbf{Q}^{-1} \cdot \mathbf{W})\right) \\
&= \gamma_{\mathrm{proj}}\left(\iota(\gamma_{\mathrm{red}}^k(\mathbf{R})) + \mathbf{Q}^{-1} \cdot \mathbf{U}\right) \\
&= \iota(\gamma_{\mathrm{red}}^k(\mathbf{R})) + \mathbf{Q}^{-1} \cdot \mathbf{U} - \iota_1 \circ q_1 \left(\nabla_{\iota(\gamma_{\mathrm{red}}^k(\mathbf{R})) + \mathbf{Q}^{-1} \cdot \mathbf{U}}\mathcal{L}\right) \\
&= \iota(\gamma_{\mathrm{red}}^k(\mathbf{R})) - \iota_1 \circ q_1 \left(\nabla_{\iota_1(\mathbf{Z}^{(k)})}\mathcal{L}\right) + \mathbf{Q}^{-1} \cdot \mathbf{U} \\
&= \iota(\gamma_{\mathrm{red}}^k(\mathbf{R})) - \iota_1 \circ \iota_2 \left(\nabla_{q_2(\mathbf{Z}^{(k)})}\mathcal{L}_{\mathrm{red}}\right) + \mathbf{Q}^{-1} \cdot \mathbf{U} \\
&= \iota\left(\gamma_{\mathrm{red}}^k(\mathbf{R}) - \nabla_{\gamma_{\mathrm{red}}^k(\mathbf{R})}\mathcal{L}_{\mathrm{red}}\right) + \mathbf{Q}^{-1} \cdot \mathbf{U} \\
&= \iota\left(\gamma_{\mathrm{red}}^{k+1}(\mathbf{R})\right) + \mathbf{Q}^{-1} \cdot \mathbf{U}
\end{aligned}
$$

where the second equality uses the induction hypothesis, the third invokes the definition of $\gamma_{\mathrm{proj}}$, the fourth uses the relation between the gradient and orthogonal transformations, the fifth and sixth use Lemma F.7 above, and the last uses the definition of $\gamma_{\mathrm{red}}$. $\qquad\square$

### F.4 EXAMPLE

We now discuss an example where projected gradient descent does not match usual gradient descent.

Let $\mathbf{n} = (1, 3, 1)$ be a dimension vector for the neural quiver $\mathcal{Q}_L$. The space of representations with this dimension vector is 10-dimensional:

$$
\mathsf{Rep}(\mathcal{Q}_L, \mathbf{n}) = \mathrm{Hom}(\mathbb{R}^2, \mathbb{R}^3) \oplus \mathrm{Hom}(\mathbb{R}^4, \mathbb{R}) \simeq \mathbb{R}^{10}.
$$

We identify a representation

$$
\mathbf{W} = \left(W_0 = \begin{bmatrix} a & b \\ c & d \\ e & f \end{bmatrix}, \ W_1 = \begin{bmatrix} g & h & i & j \end{bmatrix}\right) \in \mathsf{Rep}(\mathcal{Q}_L, (1, 3, 1))
$$

with the point $p = (a, b, c, d, e, f, g, h, i, j)$ in $\mathbb{R}^{10}$. The action of the orthogonal group $O(\mathbf{n}) = O(3)$ on $\mathsf{Rep}(\mathcal{Q}_L, \mathbf{n}) \simeq \mathbb{R}^{10}$ can be expressed as:

$$
Q \mapsto \begin{bmatrix} Q & 0 & 0 & 0 \\ 0 & Q & 0 & 0 \\ 0 & 0 & 1 & 0 \\ 0 & 0 & 0 & Q \end{bmatrix}.
$$

Consider the function[8]:

$$
\mathcal{L} : \mathsf{Rep}(\mathcal{Q}_L, \mathbf{n}) \to \mathbb{R}
$$
$$
p = (a, b, c, d, e, f, g, h, i, j) \mapsto h(a + b) + i(c + d) + j(e + f) + g
$$

By the product rule, we have:

$$
\nabla_p \mathcal{L} = (h, h, i, i, j, j, 1, a + b, c + d, e + f)
$$

One easily checks that $\mathcal{L}(Q \cdot p) = \mathcal{L}(p)$ and that $\nabla_{Q \cdot p}\mathcal{L} = Q \cdot \nabla_p \mathcal{L}$ for any $Q \in O(3)$.

The interpolating space is the subspace of $\mathsf{Rep}(\mathcal{Q}_L, \mathbf{n}) \simeq \mathbb{R}^8$ with $e = f = 0$. Suppose $p' = (a, b, c, d, 0, 0, g, h, i, j)$ belongs to the interpolating space. Then the gradient is

$$
\nabla_{p'}\mathcal{L} = (h, h, i, i, j, j, 1, a + b, c + d, 0)
$$

which does not belong to the interpolating space. So one step of usual gradient descent, with learning rate $\eta > 0$ yields:

$$
\begin{aligned}
\gamma : p' = (a, b, c, d, 0, 0, g, h, i, j) \mapsto \\
(a - \eta h, \ b - \eta h, \ c - \eta i, \ d - \eta i, \ -\eta j, \ -\eta j, \ g - \eta, \ h - \eta(a + b), \ i - \eta(c + d), \ j)
\end{aligned}
$$

---

[8]For $\mathbf{W} \in \mathsf{Rep}(\mathcal{Q}_L, \mathbf{n})$, the neural function of the neural network with affine maps determined by $\mathbf{W}$ and identity activation functions is $\mathbb{R} \to \mathbb{R}$; $x \mapsto \mathcal{L}(\mathbf{W})x$. The function $\mathcal{L}$ can appear as a loss function for certain batches of training data and cost function on $\mathbb{R}$.

On the other hand, one step of projected gradient descent yields:

$$\gamma_{\text{proj}} : p' = (a, b, c, d, 0, 0, g, h, i, j) \mapsto$$
$$(a - \eta h \,,\, b - \eta h \,,\, c - \eta i \,,\, d - \eta i \,,\, 0 \,,\, 0 \,,\, g - \eta \,,\, h - \eta(a + b) \,,\, i - \eta(c + d) \,,\, j)$$

Direct computation shows that the difference between the evaluation of $\mathcal{L}$ after one step of gradient descent and the evaluation of $\mathcal{L}$ after one step of projected gradient descent is:

$$\mathcal{L}(\gamma(p')) - \mathcal{L}(\gamma_{\text{proj}}(p')) = 2\eta j^2.$$

## G  QUIVER NEURAL NETWORKS IN GENERAL

In this section, we generalize some of our main results to a larger class of quivers beyond the neural quiver. These more general quivers are known as "neural-adapted quivers", defined presently.

### G.1  NEURAL-ADAPTED QUIVERS

**Definition G.1.** A **neural-adapted quiver** is a quiver $\mathcal{Q} = (I, E)$ together with a partition $I = I_{\text{in}} \cup B \cup H \cup \{i_{\text{out}}\}$ of the vertex set such that the following hold:

1. The quiver $\mathcal{Q}$ is acyclic and no two edges share the same source and target.

2. The vertex $i_{\text{out}}$ is the unique sink of $\mathcal{Q}$. This is the **output** vertex.

3. The set $I_{\text{in}}$ is non-empty, and every vertex therein is a source of $\mathcal{Q}$. These are the **input** vertices.

4. Every vertex in $B$ is a source of $\mathcal{Q}$. These are the **bias** vertices.

5. Every vertex in $H$ is neither a source nor a sink, and admits a path from an input vertex. These are the **hidden** vertices.

Given a quiver satisfying condition (1) above, and with a unique sink, one can partition its vertex set in order to give it the structure of a neural-adapted quiver. This can be done, for example, by setting $I_{\text{in}}$ to be the set of sinks, $H$ to be the non-sink and non-source vertices, and $B$ to be empty. In general, there are many possible partitions, since some sources can be input vertices and some can be bias vertices (keeping in mind that any hidden vertex must admit a path from a input vertex). As a final note, we observe that an acyclic directed graph with a unique sink must be connected.

**Example G.2.** We list examples of neural-adapted quivers:

1. The neural quiver with five layers:

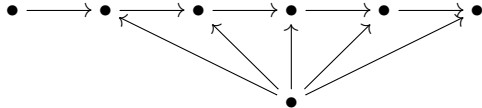

One can see that there is a unique neural-adapted structure on this quiver. Namely, the unique sink is the output vertex, the source on the far left is the sole input vertex, and the source on the bottom is the sole bias vertex. All other vertices are hidden vertices, and each admits a unique path from the input vertex.

2. The no-bias neural quiver with five layers:

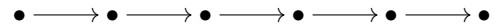

There is a unique neural-adapted structure on this quiver, and it has the property that the set of bias vertices is empty.

3. A skip-connection neural quiver with five layers:

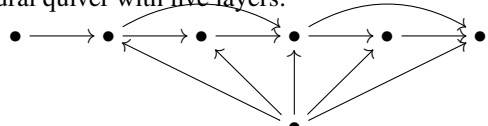

Once again, there is a unique neural-adapted structure on this quiver.

Given a neural adapted quiver, one exclusively considers dimension vectors whose value at the bias vertices is equal to $1$. Thus, a dimension vector for a neural-adapted quiver amounts to a tuple of non-negative integers indexed by the vertices in $I \setminus B = I_{\text{in}} \cup H \cup \{i_{\text{out}}\}$. We abbreviate $d_{\text{in}} := \sum_{i \in I_{\text{in}}} d_i$ and $d_{\text{out}} := d_{i_{\text{out}}}$.

**Remark G.3.** As a directed acyclic graph, a neural-adapted $\mathcal{Q}$ has a "topological ordering" of the vertices. In what follows, we will be implicitly be choosing an enumeration $\{i_1, \ldots, i_{|I|}\}$ of the set $I$ of vertices that is compatible with the topological ordering. That is, if $i_j \prec i_{j'}$ in the topological ordering, then $j < j'$. In particular, $i_{|I|} = i_{\text{out}}$. Furthermore, we will assume all source vertices (that is, those in $I_{\text{rm in}} \cup B$) appear before all non-source vertices. Hence,

$$\{i_{|I_{\text{in}}|+|B|+1}, \ldots, i_{|I|} = i_{\text{out}}\}$$

is an enumeration of the non-source vertices.

Additionally, whenever necessary, we will implicitly fix an enumeration of the edges $\{e_1, \ldots, e_{|E|}\}$. For example, this occurs in the following definition:

**Definition G.4.** Let $\mathbf{W}$ be a representation of a neural-adapted quiver $\mathcal{Q}$ with dimension vector $\mathbf{d}$. For each vertex in $H \cup \{i_{\text{out}}\}$, set $W_i$ to be the following matrix:

$$W_i \;=\; [W_{e_1} \; \ldots \; W_{e_r}]$$

where $\{e_1, \ldots, e_r\} = t^{-1}(i)$ is the set of edges whose target is $i$. Note that the size of $W_i$ is $d_i \times (d_{j_1} + \cdots + d_{j_r})$.

**Lemma G.5.** *Let $\mathcal{Q}$ be a quiver[9], and fix an enumeration of the edges. Specifying a representation of $\mathcal{Q}$ with dimension vector $\mathbf{d}$ is the same as specifying a $d_i \times (d_{j_1} + \cdots + d_{j_r})$ matrix $W_i$ for every non-source vertex $i$.*

## G.2 $\mathcal{Q}$-NEURAL NETWORKS

We now define the notion of a $\mathcal{Q}$-neural network, that is, a neural network whose underlying architecture is specified by a neural-adapted quiver.

**Definition G.6.** Let $\mathcal{Q}$ be a neural-adapted quiver. A $\mathcal{Q}$-**neural network** consists of the following:

1. **Hyperparameters.** A dimension vector $\mathbf{d}$ for the quiver $\mathcal{Q}$.

2. **Trainable parameters.** A representation $\mathbf{W}$ of $\mathcal{Q}$ with dimension vector $\mathbf{d}$.

3. **Activation functions.** Piece-wise differentiable functions $a_i : \mathbb{R}^{d_i} \to \mathbb{R}^{d_i}$ for $i \in H \cup \{i_{\text{out}}\}$. These are grouped into a tuple $\mathbf{a}$.

We denote by $\text{Neur}(\mathcal{Q}, \mathbf{d})$ the vector space of $\mathcal{Q}$-neural networks layers and dimension vector $\mathbf{d}$, and regard elements therein as pairs $(\mathbf{W}, \mathbf{a})$. The neural function (also known as the feedforward function) of a $\mathcal{Q}$-neural network is defined recursively, as explained in the following definition.

**Definition G.7.** Let $\mathcal{Q}$ be a neural-adapted quiver. Let $(\mathbf{W}, \mathbf{a})$ be a $\mathcal{Q}$-neural network with dimension vector $\mathbf{d}$. For every non-bias vertex $i \in I$, define a function:

$$f_{i,(\mathbf{W},\mathbf{a})} = f_i : \mathbb{R}^{d_{\text{in}}} \to \mathbb{R}^{d_i}$$

by setting:

- If $i$ is an input vertex, $f_i$ is the projection map.

- If $i$ is a bias vertex, $f_i \equiv 1$ is the constant function at $1 \in \mathbb{R}^{d_i} = \mathbb{R}$.

- For all other vertices, we use the recursive definition: $f_i(x) = a_i \left( \sum_{j \xrightarrow{e} i} W_e \circ f_j(x) \right)$

The **neural function** of the neural network $(\mathbf{W}, \mathbf{a})$ is defined as:

$$f_{(\mathbf{W},\mathbf{a})} := f_{i_{\text{out}}} : \mathbb{R}^{d_{\text{in}}} \to \mathbb{R}^{d_{\text{out}}}$$

---

[9]This works for any quiver, and in particular for neural-adapted quivers

Note that the neural function is well-defined due to the topological ordering of a neural-adapted quiver, regarded as a connected directed acyclic graph. Indeed, one can implement this definition using an ordering of the vertices compatible with the topological ordering (see Remark G.3).

**Definition G.8.** Let $\mathbf{W}$ be a representation of a neural-adapted quiver $\mathcal{Q}$ with dimension vector $\mathbf{d}$. Let $i$ be a non-source vertex[10] of $\mathcal{Q}$. Let $\{e_1, \ldots, e_r\} = t^{-1}(i)$ be the (ordered) set of edges whose target is $i$, and let $j_1$ be the source of $e_1$, $j_2$ of $e_2$, and so forth.

1. Set $W_i$ to be the following matrix:
$$W_i = [W_{e_1} \ \ldots \ W_{e_r}]$$
   Note that the size of $W_i$ is $d_i \times (d_{j_1} + \cdots + d_{j_r})$.

2. For $x \in \mathbb{R}^{d_{\text{in}}}$, set $f_i^{\text{pre}}(x)$ to be the following vector:
$$f_i^{\text{pre}}(x) = \begin{bmatrix} f_{j_1}(x) \\ \vdots \\ f_{j_r}(x) \end{bmatrix}.$$
   Hence, $f_i^{\text{pre}}(x)$ is a column vector of dimension $(d_{j_1} + \cdots + d_{j_r})$.

The proof of the following lemma is elementary.

**Lemma G.9.** *Let $(\mathbf{W}, \mathbf{a})$ be a representation of a neural-adapted quiver $\mathcal{Q}$. For any non-source vertex $i \in I$, we have:*
$$f_{i,(\mathbf{W},\mathbf{a})}(x) = a_i \circ W_i \circ f_{i,(\mathbf{W},\mathbf{a})}^{\text{pre}}(x)$$

## G.3 Symmetry Group

Throughout this section, we fix a neural-adapted quiver $\mathcal{Q}$ with partition $I = I_{\text{in}} \cup B \cup H \cup \{i_{\text{out}}\}$ of the vertex set as described above.

**Definition G.10.** The **parameter symmetry group** for representations of a neural-adapted quiver $\mathcal{Q}$ with dimension vector $\mathbf{d}$ is:
$$\text{GL}_{\mathbf{d}}^{\text{hidden}} = \prod_{i \in H} \text{GL}_{d_i}.$$

This group acts on $\text{Rep}(\mathcal{Q}, \mathbf{d})$ as:
$$\mathbf{g} \cdot \mathbf{W} = \left( g_{t(e)} \, W_e \, g_{s(e)}^{-1} \right)_e$$

This group acts on $\text{Neur}(\mathcal{Q}, \mathbf{d})$ as:
$$\mathbf{g} \cdot (\mathbf{W}, \mathbf{a}) = \left( \left( g_{t(e)} \, W_e \, g_{s(e)}^{-1} \right)_e, (g_i \, a_i \, g_i^{-1})_i \right)$$

In both cases, by convention, we have $g_i = \text{id}_{d_i}$ for any non-hidden vertex $i$.

**Remark G.11.** Following the notation appearing in Definition G.4, we have that $\mathbf{g} \in \text{GL}_{\mathbf{d}}^{\text{hidden}}$ transforms the matrix $W_i$ corresponding to a vertex $i$ to:
$$g_i \, W_i \begin{bmatrix} g_{j_1}^{-1} & 0 & \cdots & 0 \\ 0 & g_{j_2}^{-1} & \cdots & 0 \\ \vdots & & \ddots & \vdots \\ 0 & 0 & \cdots & g_{j_r}^{-1} \end{bmatrix}$$

We have the following generalization of Lemma D.3:

**Lemma G.12.** *For any neural network $(\mathbf{W}, \mathbf{a})$ in $\text{Neur}(\mathcal{Q}, \mathbf{d})$ and any $\mathbf{g}$ in $\text{GL}_{\mathbf{d}}^{\text{hidden}}$, the neural functions of $(\mathbf{W}, \mathbf{a})$ and $\mathbf{g} \cdot (\mathbf{W}, \mathbf{a})$ coincide:*
$$f_{\mathbf{g} \cdot (\mathbf{W},\mathbf{a})} = f_{(\mathbf{W},\mathbf{a})}$$

---

[10]The non-source vertices are precisely $H \cup \{i_{\text{out}}\}$.

*Sketch of proof.* It suffices to show that: $f_{i,\mathbf{g}\cdot(\mathbf{W},\mathbf{a})} = g_i \circ f_{i,(\mathbf{W},\mathbf{a})}$ for all vertices $i$, where, by convention, $g_i = \mathrm{id}_{d_i}$ for non-hidden vertices. One proceeds by induction using the topological ordering of the quiver, viewed as a connected directed acyclic graph. The base step is when $i$ is a bias or input vertex. The induction step involves the following calculation:

$$f_{i,\mathbf{g}\cdot(\mathbf{W},\mathbf{a})}(x) = g_i \circ a_i \circ g_i^{-1} \left( \sum_{j \xrightarrow{e} i} g_i \circ W_e \circ g_j^{-1} \circ g_j \circ f_{j,(\mathbf{W},\mathbf{a})}(x) \right)$$

$$= g_i \circ a_i \left( \sum_{j \xrightarrow{e} i} W_e \circ f_{j,(\mathbf{W},\mathbf{a})}(x) \right) = g_i \circ f_{i,(\mathbf{W},\mathbf{a})}(x)$$

Alternatively, one can use the matrices $W_i$ and Lemma G.9. $\qquad\square$

**Definition G.13.** The **stabilizer** in $\mathrm{GL}_\mathbf{d}^{\mathrm{hidden}}$ of a choice of activation functions $\mathbf{a}$ is defined as:

$$Z(\mathbf{a}) = \{\mathbf{g} = (g_i) \in \mathrm{GL}_\mathbf{d}^{\mathrm{hidden}} \quad : \quad g_i \circ a_i \circ g_i^{-1} = a_i \text{ for every hidden vertex } i\}.$$

Note that each $a_i$ is a function from $\mathbb{R}^{d_i}$ to $\mathbb{R}^{d_i}$, as is each $g_i$, so the equality $g_i \circ a_i \circ g_i^{-1} = a_i$ is as functions from $\mathbb{R}^{d_i}$ to $\mathbb{R}^{d_i}$. As a subgroup of $\mathrm{GL}_\mathbf{d}^{\mathrm{hidden}}$, the stabilizer $Z(\mathbf{a})$ acts on $\mathrm{Rep}(\mathcal{Q},\mathbf{d})$.

## G.4 THE LOSS FUNCTION

For the remainder of this section, we fix a neural-adapted quiver $\mathcal{Q}$ and a dimension vector $\mathbf{d}$ for $\mathcal{Q}$. Furthermore, fix activation functions $\mathbf{a} = (a_i : \mathbb{R}^{d_i} \to \mathbb{R}^{d_i})$, one for each hidden vertex, and one for the output vertex. To a batch of training data $\mathbb{X} = \{(x_j, y_j)\} \subseteq \mathbb{R}^{d_{\mathrm{in}}} \times \mathbb{R}^{d_{\mathrm{out}}}$, there is an associated **loss function** on $\mathrm{Rep}(\mathcal{Q},\mathbf{d})$ defined as

$$\mathcal{L} = \mathcal{L}_\mathbb{X} : \mathrm{Rep}(\mathcal{Q},\mathbf{d}) \to \mathbb{R} \qquad\qquad \mathcal{L}(\mathbf{W}) = \sum_j \mathcal{C}(f_{(\mathbf{W},\mathbf{a})}(x_j), y_j)$$

where $\mathcal{C} : \mathbb{R}^{d_{\mathrm{out}}} \times \mathbb{R}^{d_{\mathrm{out}}} \to \mathbb{R}$ is a cost function. Following the formalism of Appendix C, we obtain a gradient descent map

$$\gamma : \mathrm{Rep}(\mathcal{Q},\mathbf{d}) \to \mathrm{Rep}(\mathcal{Q},\mathbf{d})$$

for any learning rate $\eta > 0$.

We demonstrate that the loss function is invariant for the action of the group $Z(\mathbf{a})$. We omit a proof of the following proposition, as it is completely analogous to that of Proposition D.5.

**Proposition G.14.** *Let $\mathbf{a}$ be activation functions, and let $\mathcal{L}$ be the loss function associated to a batch of training data. Then, for all $\mathbf{g} \in Z(\mathbf{a})$ and $\mathbf{W} \in \mathrm{Rep}(\mathcal{Q},\mathbf{d})$, we have:*

$$\mathcal{L}(\mathbf{g} \cdot \mathbf{W}) = \mathcal{L}(\mathbf{W}).$$

## G.5 RADIAL $\mathcal{Q}$-NEURAL NETWORKS

Suppose $\mathbf{h} = (h_1, \ldots, h_L)$ is a tuple of functions [11] $h_i : \mathbb{R} \to \mathbb{R}$, and suppose $\mathbf{d}$ is a dimension vector for a neural-adapted quiver $\mathcal{Q}$. Consider the tuple of activation functions:

$$\mathbf{a}_\mathbf{h}^{(\mathbf{n})} = \left( h_i^{(d_i)} \right)_i,$$

where $i$ runs over $H \cup \{i_{\mathrm{out}}\}$. Any representation $\mathbf{W}$ of $\mathcal{Q}$ with dimension vector $\mathbf{d}$ defines a radial neural network $\left( \mathbf{W}, \mathbf{a}_\mathbf{h}^{(\mathbf{n})} \right)$. We write simply $(\mathbf{W}, \mathbf{h})$ for this radial neural network and $f_{(\mathbf{W},\mathbf{h})}$ for its neural function.

The following definition implicitly invokes the topological ordering of a neural-adapted quiver, regarded as a directed acyclic graph. Recall that, as always, the dimension vector of a neural-adapted quiver has value 1 on each bias vertex.

---

[11] We assume each $h_i$ is piece-wise differentiable and the limit $\lim_{r \to 0} \frac{h_i(r)}{r}$ exists.

**Definition G.15.** Given a dimension vector $\mathbf{d}$ for a neural-adapted quiver $\mathcal{Q}$, one defines the **reduced dimension vector** $\mathbf{d}^{\text{red}}$ as follows. First, $d_i^{\text{red}} = d_i$ for every $i \in I_{\text{in}} \cup B$. Next, for any hidden vertex $i$, recursively set $d_i^{\text{red}}$ to be minimum of $d_i$ and $\sum_{j \in I_{\to i}} d_j^{\text{red}}$. Finally, $d_{\text{out}}^{\text{red}} = d_{\text{out}}$ for the output vertex.

There is an inclusion $\iota : \mathsf{Rep}(\mathcal{Q}, \mathbf{d}^{\text{red}}) \hookrightarrow \mathsf{Rep}(\mathcal{Q}, \mathbf{d})$.

**Definition G.16.** Let $\mathbf{d}$ be a dimension vector for a neural-adapted quiver $\mathcal{Q}$. Define a subgroup of $\mathrm{GL}_{\mathbf{d}}^{\text{hidden}}$ as follows:

$$O(\mathbf{d}) = \prod_{i \in H} O(d_i).$$

We state the following conjecture:

**Conjecture G.17.** *Let $\mathbf{d}$ be a dimension vector for a neural-adapted quiver $\mathcal{Q}$ and fix $\mathbf{h}$ as above. For any $\mathbf{W} \in \mathsf{Rep}(\mathcal{Q}, \mathbf{d})$ there exist:*

$$\mathbf{Q} \in O(\mathbf{d}), \qquad \mathbf{R} = (R_1, \ldots, R_L) \in \mathsf{Rep}(\mathcal{Q}, \mathbf{d}^{\text{red}}), \qquad \text{and} \qquad \mathbf{U} \in \mathsf{Rep}(\mathcal{Q}, \mathbf{d}),$$

*such that:*

1. *The matrix $R_i$ is upper triangular for any hidden vertex $i$.*
2. *The following equality holds:* $\mathbf{W} = \mathbf{Q} \cdot \iota(\mathbf{R}) + \mathbf{U}$
3. *The neural networks defined by $\mathbf{W}$ and $\mathbf{R}$ have identical neural functions:* $f_{(\mathbf{W}, \mathbf{h})} = f_{(\mathbf{R}, \mathbf{h})}$.

Moreover, we conjecture that a generalization of Algorithm 1 produces $\mathbf{R}$, $\mathbf{Q}$ and $\mathbf{U}$ from $\mathbf{W}$. We first make the following notes:

- Fix an enumeration $\{i_1, \ldots, i_{|I|}\}$ of the set $I$ of vertices that is compatible with the topological ordering, as in Remark G.3. Furthermore, we will assume all source vertices (that is, those in $I_{\text{rm in}} \cup B$) appear before all non-source vertices. Hence,

$$\{i_{|I_{\text{in}}| + |B| + 1}, \ldots, i_{|I| - 1}\}$$

  is an enumeration of the hidden vertices.

- Fix an enumeration of the edges of $\mathcal{Q}$. Hence, a representation of $\mathcal{Q}$ with a specified dimension vector can be regarded either as a tuple $\mathbf{W} = (W_e)_e$ of matrices indexed by the edges, or as a tuple of matrices $\mathbf{W} = (W_i)_i$ of matrices indexed by the vertices (see Lemma G.5). We use the latter formulation in the algorithm, except for one step.

- It is enough to produce $\mathbf{R}$ and $\mathbf{Q}$, as we can subsequently set: $\mathbf{U} = \mathbf{W} - \mathbf{Q} \cdot \mathbf{R}$.

---

**Algorithm 2:** General version of QR Dimensional Reduction (`QRDimRed`)

---

**input**   : $\mathbf{W} \in \mathsf{Rep}(\mathcal{Q}, \mathbf{d})$
**output**  : $\mathbf{Q}, \mathbf{R}$

$\mathbf{Q}, \mathbf{R} \leftarrow [\,], [\,]$                    // initialize output matrix lists

**for** $j \leftarrow |I_{\mathrm{in}}| + |B| + 1$ **to** $|I| - 1$ **do**      // iterate through hidden vertices

> $i \leftarrow i_j$                                // current vertex
> **if** $d_i^{\mathrm{red}} < d_i$ **then**
> > $Q_i, R_i \leftarrow$ QR-decomp($V_i$, *mode = complete*)        // $V_i = Q_i \circ \mathrm{inc}_i \circ R_i$
>
> **else**
> > $Q_i, R_i \leftarrow$ QR-decomp($V_i$)                      // $V_i = Q_i \circ R_i$
>
> **end**
> Append $Q_i$ to $\mathbf{Q}$
> Append $R_i$ to $\mathbf{R}$                    // reduced weights for layer $i$
> **for** $e$ such that $s(e) = i$ **do**                    // update next layers
>
> > $W_e \leftarrow W_e Q_i$ [a]
>
> **end**

**end**
Append $W_{i_{\mathrm{out}}}$ to $\mathbf{R}$

**return Q,R**

---

[a] In terms of indexing-by-vertices, we have that $W_{t(e)}$ gets updated to the product of $W_{t(e)}$ and the block diagonal matrix $\mathrm{Diag}(\mathrm{id}, \ldots, Q_i, \ldots, \mathrm{id})$, with the identity matrix (of the appropriate size) in each block, except for the block corresponding to $e$, which has $Q_i$. Note that we have fixed an ordering of the edges.

## H  DISCUSSION

In this section, we first consider how our framework specializes to the case of no bias, and then how it generalizes to shifts within the radial functions.

### H.1  SPECIAL CASE: NO-BIAS VERSION

We now consider neural networks with only linear maps between successive layers, rather than the more general setting of affine maps. In other words, there are no bias vectors.

Let $L$ be a positive integer. The **no-bias neural quiver** $\overline{\mathcal{Q}}_L$ is the following quiver with $L + 1$ vertices:

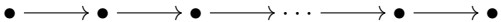

A representation of this quiver with dimension vector $\mathbf{n} = (n_0, \ldots, n_L)$ consists of a linear map from $\mathbb{R}^{n_{i-1}}$ to $\mathbb{R}^{n_i}$ for $i = 1, \ldots, L$; hence $\mathsf{Rep}(\overline{\mathcal{Q}}_L, \mathbf{n}) = \bigoplus_{i=1}^{L} \mathsf{Hom}(\mathbb{R}^{n_{i-1}}, \mathbb{R}^{n_i})$. The corresponding **no-bias reduced dimension vector** $\overline{\mathbf{n}}^{\mathrm{red}} = (\overline{n}_0^{\mathrm{red}}, \overline{n}_1^{\mathrm{red}}, \ldots, \overline{n}_L^{\mathrm{red}})$ is defined recursively by setting $\overline{n}_0^{\mathrm{red}} = n_0$, then $\overline{n}_i^{\mathrm{red}} = \min(n_i, \overline{n}_{i-1}^{\mathrm{red}})$ for $i = 1, \ldots, L - 1$, and finally $\overline{n}_L^{\mathrm{red}} = n_L$. Since $\overline{n}_i^{\mathrm{red}} \leq n_i$ for all $i$, we have an obvious inclusion $\mathsf{Rep}(\overline{\mathcal{Q}}_L, \overline{\mathbf{n}}^{\mathrm{red}}) \hookrightarrow \mathsf{Rep}(\overline{\mathcal{Q}}_L, \mathbf{n})$ and identify $\mathsf{Rep}(\overline{\mathcal{Q}}_L, \overline{\mathbf{n}}^{\mathrm{red}})$ with its image in $\mathsf{Rep}(\overline{\mathcal{Q}}_L, \mathbf{n})$. Given functions $\mathbf{h} = (h_1, \ldots, h_L)$, one adapts Construction 4.2 to define a radial neural network for every representation of $\overline{\mathcal{Q}}_L$, where the trainable parameters define linear maps rather than affine maps.

**Proposition H.1.** *Theorem 4.3 holds with the neural quiver replaced by the no-bias neural quiver $\overline{\mathcal{Q}}_L$ and the reduced dimension vector replaced by the no-bias reduced dimension vector $\overline{\mathbf{n}}^{\mathrm{red}}$.*

We illustrate an example of the reduction for no-bias radial neural networks in Figure 5 (which is the same as Figure 3). Versions of Algorithm 1 and Theorem 4.7 also hold in the no-bias case, where one uses projected gradient descent with respect to the subspace $\mathsf{Rep}^{\mathrm{int}}(\overline{\mathcal{Q}}_L, \mathbf{n})$ of representations $\mathbf{T}$ having the lower left $(n_i - \overline{n}_i^{\mathrm{red}}) \times (\overline{n}_{i-1}^{\mathrm{red}})$ block of each $T_i$ equal to zero.

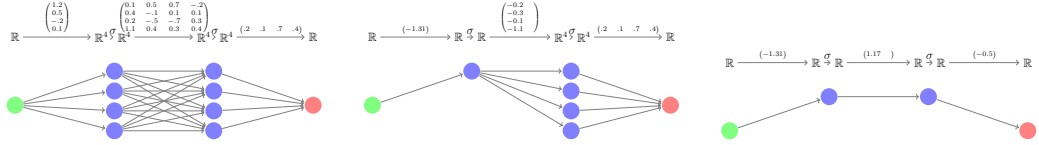

Figure 5: Parameter reduction in 3 steps. Since the activation function is radial, it commutes with orthogonal transformations. This example has $L = 3$, $\mathbf{n} = (1, 4, 4, 1)$, and no bias. The reduced dimension vector is $\overline{\mathbf{n}}^{\text{red}} = (1, 1, 1, 1)$. The number of trainable parameters reduces from 24 to 3.

**Example H.2.** We give an example where projected gradient descent does not match usual gradient descent. This example is a simpler version of that appearing in F.4.

Let $\mathbf{n} = (1, 2, 1)$ be a dimension vector for the no-bias quiver $\overline{\mathcal{Q}}_L$. The space of representations with this dimension vector is 4-dimensional:

$$\mathsf{Rep}(\overline{\mathcal{Q}}_L, \mathbf{n}) = \mathrm{Hom}(\mathbb{R}, \mathbb{R}^2) \oplus \mathrm{Hom}(\mathbb{R}^2, \mathbb{R}) \simeq \mathbb{R}^4.$$

We identify a representation

$$\mathbf{W} = \left( W_0 = \begin{bmatrix} a \\ b \end{bmatrix}, \; W_1 = [c \quad d] \right) \in \mathsf{Rep}(\overline{\mathcal{Q}}_L, (1, 2, 1))$$

with the point $p = (a, b, c, d)$ in $\mathbb{R}^4$. The action of the orthogonal group $O(\mathbf{n}) = O(2)$ on $\mathsf{Rep}(\overline{\mathcal{Q}}_L, \mathbf{n}) \simeq \mathbb{R}^4$ is the diagonal action:

$$Q \mapsto \begin{bmatrix} Q & 0 \\ 0 & Q \end{bmatrix}.$$

To be explicit: if $Q = \begin{bmatrix} \cos(\theta) & -\sin(\theta) \\ \sin(\theta) & \cos(\theta) \end{bmatrix}$ is an orthogonal transformation, then $Q \cdot (a, b, c, d) = (a\cos(\theta) - b\sin(\theta), a\sin(\theta) + b\cos(\theta), c\cos(\theta) - d\sin(\theta), c\sin(\theta) + d\cos(\theta))$.

Consider the function[12]:

$$\mathcal{L} : \mathsf{Rep}(\overline{\mathcal{Q}}_L, \mathbf{n}) \to \mathbb{R}$$
$$p = (a, b, c, d) \mapsto ac + bd$$

By the product rule, we have:

$$\nabla_p \mathcal{L} = (c, d, a, b)$$

One easily checks that $\mathcal{L}(Q \cdot p) = \mathcal{L}(p)$ and that $\nabla_{Q \cdot p} \mathcal{L} = Q \cdot \nabla_p \mathcal{L}$ for any $Q \in O(2)$.

The reduced dimension vector is $\overline{\mathbf{n}}^{\text{red}} = (1, 1, 1)$, so $\mathsf{Rep}(\overline{\mathcal{Q}}_L, \overline{\mathbf{n}}^{\text{red}}) = \mathrm{Hom}(\mathbb{R}, \mathbb{R}) \oplus \mathrm{Hom}(\mathbb{R}, \mathbb{R}) \simeq \mathbb{R}^2$. The reduced version of $\mathcal{L}$ is given by:

$$\mathcal{L}_{\text{red}} : \mathsf{Rep}(\overline{\mathcal{Q}}_L, \overline{\mathbf{n}}^{\text{red}}) \to \mathbb{R}$$
$$(a, c) \mapsto ac$$

The QR decomposition of the representation $\mathbf{W}$ corresponding to $p = (a, b, c, d)$ is:

$$(a, b, c, d) = Q \cdot \left( \sqrt{a^2 + b^2}, \; 0, \; \frac{ac + bd}{\sqrt{a^2 + b^2}}, \; 0 \right) + \left( 0, \; 0, \; \frac{-abd + b^2 c}{a^2 + b^2}, \; \frac{a^2 d - abc}{a^2 + b^2} \right)$$

where $Q = \frac{1}{\sqrt{a^2 + b^2}} \begin{bmatrix} a & -b \\ b & a \end{bmatrix}$.

The interpolating space is the subspace of $\mathsf{Rep}(\overline{\mathcal{Q}}_L, \mathbf{n}) \simeq \mathbb{R}^4$ with $b = 0$. Suppose $p' = (a, 0, c, d)$ belongs to the interpolating space. Then the gradient is

$$\nabla_{p'} \mathcal{L} = (c, d, a, 0)$$

---

[12]For $\mathbf{W} \in \mathsf{Rep}(\overline{\mathcal{Q}}_L, \mathbf{n})$, the neural function of the neural network with affine maps determined by $\mathbf{W}$ and identity activation functions is $\mathbb{R} \to \mathbb{R}$; $x \mapsto \mathcal{L}(\mathbf{W})x$. The function $\mathcal{L}$ can appear as a loss function for certain batches of training data and cost function on $\mathbb{R}$.

which does not belong to the interpolating space. So one step of usual gradient descent, with learning rate $\eta > 0$ yields:
$$\gamma : p' = (a, 0, c, d) \mapsto (a - \eta c, \ -\eta d, \ c - \eta a, \ d)$$
On the other hand, one step of projected gradient descent yields:
$$\gamma_{\text{proj}} : p' = (a, b, c, d) \mapsto (a - \eta c, \ 0, \ c - \eta a, \ d)$$

The difference between the evaluation of $\mathcal{L}$ after one step of gradient descent and the evaluation of $\mathcal{L}$ after one step of projected gradient descent is:
$$\mathcal{L}(\gamma(p')) - \mathcal{L}(\gamma_{\text{proj}}(p')) = ac - \eta(a^2 + c^2 + d^2) + \eta^2(ac) - ac - \eta(a^2 + c^2) - \eta^2(ac) = \eta d^2.$$

## H.2  GENERALIZATION: RADIAL NEURAL NETWORKS WITH SHIFTS

In this section we consider radial neural networks with an extra trainable parameter in each layer which shifts the radial function. Adding such parameters allows for more flexibility in the model, and (as shown in Theorem H.4) the QR decomposition of Theorem 4.3 holds for such radial neural networks.

Let $h : \mathbb{R} \to \mathbb{R}$ be a function[13]. For any $n \geq 1$ and any $t \in \mathbb{R}$, the corresponding **shifted radial function** on $\mathbb{R}^n$ is given by:
$$h^{(n,t)} : v \mapsto \frac{h(|v| - t)}{|v|} v.$$

The following definition is a modification of Definition 3.2.

**Definition H.3.** A **radial neural network with shifts** consists of the following data:

1. **Hyperparameters.** A positive integer $L$ and a dimension vector $\mathbf{n} = (n_0, n_1, n_2, \ldots, n_L)$ for the neural quiver $\mathcal{Q}_L$.

2. **Trainable parameters.**

   (a) A representation $\mathbf{W} = (W_1, \ldots, W_L)$ of the quiver $\mathcal{Q}_L$ with dimension vector $\mathbf{n}$. So, for $i = 1, \ldots, L$, we have a matrix $W_i \in \text{Hom}(\mathbb{R}^{1+n_{i-1}}, \mathbb{R}^{n_i})$.

   (b) A vector of shifts $\mathbf{t} = (t_1, t_2, \ldots, t_L) \in \mathbb{R}^L$.

3. **Radial activation functions.** A tuple $\mathbf{h} = (h_1, h_2, \ldots, h_L)$, where $h_i : \mathbb{R} \to \mathbb{R}$. The activation function in the $i$-th layer is given by $a_i = h^{(n_i, t_i)} : \mathbb{R}^{n_i} \to \mathbb{R}^{n_i}$ for $i = 1, \ldots, L$.

The **neural function** of a radial neural network with shifts is defined as:
$$f_{(\mathbf{W}, \mathbf{t}, \mathbf{h})} : \mathbb{R}^{n_0} \to \mathbb{R}^{n_L}; \qquad x \mapsto h^{(n_L, t_L)} \circ \widetilde{W}_L \circ \cdots \circ h^{(n_2, t_2)} \circ \widetilde{W}_2 \circ h^{(n_1, t_1)} \circ \widetilde{W}_1(x)$$

where $\widetilde{W}_i = W_i \circ \text{ext}_{n_{i-1}} : \mathbb{R}^{n_{i-1}} \to \mathbb{R}^{n_i}$ is the affine map corresponding to $W_i$. The trainable parameters form the vector space $\text{Rep}(\mathbf{n}) \oplus \mathbb{R}^L$, and the loss function of a batch of training data is defined as
$$\mathcal{L} = \mathcal{L}_{\mathbb{X}} : \text{Rep}(\mathbf{n}) \oplus \mathbb{R}^L \longrightarrow \mathbb{R}; \qquad (\mathbf{W}, \mathbf{t}) \mapsto \sum_j \mathcal{C}(f_{(\mathbf{W}, \mathbf{t}, \mathbf{h})}(x_j), y_j)$$

We have the gradient descent map:
$$\gamma : \text{Rep}(\mathbf{n}) \oplus \mathbb{R}^L \longrightarrow \text{Rep}(\mathbf{n}) \oplus \mathbb{R}^L$$

which updates the entries of both $\mathbf{W}$ and $\mathbf{t}$. The group $O(\mathbf{n}) = O(n_1) \times \cdots \times O(n_{L-1})$ acts on $\text{Rep}(\mathbf{n})$ as usual (see Section 4.1), and on $\mathbb{R}^L$ trivially. The neural function is unchanged by this action. We conclude that the $O(\mathbf{n})$ action on $\text{Rep}(\mathbf{n}) \oplus \mathbb{R}^L$ commutes with gradient descent $\gamma$.

We now state a generalization of Theorem 4.3 for the case of radial neural networks with shifts. We omit a proof, as it uses the same techniques as the proof of Theorem 4.3.

---

[13]We also assume $h$ is piece-wise differentiable and exclude those $t$ for which the limit $\lim_{r \to 0} \frac{h(r-t)}{r}$ does not exist.

**Theorem H.4.** *Let* $\mathbf{n}$ *be a dimension vector for* $\mathcal{Q}_L$ *and fix functions* $\mathbf{h} = (h_1, \ldots, h_L)$ *as above. For any* $\mathbf{W} \in \mathsf{Rep}(\mathbf{n})$ *there exist:*

$$\mathbf{Q} \in O(\mathbf{n}), \qquad \mathbf{R} = (R_1, \ldots, R_L) \in \mathsf{Rep}(\mathbf{n}^{\mathrm{red}}), \qquad \text{and} \qquad \mathbf{U} \in \mathsf{Rep}(\mathbf{n}),$$

*such that:*

1. *The matrices* $R_1, \ldots, R_{L-1}$ *are upper triangular.*

2. *The following equality holds:* $(\mathbf{W}, \mathbf{t}) = \mathbf{Q} \cdot (\mathbf{R}, \mathbf{t}) + (\mathbf{U}, \mathbf{0})$.

3. *The neural functions defined by* $(\mathbf{W}, \mathbf{t}, \mathbf{h})$ *and* $(\mathbf{R}, \mathbf{t}, \mathbf{h})$ *coincide:* $f_{(\mathbf{W}, \mathbf{t}, \mathbf{h})} = f_{(\mathbf{R}, \mathbf{t}, \mathbf{h})}$.

One can use the output of Algorithm 1 to obtain the $\mathbf{Q}$, $\mathbf{R}$, and $\mathbf{U}$ appearing in Theorem H.4. Theorem 4.7 also generalizes to the setting of radial neural networks with shifts, using projected gradient descent with respect to the subspace $\mathsf{Rep}^{\mathrm{int}}(\mathbf{n}) \oplus \mathbb{R}^L$ of $\mathsf{Rep}(\mathbf{n}) \oplus \mathbb{R}^L$.

# I CATEGORICAL FORMULATION

In this appendix, we summarize a category-theoretic approach toward the main results of the paper. While there is no substantial difference in the proofs, the language of category theory provides conceptual clarity that leads to generalizations of these results. References for category theory include Pierce (1991); Dummit & Foote (2003).

## I.1 THE CATEGORY OF QUIVER REPRESENTATIONS

In this section, we recall the category of representations of a quiver. Background references include Kirillov Jr (2016); Nakajima et al. (1998).

As in Section 3, let $\mathcal{Q} = (I, E)$ be a quiver with source and target maps $s, t : E \to I$, and let $\mathbf{d} : I \to \mathbb{Z}_{\geq 0}$, $i \mapsto d_i$ be a dimension vector for $\mathcal{Q}$. Recall that a **representation** of $\mathcal{Q}$ with dimension vector $\mathbf{d}$ consists of a tuple $\mathbf{A} = (A_e)_{e \in E}$ of linear maps, where

$$A_e : \mathbb{R}^{d_{s(e)}} \to \mathbb{R}^{d_{t(e)}}.$$

Let $\mathbf{A}$ and $\mathbf{B}$ be representations of the quiver $\mathcal{Q}$, with dimension vectors $\mathbf{d} = \dim(\mathbf{A})$ and $\mathbf{k} = \dim(\mathbf{B})$. A **morphism** of representations from $\mathbf{A}$ to $\mathbf{B}$ consists of the data of a linear map

$$\alpha_i : \mathbb{R}^{d_i} \to \mathbb{R}^{k_i},$$

for every $i \in I$, subject to the condition that the following diagram commutes for every $e \in E$:

$$\begin{array}{ccc} \mathbb{R}^{d_{s(e)}} & \xrightarrow{A_e} & \mathbb{R}^{d_{t(e)}} \\ {\scriptstyle \alpha_{s(e)}}\downarrow & & \downarrow{\scriptstyle \alpha_{t(e)}} \\ \mathbb{R}^{k_{s(e)}} & \xrightarrow{B_e} & \mathbb{R}^{k_{t(e)}} \end{array}$$

The resulting category $\mathcal{R}(\mathcal{Q})$ is known as the **category of representations** of $\mathcal{Q}$.

## I.2 THE CATEGORY $\mathcal{I}(\mathcal{Q}_L)$

In this section, we define a certain subcategory $\mathcal{I}(\mathcal{Q}_L)$ of the category $\mathcal{R}(\mathcal{Q}_L)$. Its objects are the same as the objects of $\mathcal{R}(\mathcal{Q}_L)$, that is, representations of the neural quiver, while morphisms in $\mathcal{I}(\mathcal{Q}_L)$ are given by isometries.

Let $L$ be a positive integer, and recall the neural quiver $\mathcal{Q}_L$ from Section 3.2:

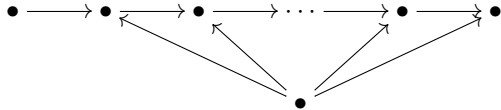

As a reminder, the vertices in the top row are indexed from $i = 0$ to $i = L$, and we only consider dimension vectors whose value at the bias vertex is equal to 1. So a dimension vector for $\mathcal{Q}_L$ will refer to a tuple $\mathbf{n} = (n_0, n_1, \ldots, n_L)$. Recall the isomorphism

$$\mathsf{Rep}(\mathcal{Q}_L, \mathbf{n}) \simeq \bigoplus_{i=1}^{L} \mathsf{Hom}(\mathbb{R}^{1+n_{i-1}}, \mathbb{R}^{n_i})$$

from Lemma 3.1. We denote a representation of $\mathcal{Q}_L$ as a tuple $\mathbf{W} = (W_i)_{i=1}^{L}$, where each $W_i$ belongs to $\mathsf{Hom}(\mathbb{R}^{1+n_{i-1}}, \mathbb{R}^{n_i})$. Let $\mathbf{X} = (X_i)_{i=1}^{L}$ be a representation of $\mathcal{Q}_L$ with dimension vector $\mathbf{m}$. Tracing through the proof of Lemma 3.1 and the definitions in Section I.1, we see that a morphism $\alpha : \mathbf{X} \to \mathbf{W}$ in $\mathcal{R}(\mathcal{Q}_L)$ consists of the following data:

- a linear map $\alpha_i : \mathbb{R}^{m_i} \to \mathbb{R}^{n_i}$, for $i = 0, 1, \ldots, L$, and
- a scalar $\alpha_b \in \mathbb{R}$ at the bias vertex,

making the following diagram commute, for $i = 1, \ldots, L$:

$$
\begin{array}{ccc}
\mathbb{R}^{1+m_{i-1}} & \xrightarrow{\ X_i\ } & \mathbb{R}^{m_i} \\
{\scriptstyle\begin{bmatrix} \alpha_b & 0 \\ 0 & \alpha_{i-1} \end{bmatrix}}\Big\downarrow & & \Big\downarrow{\scriptstyle \alpha_i} \\
\mathbb{R}^{1+n_{i-1}} & \xrightarrow{\ W_i\ } & \mathbb{R}^{n_i}
\end{array}
$$

**Definition I.1.** We define a subcategory $\mathcal{I}(\mathcal{Q}_L)$ of $\mathcal{R}(\mathcal{Q}_L)$ as follows. The objects of $\mathcal{I}(\mathcal{Q}_L)$ are the same as the objects of $\mathcal{R}(\mathcal{Q}_L)$, that is, representations of $\mathcal{Q}_L$. Let $\mathbf{X}$ and $\mathbf{W}$ be such representations, with dimension vectors $\mathbf{m}$ and $\mathbf{n}$, respectively. A morphism $\alpha : \mathbf{X} \to \mathbf{W}$ in $\mathcal{R}(\mathcal{Q}_L)$ belongs to $\mathcal{I}(\mathcal{Q}_L)$ if the following hold:

- $n_0 = m_0$ and $\alpha_0$ is the identity on $\mathbb{R}^{n_0}$,
- $n_L = m_L$ and $\alpha_L$ is the identity on $\mathbb{R}^{n_L}$,
- for $i = 1, \ldots, L-1$, the linear map $\alpha_i : \mathbb{R}^{m_i} \to \mathbb{R}^{n_i}$ is norm-preserving, i.e. $|\alpha_i(v)| = |v|$ for all $v \in \mathbb{R}^{m_i}$, and
- $\alpha_b = 1$.

**Remark I.2.** A norm-preserving map is called an *isometry*, which explains the notation $\mathcal{I}(\mathcal{Q}_L)$.

**Lemma I.3.** *We have:*

1. *The category $\mathcal{I}(\mathcal{Q}_L)$ is a well-defined subcategory of $\mathcal{R}_L$.*

2. *Let $\alpha : \mathbf{X} \to \mathbf{W}$ be a morphism in $\mathcal{I}(\mathcal{Q}_L)$. For $i = 0, 1, \ldots, L$, the linear map $\alpha_i : \mathbb{R}^{m_i} \to \mathbb{R}^{n_i}$ is injective.*

*Proof.* The claims follow from the facts that (1) the composition of two norm-preserving maps is norm-preserving, and (2) any linear norm-preserving map is injective. $\square$

**Definition I.4.** Let $\alpha : \mathbf{X} \to \mathbf{W}$ be a morphism in $\mathcal{I}(\mathcal{Q}_L)$, and let $\mathbf{m} = \dim(\mathbf{X})$ and $\mathbf{n} = \dim(\mathbf{W})$ be the dimension vectors. An **orthogonal factorization** of $\alpha_i$ is an element $\mathbf{Q} = (Q_1, \ldots, Q_{L-1})$ of $O(\mathbf{n}) = O(n_1) \times \cdots \times O(n_{L-1})$ such that

$$\alpha_i = Q_i \circ \mathsf{inc}_{m_i, n_i}$$

for $i = 1, \ldots, L-1$. The **correction term** corresponding to an orthogonal factorization is:

$$\mathbf{U} = \mathbf{W} - \mathbf{Q} \cdot \mathbf{X}.$$

The correction term belongs to $\mathsf{Rep}(\mathcal{Q}_L, \mathbf{n})$.

**Remark I.5.** Orthogonal factorizations always exist, since any norm-preserving linear map $\mathbb{R}^m \to \mathbb{R}^n$ can be written as the composition $Q \circ \mathsf{inc}_{m,n}$ for some orthogonal $Q \in O(n)$.

**Remark I.6.** Suppose $\alpha$ is a morphism in $\mathcal{I}(\mathcal{Q}_L)$. For $i \in \{1, \ldots, L-1\}$, the map $\alpha_i$ is an isomorphism if and only if $m_i = n_i$. In this case, the choice of $Q_i$ is unique and $U_{i+1} = 0$. Conversely, $\alpha_i$ is not an isomorphism if and only if $m_i < n_i$. In this case, there are $O(n_i - m_i)$ choices for $Q_i$.

Fix functions $h_i : \mathbb{R} \to \mathbb{R}$ for $i = 1, \ldots, L$. Hence we obtain radial functions $h_i^{(n)} : \mathbb{R}^n \to \mathbb{R}^n$ for any $n \geq 1$. We group the $h_i$ into a tuple $\mathbf{h} = (h_1, \ldots, h_L)$. Given $\mathbf{h}$, we attach a neural network (and hence a neural function) to every object in $\mathcal{I}(\mathcal{Q}_L)$ as in Construction 4.2.

**Proposition I.7.** *Let $\mathbf{X}$ and $\mathbf{W}$ be a representations of $\mathcal{Q}_L$. Suppose there is a morphism in $\mathcal{I}(\mathcal{Q}_L)$ from $\mathbf{X}$ to $\mathbf{W}$. Then the neural functions of the radial neural networks $(\mathbf{X}, \mathbf{h})$ and $(\mathbf{W}, \mathbf{h})$ coincide:*

$$f_{(\mathbf{X}, \mathbf{h})} = f_{(\mathbf{W}, \mathbf{h})}.$$

*Sketch of proof.* The key is to show that, for $i = 1, \ldots, L$, we have:

$$h_i^{(n_i)} \circ \widetilde{W}_i \circ \begin{bmatrix} 1 & 0 \\ 0 & \alpha_{i-1} \end{bmatrix} = \alpha_i \circ h_i^{(m_i)} \circ \widetilde{X}_i$$

where $\widetilde{W}_i$ and $\widetilde{X}_i$ are the affine map corresponding to $W_i$ and $X_i$. The first step in the verification of this identity is to choose an orthogonal factorization of $\alpha$. The rest of the proof proceeds along the same lines as the proof of Equation E.2 in Section E.2. $\square$

### I.3 PROJECTED GRADIENT DESCENT: SET-UP

In this section, we collect notation necessary to state results about projected gradient descent.

Let $\mathbf{m} = (m_0, \ldots, m_L)$ and $\mathbf{n} = (n_0, \ldots, n_L)$ be dimension vectors for $\mathcal{Q}_L$. We write $\mathbf{m} \preceq \mathbf{n}$ if:

- $m_0 = n_0$ and $m_L = n_L$,
- $m_i \leq n_i$ for $i = 1, \ldots, L-1$.

Consequently, if $\alpha : \mathbf{X} \to \mathbf{W}$ is a morphism in $\mathcal{I}(\mathcal{Q}_L)$, then the dimension vectors satisfy $\dim(\mathbf{X}) \preceq \dim(\mathbf{W})$. For $\mathbf{m} \preceq \mathbf{n}$, we make the following abbreviations, for $i = 0, 1, \ldots, L$:

$$\mathrm{inc}_i = \mathrm{inc}_{m_i, n_i} : \mathbb{R}^{m_i} \hookrightarrow \mathbb{R}^{n_i} \qquad \widetilde{\mathrm{inc}}_i = \mathrm{inc}_{1+m_i, 1+n_i} : \mathbb{R}^{1+m_i} \hookrightarrow \mathbb{R}^{1+n_i}$$

$$\pi_i : \mathbb{R}^{n_i} \twoheadrightarrow \mathbb{R}^{m_i} \qquad \widetilde{\pi}_i : \mathbb{R}^{1+n_i} \twoheadrightarrow \mathbb{R}^{1+m_i}$$

Using these maps, one defines an inclusion $\iota : \mathrm{Rep}(\mathbf{m}) \hookrightarrow \mathrm{Rep}(\mathbf{n})$ taking $\mathbf{X} = (X_1, \ldots, X_L) \in \mathrm{Rep}(\mathbf{m})$ to the representation with $\iota(\mathbf{X})_i = \mathrm{inc}_i \circ X_i \circ \widetilde{\pi}_{i-1}$.

Recall the functions $\mathbf{h} = (h_1, \ldots, h_L)$. As in Construction 4.2, these define activation functions $\left( h_1^{(n_1)}, \ldots, h_L^{(n_L)} \right)$ (resp. $\left( h_1^{(m_1)}, \ldots, h_L^{(m_L)} \right)$) for a representation with dimension $\mathbf{n}$ (resp. $\mathbf{m}$).

Finally, we fix a batch of training data $\mathbb{X} = \{(x_j, y_j)\} \subseteq \mathbb{R}^{n_0} \times \mathbb{R}^{n_L} = \mathbb{R}^{m_0} \times \mathbb{R}^{m_L}$. Using the activation functions defined by $\mathbf{h}$, we have loss functions on $\mathrm{Rep}(\mathbf{n})$ and $\mathrm{Rep}(\mathbf{m})$ (see Section 3.4):

$$\mathcal{L}_\mathbf{n} : \mathrm{Rep}(\mathbf{n}) \to \mathbb{R} \qquad\qquad \mathcal{L}_\mathbf{m} : \mathrm{Rep}(\mathbf{m}) \to \mathbb{R}$$
$$\mathbf{W} \mapsto \sum_j \mathcal{C}(f_{(\mathbf{W}, \mathbf{h})}(x_j), y_j) \qquad\qquad \mathbf{X} \mapsto \sum_j \mathcal{C}(f_{(\mathbf{X}, \mathbf{h})}(x_j), y_j)$$

Fixing a learning rate $\eta > 0$, there are resulting gradient descent maps on $\mathrm{Rep}(\mathbf{n})$ and $\mathrm{Rep}(\mathbf{m})$ given by:

$$\gamma_\mathbf{n} : \mathrm{Rep}(\mathbf{n}) \to \mathrm{Rep}(\mathbf{n}) \qquad\qquad \gamma_\mathbf{m} : \mathrm{Rep}(\mathbf{m}) \to \mathrm{Rep}(\mathbf{m})$$
$$\mathbf{W} \mapsto \mathbf{W} - \eta \cdot \nabla_\mathbf{W} \mathcal{L}_\mathbf{n} \qquad\qquad \mathbf{X} \mapsto \mathbf{X} - \eta \cdot \nabla_\mathbf{X} \mathcal{L}_\mathbf{m}$$

The verification of the following lemma is analogous to the proof of Part 1 of Lemma F.7.

**Lemma I.8.** *We have that $\mathcal{L}_\mathbf{n} \circ \iota = \mathcal{L}_\mathbf{m}$.*

### I.4 THE INTERPOLATING SPACE

We first define a space that interpolates between $\mathsf{Rep}(\mathbf{m})$ and $\mathsf{Rep}(\mathbf{n})$. The discussion of this section is completely analogous to that in Sections F.1 and F.2.

**Definition I.9.** Let $\mathsf{Rep}^{\text{int}}(\mathbf{m}, \mathbf{n})$ denote the subspace of $\mathsf{Rep}(\mathbf{n})$ consisting of those $\mathbf{T} = (T_1, \ldots, T_L) \in \mathsf{Rep}(\mathbf{n})$ such that, for $i = 1, \ldots, L$, we have:

$$(\mathrm{id}_{n_i} - \mathrm{inc}_i \circ \pi_i) \circ T_i \circ \widetilde{\mathrm{inc}}_{i-1} \circ \widetilde{\pi}_{i-1} = 0.$$

Just as in Section F.1, the space $\mathsf{Rep}^{\text{int}}(\mathbf{m}, \mathbf{n})$ consists of representations $\mathbf{T} = (T_1, \ldots, T_L) \in \mathsf{Rep}(\mathbf{n})$ for which the bottom left $(n_i - m_i) \times (1 + n_{i-1}^{\text{red}})$ block of $T_i$ is zero for each $i$. Consider the maps:

$$\mathsf{Rep}(\mathbf{m}) \underset{q_2}{\overset{\iota_2}{\rightleftarrows}} \mathsf{Rep}^{\text{int}}(\mathbf{m}, \mathbf{n}) \underset{q_1}{\overset{\iota_1}{\rightleftarrows}} \mathsf{Rep}(\mathbf{n})$$

- The map $\iota_1 : \mathsf{Rep}^{\text{int}}(\mathbf{m}, \mathbf{n}) \hookrightarrow \mathsf{Rep}(\mathbf{n})$ is the natural inclusion.

- The map $q_1 : \mathsf{Rep}(\mathbf{n}) \to \mathsf{Rep}^{\text{int}}(\mathbf{m}, \mathbf{n})$ takes $\mathbf{T} \in \mathsf{Rep}(\mathbf{n})$ to the representation whose $i$-th slot is
$$W_i - (\mathrm{id}_{n_i} - \mathrm{inc}_i \circ \pi_i) \circ W_i \circ \widetilde{\mathrm{inc}}_{i-1} \circ \widetilde{\pi}_{i-1}.$$

- The map $\iota_2 : \mathsf{Rep}(\mathbf{m}) \hookrightarrow \mathsf{Rep}^{\text{int}}(\mathbf{m}, \mathbf{n})$ is defined by taking $\mathbf{X}$ to the representation whose $i$-th coordinate is $\mathrm{inc}_i \circ X_i \circ \widetilde{\pi}_{i-1}$.

- The map $q_2 : \mathsf{Rep}^{\text{int}}(\mathbf{n}) \twoheadrightarrow \mathsf{Rep}(\mathbf{m})$ is defined by taking $\mathbf{T}$ to the representation whose $i$-th slot is $\pi_i \circ T_i \circ \widetilde{\mathrm{inc}}_{i-1}$.

**Definition I.10.** The projected gradient descent map on $\mathsf{Rep}(\mathbf{n})$ with respect to $\mathsf{Rep}^{\text{int}}(\mathbf{m}, \mathbf{n})$ and learning rate $\eta > 0$ is defined as:

$$\overline{\gamma}_{\mathbf{n}} : \mathsf{Rep}(\mathbf{n}) \to \mathsf{Rep}(\mathbf{n}) \qquad \mathbf{W} \mapsto \mathbf{W} - \eta \cdot i_1 \circ q_1(\nabla_{\mathbf{W}}\mathcal{L})$$

We now state the main result of this appendix. We omit a proof, as it follows the same ideas as the proof of Theorem 4.7 given in Section F.3. The main difference is that all appearances of $\mathbf{n}^{\text{red}}$ must be replaced by $\mathbf{m}$.

**Theorem I.11.** *Let $\alpha : \mathbf{X} \to \mathbf{W}$ be a morphism in $\mathcal{I}(\mathcal{Q}_L)$. Let $\mathbf{Q}$ be an orthogonal factorization of $\alpha$ and set $\mathbf{U} = \mathbf{W} - \mathbf{Q} \cdot \mathbf{X}$.*

1. *The representation $\mathbf{Q}^{-1} \cdot \mathbf{W} = \mathbf{X} + \mathbf{Q}^{-1} \cdot \mathbf{U}$ belongs to $\mathsf{Rep}^{\text{int}}(\mathbf{m}, \mathbf{n})$.*

2. *For any $k \geq 0$, we have*
$$\gamma_{\mathbf{n}}^k(\mathbf{W}) = \mathbf{Q} \cdot \gamma_{\mathbf{n}}^k(\mathbf{Q}^{-1} \cdot \mathbf{W}).$$

3. *For any $k \geq 0$, we have*
$$\overline{\gamma}_{\mathbf{n}}^k(\mathbf{Q}^{-1} \cdot \mathbf{W}) = \gamma_{\mathbf{m}}^k(\mathbf{X}) + \mathbf{Q}^{-1} \cdot \mathbf{U}.$$

### I.5 RELATION TO ALGORITHM 1

In this final section, we relate the general categorical results of this appendix to Algorithm 1. Recall that, in Section 4.2, we associated a reduced dimension vector $\mathbf{n}^{\text{red}}$ to any dimension vector $\mathbf{n}$ of the neural quiver $\mathcal{Q}_L$.

**Proposition I.12.** *Let $\mathbf{W}$ be a representation of $\mathcal{Q}_L$ with dimension vector $\mathbf{n}$. Then $\mathbf{W}$ admits a morphism in $\mathcal{I}(\mathcal{Q}_L)$ from a representation with dimension vector $\mathbf{n}^{\text{red}}$.*

*Proof.* We proceed by induction on the number $N$ of $i \in \{1, \ldots, L-1\}$ such that $n_i^{\mathrm{red}} < n_i$. If $N = 0$, there is nothing to show. Otherwise, let $j$ be the smallest element of $\{1, \ldots, L-1\}$ such that $n_j^{\mathrm{red}} < n_j$. Then $n_j^{\mathrm{red}} = n_{j-1} + 1$. Let $W_j = Q \circ \mathrm{inc}_{1+n_{j-1}, n_j} \circ R$ be a QR decomposition of $W$, where $Q \in O(n_j)$ and $R$ is an upper-triangular $n_{j-1} + 1$ by $n_{j-1} + 1$ matrix. Consider the representation of $\mathcal{Q}_L$ given by:

$$\mathbf{X} = \left( \, W_1 \, , \, \ldots \, , \, W_{j-1} \, , \, R \, , \, W_{j+1} \circ \begin{bmatrix} 1 & 0 \\ 0 & Q \end{bmatrix} \circ \mathrm{inc}_{1+n_{j-1}, n_j}, \, W_{j+2} \, , \, \ldots \, , W_L \, \right)$$

Then

$$\alpha = (\, \mathrm{id}_{n_1} \, , \, \ldots \, , \, \mathrm{id}_{n_{j-1}} \, , \, Q \circ \mathrm{inc}_{1+n_{j-1}, n_j} \, , \, \mathrm{id}_{n_{j+1}} \, , \, \ldots \, , \, \mathrm{id}_{n_L})$$

is a morphism form $\mathbf{X}$ to $\mathbf{W}$. The dimension vector of $\mathbf{X}$ is $(n_0^{\mathrm{red}}, \ldots, n_{j-1}^{\mathrm{red}}, n_j^{\mathrm{red}}, n_{j+1}, \ldots, n_L)$ and has one less coordinate that $\mathbf{n}$ not equal to $\mathbf{n}^{\mathrm{red}}$, so the induction hypothesis applies. $\qquad \square$

Consequently, Theorems 4.3 and 4.7 can be viewed as a corollaries of Proposition I.7, Theorem I.11, and Proposition I.12.

**Remark I.13.** Let $\mathbf{Q}$, $\mathbf{R}$ and $\mathbf{U}$ be the outputs of Algorithm 1 applied to a representation $\mathbf{W}$ of dimension vector $\mathbf{n}$. Then $\mathbf{R}$ defines a representation of dimension vector $\mathbf{n}^{\mathrm{red}}$, and $\mathbf{Q}$ defines a morphism in $\mathcal{I}(\mathcal{Q}_L)$ from $\mathbf{R}$ to $\mathbf{W}$. Indeed, for $i = 1, \ldots, L$, Algorithm 1 provides the equality

$$W_i \circ \begin{bmatrix} 1 & 0 \\ 0 & Q_{i-1} \end{bmatrix} \circ \widetilde{\mathrm{inc}}_{i-1} = V_i = Q_i \circ \mathrm{inc}_i \circ R_i,$$

(where $Q_0 = \mathrm{id}_{n_0} = \mathrm{inc}_0$ and $Q_L = \mathrm{id}_{n_L} = \mathrm{inc}_L$), which implies that the following diagram commutes:

$$
\begin{array}{ccc}
\mathbb{R}^{1+n_{i-1}^{\mathrm{red}}} & \xrightarrow{\quad R_i \quad} & \mathbb{R}^{n_i^{\mathrm{red}}} \\
{\scriptsize \begin{bmatrix} 1 & 0 \\ 0 & Q_{i-1} \end{bmatrix} \circ \widetilde{\mathrm{inc}}_{i-1}} \Big\downarrow & & \Big\downarrow {\scriptstyle Q_i \circ \mathrm{inc}_i} \\
\mathbb{R}^{1+n_{i-1}} & \xrightarrow{\quad W_i \quad} & \mathbb{R}^{n_i}
\end{array}
$$

