# OpenReview forum: "Model Compression via Symmetries of the Parameter Space"
_ICLR.cc/2022/Conference — ICLR 2022 Submitted_

### Official Review · Reviewer_uaQ7 · 2021-10-29

**Correctness:** 3
**Technical Novelty And Significance:** 3
**Empirical Novelty And Significance:** 3
**Recommendation:** 5
**Confidence:** 4

**Main Review:**

# Comments on novelty and related work

What the authors regard as reduced widths, which represent the dimensions of a radial neural network after compression, has an evident relationship to dimensionality loss of the affine transformations along the neural network. I believe that the authors should point out at related work on neural networks regarding the bottleneck effect on the outputs caused by the width and number of activated ReLUs [1].

The paper somewhat implies this work as the first lossless compression approach of neural networks: "leading to a lossless model compression algorithm [...] Whereas previous approaches to model compression are based on weight pruning, quantization, matrix factorization, or knowledge distillation." However, there were three papers published on the topic in the last couple of years [2-4].

The comment regarding the classic LTH paper is inaccurate: "Recent work has shown pruning can be done before training by identifying certain key network structures (Frankle & Carbin, 2018)." In fact, that first LTH paper does pruning while training by (1) doing a few steps of training, (2) identifying lesser relevant weights, and (3) rewinding to the original parameters while excluding those parameters, which is not the same thing as pruning before training. However, the authors may (and perhaps should) cite subsequent work that does prune before training [5-8].

As a survey, Blalock et al. (2020) also describes gradient-based pruning as the typical (and competitive) alternative to weight pruning. That stream of work is classic and widely known, but it is completely ignored by the authors [7,9-16].

# Comments on improving the paper

I could not understand the necessity and relevance of quivers for what the authors did. I am not saying that there is not one, and in fact deep learning has benefited a lot from the many approaches brought in from other disciplines, but I believe that the paper would benefit immensely from an even smoother introduction of the topics that are not commonly used in our literature. To mention one example of that being done in a masterful way, I would recommend that the authors look at how [17] describes tropical algebra before applying it to neural network.

In Example 5.1, for the shifted ReLU as h(r) = ReLU(r-b) for b positive, I believe you should be explicit about r being nonnegative. By checking the reference cited, I actually found out the definition of a Norm-ReLU as h(|f(x)|) = ReLU(|f(x)|-b) for b positive, which is actually easier to understand.

I also missed some figures illustrating what is it that makes an activation useful for your approach. I spent some time playing around with your definitions to make sense of it, which is how I got some intuition for the results that you obtained and why they should be valid. Please consider including examples like this one:

https://www.wolframalpha.com/input/?i=plot+%7C%7Cx%7C+-+5%7C

While playing with it, I also started wondering that perhaps there is no need to require b to be positive in the case of what you call a shifted ReLU. Is it not true that your approach would work with the function below?

https://www.wolframalpha.com/input/?i=plot+%7C%7Cx%7C+%2B1%7C

# References cited

[1] https://arxiv.org/abs/1711.02114

[2] https://arxiv.org/abs/2001.00218

[3] https://arxiv.org/abs/2007.06567

[4] https://arxiv.org/abs/2102.07804

[5] https://arxiv.org/abs/1810.02340

[6] https://arxiv.org/abs/1906.06307

[7] https://arxiv.org/abs/2002.07376

[8] https://arxiv.org/abs/2006.05467

[9] https://proceedings.neurips.cc/paper/1989/file/6c9882bbac1c7093bd25041881277658-Paper.pdf

[10] https://proceedings.neurips.cc/paper/1992/file/303ed4c69846ab36c2904d3ba8573050-Paper.pdf

[11] https://arxiv.org/abs/1506.02515

[12] https://arxiv.org/abs/1611.06440

[13] https://arxiv.org/abs/1705.07565

[14] https://arxiv.org/abs/1810.02340

[15] https://arxiv.org/abs/1905.05934

[16] https://arxiv.org/abs/2004.14340

[17] https://arxiv.org/abs/1805.07091

**Summary Of The Paper:**

This paper presents a dimensionality reduction technique that can preserve the model of a neural network by leveraging radial symmetry. Consequently, when the required conditions apply, they also show that training the compressed model is equivalent to training the original model on a particular projection of the parameter space.

The first two sections of the paper are very clear and well written, although some definitions would have helped (more on that later), but the sudden change of pace starting at section 3 makes it difficult to follow if you do not have the same background. I believe that I have captured the gist of their results, which seem to be intuitively correct, but I cannot comment on how they were derived. Hence, what follows is a description in my own words and perspective of what they did.

The authors present a method that would reduce the width of each layer of the neural network to the smallest width of any preceding layer, including the input layer. That is possible in radial neural networks because (1) the activation function involves the entire layer as opposed to each neuron individually and (2) the activation function is symmetric along any direction considered.

**Summary Of The Review:**

# Final comments

There are two items affecting my score at this point:

1) How applicable is this result? In other words, how often would we consider using radial neural networks in practice, or how much of this paper could be applied in a broader sense?

2) It is not easy to follow this paper when it gets more technical. I know that many reviewers would not be willing to reassess their score if the authors put the effort of rewriting the paper considerably, but you have my word that I would reread the paper and reconsider my assessment in case you do that.

---

> ### Author Response · Authors · 2021-11-20
> **Response to specific questions**
>
>
> We found the reviewer's comments to be particularly  insightful, straightforward, and  helpful. Implementing the suggested changes has certainly improved the paper.
>
> **Clarification about the reduction**
>
> We would like to provide clarification on a (relatively minor) point about the reduced network. The reviewer claims that our algorithm "would reduce the width of each layer of the neural network to the smallest width of any preceding layer, including the input layer." This is indeed the case when there are no biases in the network. However, in the general case, the rule is:  if  the width of a non-output layer is larger than  the smallest width of any preceding layer plus the number of layers in between, then reduce the width to that value.  For example, if the layer widths are (10, 15, 25, 10, 5, 10), then the reduced layers are (10, 11, 12, 10, 5, 10).
>
>
> **Comments on novelty and related work**
>
> We are grateful that the reviewer shared their expertise and familiarity with the literature.
>
> > ...should point out at related work on neural networks regarding the bottleneck effect on the outputs caused by the width and number of activated ReLUs [1].
>
> Thank you for pointing out this work.  The bottleneck condition they find certainly bears a strong similarity to our condition on dimensional reduction depending on both the width of earlier layers and number of layers between.
>
> > The paper somewhat implies this work as the first lossless compression approach... However, there were three papers published on the topic in the last couple of years [2-4].
>
> We have reworded this section.  We certainly do not want to imply we are first to consider a lossless algorithm.  Thank you for the additional references which we have added.
>
> > The comment regarding the classic LTH paper is inaccurate
>
> We have made this correction and added the additional suggested references.
>
> **Improving the paper: smoother mathematical transitions**
>
> The reviewer's comments about an abrupt transitions of mathematical abstraction are justified. We have updated the paper appropriately, and say more about this in the common reply.
>
> **Improving the paper: figures illustrating the activation functions**
>
> Thank you for taking the time to understand this aspect of our paper.  We've added the restriction $r>0$.  Excellent point that $b$ does not need to be positive.  Negative $b$ still provides ample non-linearity when applied radially, for example, $\sigma(v) = v ((|v| + 2)/|v|).$  We have also added a figure illustrating different radial activation functions.
>
> **Final comments**
>
> We address the applicability of our results in the common reply. Also, we are very appreciative of the reviewer's willingness to reassess their score. We have indeed made an effort to rewrite the paper in order to make the technical sections easier to follow.

---

> > ### Comment · Reviewer_uaQ7 · 2021-11-22
> > **Feedback on updated manuscript**
> >
> > I am glad that the reviewers found the comments useful and were able to improve the work accordingly.
> >
> > I also appreciate the use of another color to emphasize what was changed. Speaking of which, where you say "r > 0 a real number b" in Example 4.1, is an "and" missing after "r > 0"?
> >
> > I still have two concerns after the updates:
> >
> > 1) Although a few more paragraphs were added with the revision, they mostly remain in line with the heavy notation starting in Section 3. I am still not able to see what is it that makes the quiver representation more convenient to obtain your results. Note that many papers will send a lot of notation and heavy mathematical discussions to the appendix to use the paper itself for a more holistic discussion. I fell that you need to go a bit further in that direction to make it easier to understand your methods.
> >
> > 2) The comments about applicability in the general response do not address my point, which is the following: are radial activation functions being used in practice, or should we expect them to be used in the future for any particular reason?

---

> > > ### Author Response · Authors · 2021-11-23
> > > **Response to new comments**
> > >
> > > Thanks for catching the typo, and for the feedback.
> > >
> > > Unfortunately we don't have time to update the paper based on your concerns (we were a bit dilatory with our initial reply). Just a few words:
> > >
> > > 1. Thanks for pointing out that more drastic change is needed to make the paper more accessible. As to what makes the quiver representation perspective more convenient, the short answer is that it (1) captures the underlying neural network architecture is as simple a way as possible, (2) packages the trainable parameters of a neural network into a vector space, and (3) emphasizes the parameter space symmetries which are crucial for our results. In later versions of the paper, we will try to hit this point harder.
> > >
> > > 2. It's definitely a fair point that the applicability of radial neural networks is a big unknown at the moment. As theoreticians rather than practitioners, our intuition is lacking here. We are looking into this concern, but don't have much specific to say at the moment.

---

> > > > ### Author Response · Authors · 2021-11-28
> > > > **Radial Activations**
> > > >
> > > > Radial activations are being used now.  The examples we cite come from [1].  However, they are also used in [2, Sec 4.3], [3],  [4] and others.  All of these works consider rotational equivariance and choose radial activation functions since they are equivariant with respect to the rotation group $SO(2)$ or $SO(3)$.
> > > >
> > > > A distinction between these works and our setting is that they consider only the 2-dimensional or 3-dimensional rotation group acting separately at each point in a point cloud or pixel in a grid and separately on each irrep.    In contrast, we consider the group $O(n)$ acting on the entire layer $\mathbb{R}^n$ at once.  Radial activations could be used to build $O(n)$-equivariant networks, but may have other uses as well.
> > > >
> > > >  - [1] Weiler, Maurice, and Gabriele Cesa. "General $ E (2) $-Equivariant Steerable CNNs." NeurIPS (2019).
> > > >  - [2] Thomas, Nathaniel, et al. "Tensor field networks: Rotation-and translation-equivariant neural networks for 3d point clouds." arXiv preprint arXiv:1802.08219 (2018).
> > > > - [3] Worrall, Daniel E., et al. "Harmonic networks: Deep translation and rotation equivariance." Proceedings of the IEEE Conference on Computer Vision and Pattern Recognition. 2017.
> > > > - [4] Fuchs, Fabian, et al. "SE (3)-Transformers: 3D Roto-Translation Equivariant Attention Networks." Advances in Neural Information Processing Systems 33 (2020).

---

> > > > > ### Comment · Reviewer_uaQ7 · 2021-11-30
> > > > > **Following up**
> > > > >
> > > > > I appreciate the follow up regarding the applications of radial networks in the literature.

---

### Official Review · Reviewer_EGB6 · 2021-10-31

**Correctness:** 3
**Technical Novelty And Significance:** 2
**Empirical Novelty And Significance:** 2
**Recommendation:** 3
**Confidence:** 5

**Main Review:**

The proposed neural quiver is a nice application of matrix representations in NNs, which have been studied in the literature as reviewed in the paper. The proposed paper employs them to study compression of models. However, the proposed theoretical and experimental results should be improved to justify the proposed main claim, which is proposing a framework for efficient model compression.

To sum up, there are two major issues with the paper:

First, the proposed main theoretical results have been studied in various previous works for training NNs on Stiefel or low-rank matrices. In addition, additional works employed more efficient retractions (e.g. Cholesky decomp.) than QR decomposition for optimization (e.g. projected or Riemannian GD) on low-rank matrices or orthonormal matrices. Therefore, the theoretical novelty of the proposed method is limited.

Second, the main claim (also the title) of the paper is about model compression. Therefore, the proposed methods should be compared with the other model compression methods (at least, with those which employ PGD on low-rank parameters as proposed in this paper) experimentally on benchmark datasets in detail.

Some minor comments:

In Fig. 1, is the bias shared among layers? If it is, then does the proposed neural quiver represent a particular class of NNs?

Please define R in Theorem 1. How do you compress W to obtain R?



**Summary Of The Paper:**

This paper provides a framework to study compression of NN models by analyses of symmetry properties of their parameter spaces. For lossless model compression, an algorithm employing QR decomposition for parameter compression is proposed. In the theoretical analyses, the decomposition of the proposed neural quiver is studied. The proposed methods were experimentally analyses in synthetic datasets.

**Summary Of The Review:**

The paper proposes a theoretical framework to study compression of NNs employing the representation theory of quivers. However, the proposed theory and its experimental analyses are incomplete.

To improve theoretical results, I recommend first explicating the theoretical results in comparison with the related work. Second, experimental analyses should be extended by applying the theoretical framework for the target model compression tasks in comparison with the related state-of-the-art.

---

> ### Author Response · Authors · 2021-11-20
> **Response to Specific Questions**
>
>
> We are grateful for the feedback provided by this reviewer, particularly the comments about the relation to other work, and the specific questions about our set-up and results.
>
> **Previous works**
>
> We have expanded our related works section to address the comments of the reviewer. However,  as far as we can tell, none of the previous works mentioned by the reviewer consider radial activation functions. This is the distinguishing feature of our work. To make sure that we are not mistaken, we would be happy to take a look at any specific references the reviewer can  provide, especially ones whose results may overlap with ours.
>
> **Model compression**
>
> We agree that there would be value in more experimental comparisons of our method with other model compression methods. However, we would  like to emphasize that our work is theoretical. Our goal is to provide a foundational mathematical viewpoint on neural networks, and explain how this framework leads to a particular model compression method.  From this point of view, our methods merit significance even without extensive comparative studies and experimental evidence.
>
> **Question about the bias**
>
> The bias is not shared among the layers in Figure 1.  To clarify this point, note that a vector in the vector space $\mathbb{R}^n$ is the same as a linear map from the one-dimensional space $\\mathbb{R}$ to $\\mathbb{R}^n$. Indeed, such a map is determined by where $1 \in \\mathbb{R}$ goes.  Therefore, having a bias in each layer means having a linear map from $\\mathbb{R}$ to each $\\mathbb{R}^{n_i}$. Since each of these maps has source $\\mathbb{R}$, we have a single bias vertex (with dimension vector equal to 1) and an arrow from the bias vertex to each layer.
>
> Another perspective on this point is as follows. Consider the following quiver:
>
> $$\\require{AMScd}
> \\begin{CD}
> \\bullet @>>>  \\bullet @>>>  \\bullet @>>> \\ldots @>>> \\bullet @>>> \\bullet \\\\
>      @.  @AAA @AAA @. @AAA @AAA \\\\
>      @.   \\bullet @. \\bullet @. @. \\bullet @. \\bullet
> \\end{CD}$$
>
> This quiver is like the neural quiver, but has a separate bias vertex for each non-input vertex. So instead of just one vertex on the bottom, there is a row of L bias vertices, each connected to the non-input vertex directly above in the top row. Consider only representations whose value at each bias vertex is equal to 1.  Any such dimension vector also defines a dimension vector for the neural quiver, and the corresponding space of representations are isomorphic. Hence, one can just as well consider the above quiver instead of the neural quiver; we find the latter to be simpler.
>
> **Question about $\mathbf{R}$**
>
> The way $\mathbf{R}$ is obtained from $\mathbf{W}$ is via Algorithm 1. While the algorithm is relatively straightforward (and, we hope, well-explained) in the Section ``Main Theoretical Results'', we decided not to elaborate on the details of the compression algorithm in the introduction. This is in order to avoid excessive technicalities, and to keep the introduction succinct and focused. However, to address the reviewer's concern, we did add some explanation to  the introduction to say that the main algorithm depends on successive QR decompositions, and to include a link  to Algorithm 1.

---

### Official Review · Reviewer_6ipT · 2021-11-02

**Correctness:** 4
**Technical Novelty And Significance:** 3
**Empirical Novelty And Significance:** 2
**Recommendation:** 3
**Confidence:** 3

**Main Review:**

Overall, I found the paper well-written, and presenting an seemingly interesting observation on neural networks whose activation function commutes with change of basis (orthogonal linear maps). However, I have serious concerns regarding the “radial” assumption on the non-linearity, and I was not convinced by the utility of the introduction of an elaborate mathematical framework to demonstrate a phenomenon which can simply (and plainly) be explained through some standard linear algebra. I believe that the authors could greatly improve the paper by either 1) demonstrating some further insight from the view of the network as a representation of a quiver, or 2) writing the paper in straightforward fashion through standard linear algebra arguments. Additionally, I feel that the overall results in this paper are not surprising (given the assumption on the activation functions), and I fail to see how such a phenomenon may generalize to typical activations encountered in practice. Detailed comments can be found below.

**On the “radial activation” assumption and generalizing to further activation functions.**
The authors introduce an assumption on the activation function that it is “radial”. At first glance, this seems to me to render the network essentially linear up to a scaling factor. Indeed, the radial assumption means that $\sigma(x) = ax$ for some scalar $a$ (which may depend on $x$), and thus for a two layer with weights $W_1, W_2$ we have $W_2 \sigma(W_1 x) = a W_2 W_1 x$, and the network is thus a deep linear networks with some (scalar) scaling factor. Given this fact, it is not surprising that a reduced representation is possible, but this identity also significantly jeopardizes the expressive power of the network. The authors indicate that they expect their work to extend “non-radial activation functions”. I believe that such an extension is essential given the strong limits of the radial assumption. However, it also appears to me that the assumption that the activation commutes with orthogonal linear transforms is central to the proposed procedure, which places strong constraints on the expressivity of the network and does not seem generalizable to networks used in practice.

**On the mathematical formalism and arguments.**
The authors make use of a fairly new formalism of viewing neural networks as representation of quivers. However, it is not clear to me that this view provides any specific insight into the claims of the paper, and it seems that it would only hinder accessibility of the results to a wider audience. Indeed, I believe that the claimed result can be obtained (in a very similar fashion) through a proof by induction (on the number of layers) and standard linear algebra arguments. Given this, I believe that the authors should either leverage the proposed formalism to obtain insights specific to that formalism, or abandon the formalism altogether.

**On the gradient algorithm.**
In addition to the reduced representation, the authors show that projected gradient descent as applied to the reduced representation is equivalent to gradient descent in the original space, due to the fact that the gradient commutes with orthogonal transforms. However, this is not the case for many methods used in practice (e.g. Adam, RMSprop etc.), which explicitly depend on the choice of coordinates as they compute axis-aligned normalization quantities. It would be helpful for the authors to address these practicalities if possible.

Typos: p. 3: Clebsh-Gordan

**Summary Of The Paper:**

The authors make use of a theoretical framework for viewing neural networks as representation of quivers to introduce a reparametrization strategy for neural networks with radial activation functions. This reparametrization is based on a QR decomposition, and leads to a lossless compression of the number of parameters by factoring redundant symmetries of the model. A corresponding gradient descent algorithm is derived on the compressed parameter space corresponding to gradient descent in the original space.

**Summary Of The Review:**

The paper presents some potentially interesting ideas. However, I believe that the unnecessary usage of an esoteric formalism and the constraining assumptions do not warrant publication at this time.

---

> ### Author Response · Authors · 2021-11-20
> **Answers to Specific Questions**
>
> We thank the reviewer for their thoughtful review, questions, and suggestions for improvement.
>
> Many comments appearing in this review, while reflecting relevant concerns,  are predicated on a possible misunderstanding about radial neural networks. In order to make sure we're all on the same page, we would like to begin with some clarifications. We then address  the reviewer's specific remarks and suggestions.
>
> **Clarification about radial neural networks**
>
> The reviewer claims that a radial neural network is "essentially linear up to a scaling factor". We agree with this claim in the special case of no biases, since scaling factors commute with linear maps. However, the reviewer's observation is incorrect when biases are included, as in our main results. This comes down to the fact that scaling factors do not commute with affine maps.
>
> For example, take a two-layer radial neural network with weight matrices $W_1$ and $W_2$, biases $b_1$ and $b_2$, and radial activation functions in both hidden layers. Then the feedforward function is:
> $$x \mapsto p_1(x) \cdot W_2 ( W_1 x  + b_1) + p_2(x) \cdot b_2$$
> for some scalar-valued functions $p_1$ and $p_2$ of the input vector $x$.
>
> In general, with $L$ layers, the value of feedforward function at an input vector $x$ is a linear combination of the $L$ vectors
> $$W_L W_{L-1} \cdots W_2 (W_1 x + b_1),  W_L W_{L-1} \cdots W_3 b_2,  \dots ,  W_L b_{L-1},  b_L$$
> where the cofficients are given by different scalar-valued functions of $x$. This is manifestly more elaborate than a deep affine network with a nonlinear rescaling. We have not done a detailed analysis of any possible interdependencies among the coefficient functions; still, these functions are all different, and there is a high degree of flexibility when choosing them.
>
> **Radial assumption and generalizing to further activation functions**
>
> With the above discussion in hand, we address several of the other comments appearing in the review.
>
> - Expressivity: We agree with the reviewer that the no-bias case is not expressive and the results in that case are not very meaningful. However, as noted above, the general case has much higher expressive power.
>
> - Surprisingness: Our results may well not be particularly surprising in some cases, particularly the no-bias case that the reviewer mentions. However, the general case (with biases included) is significantly more intricate. While the theorem statements may be intuitive, the proofs require insight and attention. Additionally, besides the work (Sourek et al, 2020) pointed out by another reviewer, we have seen few works taking advantage of symmetries of the parameter space to perform model compression.
>
> - Generalization to non-radial activations: First, the radial assumption may not be as limiting as the reviewer indicates (see above), so our results are still significant even without further generalization. Second, there are some alternative activation functions that may lead to model compression via parameter space symmetry:
>
>     1. ReLU. The symmetry group is a product of copies of $\mathbb{R}_{\geq 0}$, one for every hidden neuron (across all layers), and symmetric groups for permuting the neurons in each hidden layer.  Breaking the symmetry of the symmetric group could lead to more efficient optimization.
>     2.  There is also the case of radial activations in the context of steerable CNNs which admit an $O(2)^c \ltimes S_c$ symmetry group. This symmetry can also result in dimensional reduction.
>
> The reference cited above is: Gustav Sourek, Filip Zelezny, and Ondrej Kuzelka. Lossless compression of structured convolutional models via lifting. arXiv preprint arXiv:2007.06567, 2020.
>
> **Mathematical formalism**
>
> We address the reviewer's comments about an excess of mathematical formalism in the common reply. Additionally, we agree that special cases of our main results (such as the no-bias case) can be obtained through standard linear algebra arguments. To some extent this is also true in the general case. Even so, we believe our proofs provide the appropriate level of abstraction for conceptual clarity, and we have left this formalism in the appendix.
>
> **On the gradient descent algorithm**
>
> The reviewer is correct to point out possible incompatibilities with common gradient descent methods. For example, the element-wise squaring step of Adam depends on coordinates, so does not interact nicely with our QR decomposition.  We have added this observation to the paper.
>
> **Typo**
>
> It is indeed "Clebsch-Gordan", not "Clebsch-Gordon" or "Clebsh-Gordan".

---

### Author Response · Authors · 2021-11-20
**Common Questions; Summary of Revisions**

We are very grateful for the responses of the reviewers. Their valuable comments have helped us significantly improve the paper.

**Common questions**

We post here some replies to common reviewer questions and concerns.  More specific questions are addressed in responses to each individual reviewer.

- The reviewers' comments about an excess of mathematical formalism are justified. Upon rereading our manuscript, we agree that various notation is extraneous and distracting. Therefore, we have rewritten sections of the paper to ensure greater accessibility, and smoother transitions into more technical material.
- To address concerns about the limitations of the quiver perspectives, we restructured our definition of neural networks to emphasize that it is a special case of a vastly more general procedure. We  elaborate on the general framework in the appendix. Briefly, for any acyclic quiver $Q$ satisfying some conditions, one can consider $Q$-neural networks, whose underlying architecture is governed by $Q$. We also briefly illustrate how this formalism captures residual connections, for example.  (See the new version of Figure 1.)
- In terms of the applicability of our results, it is true that there are still various unknowns about radial neural networks, as they have not been extensively studied. We view this paper as a step in exploring a potentially  vast new area of research. By establishing theoretical foundations, we hope that our results generalize in practical ways.   For example, ultimately, the loss function of the neural network corresponds to a function on a certain moduli space, known as a quiver variety, that appears as a quotient of the space of representations of the quiver. The aim of model compression is to reduce to this space; one can then examine convexity properties of the loss function on this space.

**List of revisions**

We have made many revisions to the draft which are highlighted in blue for convenience and which we summarize here:

- Removed "Preliminaries" Section, absorbing the material appearing in that section to places where it is immediately relevant.
- Included clarifications about quivers and quiver representations, and a smoother transition to the technical parts of the paper.
- Simplified or standardized mathematical notation where possible.
- Added clarifications in the introduction near  Theorem 1.1 to indicate that $\bf R$ is obtained  from $\bf W$ via an algorithm depending on successive QR decompositions that appears later in the paper.
- Expanded the  "Model compression and weight pruning" paragraph of the related works section.
- Emphasized that our definition of neural networks (Definition 3.2) is a special case of a more general phenomenon.
- Added Figure 2 to illustrate the radial activation functions.
- Included a new appendix (Appendix G) to explain quiver neural networks in general.

---

### Decision · Program_Chairs · 2022-01-20

**Decision:**

Reject

**Comment:**

Authors developed a reparameterization scheme using QR decomposition to reveal symmetries in networks with radial activation. While I welcome new ideas and formalisms from other fields, the ideas presented in this manuscript are fairly straightforward under the radial activation assumption. Although the authors claim that the results may generalize, no evidence was provided. The practical contributions are marginal and lacks comparisons with related DNN compression schemes. Through the review process, the paper has been greatly improved, but unfortunately, this interesting paper does not meet ICLR's standard as is.